# IEC: When Information-Driven Exploration Meets Spectral Consensus via Primal–Dual Reward Regularization in Decentralized MARL

**Xuefeng Du** [* 1 2]  **Jiajun Wu** [* 2 3]  **Yuduo Zheng** [* 1 2]  **Fengqi Li** [2 3]

## Abstract

Decentralized multi-agent reinforcement learning faces a persistent exploration–coordination tension: intrinsic rewards promote exploration under sparse feedback, yet effective cooperation requires agents' behaviors to remain consistent over a limited communication graph. Existing methods often combine exploration bonuses and coordination regularizers with fixed-weight schedules, making them hard to tune and prone to either fragmented conventions or premature behavioral collapse. We propose the IEC (Isomorphic Exploration-Consensus) framework that couples exploration and coordination through a single constrained objective: maximize task return augmented with two complementary exploration signals, dynamics-based information gain and state-coverage novelty, while constraining graph-induced policy disagreement via a spectral smoothness penalty on neighboring agents, which can be interpreted as a Dirichlet-energy regularizer on the communication graph. IEC optimizes the resulting Lagrangian with a lightweight primal–dual update that adapts the consensus multiplier from observed constraint violations, yielding an automatic shift from diverse exploration to stable cooperative conventions. Across three distinct benchmarks, IEC achieves superior performance.

## 1. Introduction

Multi-agent reinforcement learning is increasingly deployed in distributed systems that prohibit centralized coordination (Jiang et al., 2024b). Decentralized multi-agent reinforce-ment learning (Dec-MARL) studies how a team of agents can learn cooperative behaviors (Jiang & Lu, 2018) by coordinating through limited, graph-structured communication (Nayak et al., 2023). This setting is particularly challenging when tasks provide sparse or delayed rewards and communication is bandwidth-limited or unreliable.

A core difficulty lies in the exploration–coordination tension inherent to decentralized learning. On the one hand, sparse feedback often necessitates information-driven exploration (Jarrett et al., 2022; Saade et al., 2023): agents must visit uncertain states and probe poorly understood transitions to acquire epistemic knowledge about the environment (Xu & Liu, 2023). On the other hand, effective cooperation requires behavioral consistency (Lowe et al., 2017): agents must converge to compatible conventions that remain coherent across the communication graph, since local disagreements can propagate into global coordination failures (Ding et al., 2024). In practice, excessive exploration can fragment the team into incompatible local behaviors, whereas overly strong early coordination can cause premature collapse into suboptimal conventions.

Existing Dec-MARL pipelines typically address this tension by combining largely independent components: intrinsic rewards for exploration (e.g., curiosity (Jarrett et al., 2022) or novelty-based bonuses (Raileanu & Rocktäschel, 2020)) and regularizers or communication mechanisms for coordination. While effective in isolation, these components are most often coupled through fixed coefficients or hand-crafted schedules (as Fig. 1), making performance brittle and highly sensitive to tuning across tasks, topologies, and training phases. Related work has shown that explicit policy regularization can promote coordination and that Lagrange multipliers provide a principled mechanism for enforcing constraints (Montenegro et al., 2024; Sohrabi et al., 2024); however, these tools have not been used to directly couple information-driven exploration with a topology-aware notion of behavioral compatibility through a single constrained objective in fully decentralized MARL.

The key idea of this paper is to treat the exploration–coordination trade-off as a constrained optimization problem rather than as a hand-tuned mixture of heuristics. Concretely, we aim to maximize task return augmented with

---
[*]Equal contribution [1]School of Mechanical Engineering, University of Dalian Jiaotong, Dalian, China [2]Blockchain and Intelligent Information Systems Laboratory [3]School of Rail Transit and Intelligent Engineering, University of Dalian Jiaotong, Dalian, China. Correspondence to: Fengqi Li <Fengqi-li@outlook.com>.

*Proceedings of the 43rd International Conference on Machine Learning*, Seoul, South Korea. PMLR 306, 2026. Copyright 2026 by the author(s).

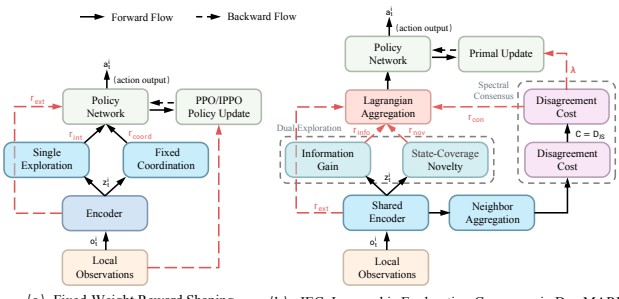

*Figure 1.* Overview: (a) Fixed-weight open-loop mixing of exploration and coordination. (b) IEC closes loop by enforcing spectral consensus and adapting via primal–dual violation feedback.

exploration incentives while constraining graph-induced policy disagreement to remain within an allowable budget: $\max \; \mathbb{E}\big[r_{\text{ext}} + r_{\text{info}} + r_{\text{nov}}\big]$, s.t. $C(\pi; G) \leq \delta$, where $r_{\text{info}}$ denotes a dynamics-based information-gain proxy targeting epistemic uncertainty, $r_{\text{nov}}$ denotes a state-coverage novelty signal, and $C(\pi; G)$ measures local policy disagreement over the communication graph $G$ via neighbor-wise divergences between action distributions. When constructed from local neighbor divergences, $C(\pi; G)$ serves as a graph-smoothness surrogate and admits an interpretation as a Dirichlet-energy-style regularizer that suppresses neighbor-inconsistent (high-frequency) policy components.

We propose IEC (Isomorphic Exploration–Consensus), which optimizes this constrained objective via lightweight primal-dual reward regularization. IEC applies standard decentralized policy optimization to an augmented reward and updates a nonnegative multiplier $\lambda$ based on observed constraint violations: $r_{\text{IEC}} = r_{\text{ext}} + r_{\text{info}} + r_{\text{nov}} - \lambda \cdot r_{\text{con}}$, $\lambda \leftarrow [\lambda + \eta(C - \delta)]_{+}$, where $[\cdot]_{+}$ denotes projection onto the nonnegative reals. This mechanism provides an explicit coupling between exploration and coordination: when disagreement exceeds the budget, $\lambda$ increases and strengthens consensus pressure; when agents remain aligned, $\lambda$ relaxes and preserves exploration capacity. Importantly, IEC is implemented as a minimal reward-level modification and naturally reduces to fixed-weight reward shaping when the dual update is disabled. Our contributions are as follows:

1. We cast the exploration-coordination tension in Dec-MARL as a constrained spectral optimization problem that couples epistemic discovery with topology-induced behavioral consensus.

2. We introduce IEC, a lightweight primal-dual reward regularization framework that adapts a consensus multiplier based on constraint violations to automatically balance exploration and convention formation.

3. We provide extensive empirical validation on benchmarks including **GridWorld** (Jiang et al., 2024a) explo-

ration, **Overcooked** (Carroll et al., 2019) coordination, and **SMAC** (Samvelyan et al., 2019) combat, together with systematic ablation studies.

## 2. Related Work

### 2.1. MARL under communication constraints

Most cooperative MARL pipelines adopt centralized training with decentralized execution (CTDE) to address non-stationarity and credit assignment, including value factorization methods (e.g., VDN (Sunehag et al., 2017), QMIX (Rashid et al., 2020)) and actor-critic approaches with centralized critics (e.g., COMA (Foerster et al., 2018), MAD-DPG (Lowe et al., 2017), MAPPO (Yu et al., 2022)). While empirically strong, these approaches rely on privileged global information or centralized components that are not applicable to fully decentralized regimes, where communication constraints also hold during training. This limitation motivates fully decentralized baselines such as independent learners (e.g., IPPO-style training (De Witt et al., 2020)) and topology-aware coordination mechanisms based on learned communication or graph modules (Sukhbaatar et al., 2016). However, coordination pressure in these methods is typically enforced through architectural inductive biases or auxiliary regularization with fixed-weight or schedules (Jiang & Lu, 2018), which can be brittle across tasks, communication topologies, and training phases.

### 2.2. Multi-agent exploration

Exploration under sparse rewards has been widely studied through intrinsic motivation, including count-based or pseudo-count-based novelty (Bellemare et al., 2016), curiosity from prediction errors (e.g., ICM (Pathak et al., 2017)), distillation-based novelty (e.g., RND (Burda et al., 2018)), and information-gain proxies (e.g., VIME (Houthooft et al., 2016)), with disagreement-based ensembles providing a more epistemic signal. In multi-agent settings, naively applying intrinsic rewards independently to each agent can amplify redundancy and induce locally novel yet globally incompatible behaviors under partial observability and limited communication. MARL-specific exploration methods, such as latent-variable or mutual-information-driven approaches (e.g., MAVEN (Mahajan et al., 2019)), promote coordinated diversity but typically rely on CTDE or couple exploration and coordination through fixed scalarization, leaving open the question of how to balance information-driven discovery with topology-induced behavioral compatibility in a fully decentralized regime (Jiang et al., 2024a).

### 2.3. Limitations

Exploration without coordination awareness. Intrinsic-motivation methods, from novelty- and curiosity-based sig-

nals to disagreement-based ensembles(Pathak et al., 2019), improve exploration but are largely agnostic to multi-agent coordination. When used independently, they can amplify redundant exploration and produce locally novel yet globally incompatible behaviors. Recent MARL exploration methods partly address this through shared exploration objectives(Liu et al., 2021), episodic curiosity(Zheng et al., 2021), agent-specific scaffolds(Li et al., 2024), activity incentives(Liu et al., 2023), value-disagreement exploration(Learning), or replay-derived subgoals(Jeon et al., 2022). However, they still do not explicitly regulate whether exploration outcomes remain compatible over the communication graph, and typically treat exploration and coordination as separate objectives with fixed relative weighting.

Coordination without adaptive exploration balancing. On the coordination side, influence- or MI-based methods (Influence-Based Exploration(Wang et al., 2019), Social Influence(Jaques et al., 2019), PMIC(Li et al., 2022)) encourage coordinated behavior through interaction-aware auxiliary signals, while structural approaches such as RODE(Wang et al., 2020) and trust-region MARL methods such as HAPPO(Kuba et al., 2021) improve coordination through decomposition or optimization design. Adaptive entropy regularization(Kim & Sung, 2023)and MAVEN(Mahajan et al., 2019) also improve exploration/coordination behavior, but they do not provide a principled mechanism to adapt the exploration–coordination trade-off itself as training evolves.

Automatic reward calibration in simpler settings. AIRS(Yuan et al., 2023) shares IEC's philosophy of reducing hand-tuned shaping through adaptive reward calibration, but it operates in the single-agent setting where coordination is absent. Extending this idea to decentralized MARL requires jointly adapting exploration and coordination, which is precisely the constrained optimization problem addressed by IEC.

# 3. Preliminary

## 3.1. Notation

Scalars, vectors, matrices, and sets are denoted by $a$, $\mathbf{a}$, $\mathbf{A}$, and $\mathcal{A}$, respectively, and $\triangleq$ denotes a definition. Let $G = (\mathcal{V}, \mathcal{E})$ be an undirected communication graph with $|\mathcal{V}| = N$ agents and neighbor set $\mathcal{N}(i) \triangleq \{j \mid (i, j) \in \mathcal{E}\}$.

To quantify local policy disagreement, we use a symmetric, bounded divergence. Unless stated otherwise, we instantiate it with the Jensen–Shannon divergence computed with log base 2: We have $D_{\mathrm{JS}}(p\|q) \in [0, 1]$. For compactness, we write $D_{\mathrm{JS}}$ throughout; switching to a KL-based variant only requires consistently replacing $D_{\mathrm{JS}}$.

To model link failures, each edge $(i, j) \in \mathcal{E}$ is active at

time $t$ with indicator $\xi_t^{ij} \sim \mathrm{Bernoulli}(1 - p_{\mathrm{drop}})$. The active graph is $G_t = (\mathcal{V}, \mathcal{E}_t)$, where $\mathcal{E}_t \triangleq \{(i, j) \in \mathcal{E} \mid \xi_t^{ij} = 1\}$, and $\mathcal{N}_t(i)$ denotes the active neighbors. Agents exchange lightweight messages with one-hop neighbors only.

## 3.2. Decentralized Cooperative Markov Game

We consider a cooperative partially observable Markov game (Dec-POMDP/Dec-MG) (Oliehoek et al., 2016). At time $t$, the environment state is $s_t \in \mathcal{S}$, agent $i$ receives a local observation $o_t^i \in \mathcal{O}_i$, and selects an action $a_t^i \in \mathcal{A}_i$ according to a decentralized policy $\pi_i(\cdot \mid h_t^i)$. The local information state $h_t^i$ summarizes the observation, action, and message history (e.g., an RNN hidden state). The joint action $a_t = (a_t^1, \ldots, a_t^N)$ induces a transition $s_{t+1} \sim \mathcal{P}(\cdot \mid s_t, a_t)$. All agents share a team reward $r_{\mathrm{ext}}(s_t, a_t)$ and aim to maximize the discounted return.

$$J_{\mathrm{ext}}(\pi) \triangleq \mathbb{E}_\pi \left[ \sum_{t=0}^\infty \gamma^t r_{\mathrm{ext}}(s_t, a_t) \right], \qquad \gamma \in (0, 1). \quad (1)$$

## 3.3. Problem Formulation

We couple exploration and coordination through a single constrained objective. Specifically, we maximize extrinsic return augmented with two complementary exploration signals while constraining graph-induced policy disagreement.

At each step, agent $i$ receives neighbors' action distributions (or logits) and forms a local reference distribution,

$$\bar{\pi}_{i,t}(\cdot) \triangleq \sum_{j \in \mathcal{N}_t(i)} w_t^{ij} \pi_j(\cdot \mid h_t^j),$$
$$\sum_{j \in \mathcal{N}_t(i)} w_t^{ij} = 1, \quad w_t^{ij} \geq 0. \quad (2)$$

We define the per-step disagreement cost as:

$$c_t^i \triangleq D_{\mathrm{JS}}\Big(\pi_i(\cdot \mid h_t^i), \, \bar{\pi}_{i,t}(\cdot)\Big),$$
$$C(\pi; G) \triangleq \mathbb{E}_\pi \left[ \sum_{t=0}^\infty \gamma^t \frac{1}{N} \sum_{i=1}^N c_t^i \right]. \quad (3)$$

When constructed from neighbor-wise divergences, $C(\pi; G)$ serves as a graph-smoothness surrogate and can be interpreted as a Dirichlet-energy-style regularizer that suppresses neighbor-inconsistent (high-frequency) components of the policy signal over the communication graph. The Disagreement budget may be constant or decay over training:

$$\delta_k = \begin{cases} \delta & \text{(constant)} \\ \delta_{\min} + (\delta_0 - \delta_{\min}) \exp(-k/\tau) & \text{(decay)} \end{cases}, \quad (4)$$

where $k$ denotes the optimization iteration index. We ablate constant versus decaying budgets in experiments.

IEC constrained objective: the constrained problem is:

$$\max_{\pi} \mathbb{E}_{\pi}\left[\sum_{t=0}^{\infty} \gamma^t \left( r_{\text{ext}}(s_t, a_t) + \alpha_{\text{info}} \frac{1}{N} \sum_i r_{\text{info},t}^i \right. \right.$$
$$\left. \left. + \alpha_{\text{nov}} \frac{1}{N} \sum_i r_{\text{nov},t}^i \right) \right] \quad \text{s.t. } C(\pi; G) \leq \delta_k. \tag{5}$$

In the conceptual formulation, $\alpha_{\text{info}}$ and $\alpha_{\text{nov}}$ can be set to 1; we retain them explicitly to support stable scaling and controlled ablations.

# 4. Methodology

IEC balances exploration–coordination via primal–dual regularization (Fig. 2), adapting $\lambda$ from constraint violations to shift from exploration to consensus.

## 4.1. Reward Regularization

We solve (5) via Lagrangian relaxation with a nonnegative multiplier $\lambda \geq 0$:

$$\mathcal{L}(\pi, \lambda) \triangleq J_{\text{ext}}(\pi) + \alpha_{\text{info}} J_{\text{info}}(\pi) + \alpha_{\text{nov}} J_{\text{nov}}(\pi)$$
$$- \lambda \big( C(\pi; G) - \delta_k \big). \tag{6}$$

IEC instantiates relaxation through reward-level shaping:

$$r_{\text{IEC},t}^i \triangleq r_{\text{ext}}(s_t, a_t) + \alpha_{\text{info}} r_{\text{info},t}^i + \alpha_{\text{nov}} r_{\text{nov},t}^i$$
$$- \lambda r_{\text{con},t}^i, \quad r_{\text{con},t}^i \triangleq c_t^i. \tag{7}$$

The primal step updates decentralized policies using a standard policy-gradient learner (we employ PPO/IPPO-style updates (Schulman et al., 2017; De Witt et al., 2020)). The dual step updates $\lambda$ via projected ascent on the constraint violation:[1]

$$\lambda \leftarrow \Pi_{[0,\lambda_{\max}]}\left(\lambda + \eta_\lambda(\widehat{C} - \delta_k)\right), \tag{8}$$

where $\widehat{C}$ denotes the empirical estimate of (3) computed on the collected batch, and $\Pi_{[0,\lambda_{\max}]}$ denotes projection, with $\lambda_{\max}$ optional but recommended for numerical stability. When $\eta_\lambda = 0$, $\lambda$ is fixed and IEC reduces to fixed-weight reward shaping, enabling a clean ablation.

The multiplier $\lambda$ can be interpreted as the shadow price of violating the consensus budget: when disagreement exceeds $\delta_k$, the effective penalty increases, automatically rebalancing exploration incentives against topology-induced compatibility requirements.

---

[1] For computational efficiency, we aggregate $\hat{C}$ centrally during training, analogous to policy parameter synchronization in IPPO. Execution remains fully decentralized.

## 4.2. Instantiating Exploration and Consensus

To operationalize the IEC framework, we instantiate the primal exploration objectives and the dual consensus constraint with distinct, calculable proxies.

**Information Gain $r_{info}$**: to encourage epistemic discovery, we use a dynamics-based proxy that focuses on reducible model uncertainty. We maintain an ensemble of $K$ forward dynamics predictors operating in a latent space. Let $z_t^i = \phi(o_t^i)$ denote the encoder output and $\hat{z}_{t+1}^{i,(k)} = f_\varphi^{i,(k)}(z_t^i, a_t^i)$ the $k$-th ensemble prediction. Epistemic uncertainty is quantified via ensemble disagreement:

$$u_t^i \triangleq \frac{1}{K} \sum_{k=1}^K \left\| \hat{z}_{t+1}^{i,(k)} - \frac{1}{K} \sum_{\ell=1}^K \hat{z}_{t+1}^{i,(\ell)} \right\|_2^2, \quad r_{\text{info},t}^i \triangleq u_t^i. \tag{9}$$

This choice is robust to "noisy-TV" distractions, as pure stochasticity does not systematically reduce ensemble disagreement during learning.

When using a single predictor $f_\varphi$ instead of an ensemble, a simpler proxy is:

$$r_{\text{info},t}^i \triangleq \left\| f_\varphi^i(z_t^i, a_t^i) - z_{t+1}^i \right\|_2^2, \tag{10}$$

which approximates epistemic uncertainty under near-deterministic dynamics. We include this variant as a baseline for completeness and code alignment.

Under resource constraints, optimal information acquisition prioritizes uncertainty that most affects return (Theorem 5.5). Motivated by this principle, we optionally define:

$$r_{\text{info},t}^i \triangleq \text{stopgrad}(|\widehat{A}_t^i|) \cdot u_t^i, \tag{11}$$

where $\widehat{A}_t^i$ denotes an advantage estimate and $\text{stopgrad}$ prevents backpropagation through the critic.

**State-Coverage Novelty $r_{nov}$**: to promote broad state coverage, each agent maintains a visitation statistic over discretized latent features. Let $b_t^i = \text{hash}(z_t^i)$ denote the discretized latent code and $n^i(b)$ the corresponding count table.

$$r_{\text{nov},t}^i \triangleq \frac{1}{\sqrt{n^i(b_t^i) + 1}}, \quad n^i(b_t^i) \leftarrow n^i(b_t^i) + 1. \tag{12}$$

This signal complements $r_{\text{info}}$ by rewarding coverage even when local model uncertainty is low.

**Spectral Consensus Penalty $r_{con}$**: to enforce topology-aware behavioral compatibility (the constraint), IEC uses a local disagreement cost. At each step, agent $i$ computes $\bar{\pi}_{i,t}$ from neighbor messages and sets:

$$r_{\text{con},t}^i \triangleq c_t^i = D_{\text{JS}}\Big( \pi_i(\cdot \mid h_t^i), \bar{\pi}_{i,t}(\cdot) \Big). \tag{13}$$

This penalty is locally computable under decentralized communication; if $\mathcal{N}_t(i) = \emptyset$, we set $r_{\text{con},t}^i = 0$.

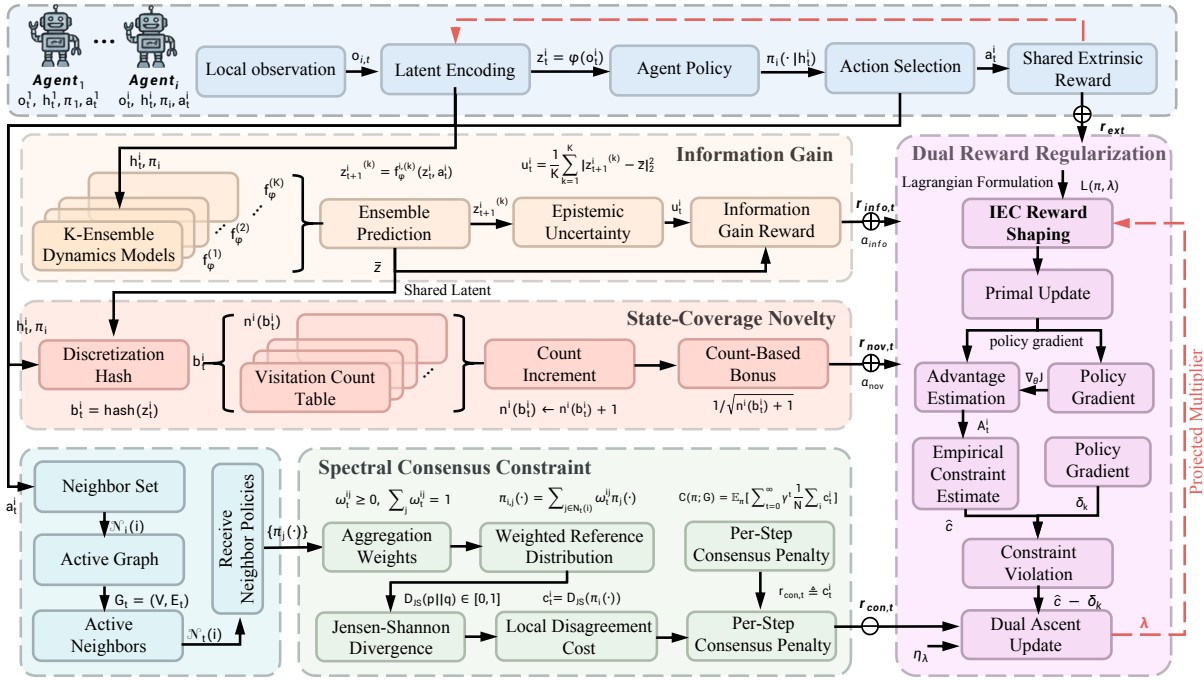

*Figure 2.* **The IEC Framework Architecture** couples information-driven exploration with spectral consensus via constrained optimization. Agents maximize returns augmented with dual intrinsic signals, epistemic uncertainty and state novelty, while enforcing topology-aware consistency through a Spectral Consensus Constraint that interprets neighbor divergence as Dirichlet energy. To ensure robust convergence without manual tuning, Primal-Dual Reward Regularization adapts the Lagrangian multiplier $\lambda$ via projected ascent, acting as a dynamic shadow price to automatically gate coordination pressure based on budget $\delta$ violations.

### 4.3. Implementation Details and Complexity

We implement IEC on top of decentralized PPO / IPPO (Schulman et al., 2017) which is learner-agnostic since it only reshapes rewards. Per step, each active edge exchanges an action distribution (or logits) of size $|\mathcal{A}|$. Thus communication is $O(|\mathcal{E}_t||\mathcal{A}|)$ per step. Computing $\bar{\pi}_{i,t}$ and one $D_{JS}$ per agent yields $O(N|\mathcal{A}|)$ divergence computation per step. Under dropout, the expected messaging cost scales with $\mathbb{E}[|\mathcal{E}_t|] = (1 - p_{drop})|\mathcal{E}|$. We report task-specific final values in the experimental section, and provide default ranges. See appendix for all details.

## 5. Theoretical Analysis

Exact planning for finite-horizon DEC-POMDPs is NEXP-complete (Ellis et al., 2023), motivating the use of scalable gradient-based learning methods and, consequently, a focus on principled objective design and stable optimization.

**Disagreement control implies behavioral compatibility.** We formalize why bounding $C(\pi; G)$ enforces neighbor-level behavioral compatibility.

**Lemma 5.1** (Variation is controlled by divergence). *For common choices of $D_{JS}$ (including KL via Pinsker's inequality and JS via standard bounds), there exists a con-*

*stant $\kappa > 0$ such that $d_{TV}(p, q)^2 \leq \kappa\, D_{JS}(p, q)$ for all distributions $p, q$ over a finite alphabet.*

*Proof.* Pinsker-type inequalities imply $d_{TV}(p, q)^2 \leq \kappa\, D_{KL}(p, q)$ for both KL and JS divergences. A detailed proof can be found in the appendix. $\square$

**Proposition 5.2** (Bounding average neighbor mismatch). *If $C(\pi; G) \leq \delta_k$, then the expected discounted average neighbor mismatch is bounded as:*

$$\mathbb{E}_\pi\left[\sum_{t\geq 0} \gamma^t \frac{1}{N} \sum_{i=1}^N d_{TV}\Big(\pi_i(\cdot \mid h_t^i), \bar{\pi}_{i,t}\Big)^2\right] \leq \kappa\, \delta_k, \quad (14)$$

*where $\kappa$ is the constant from Lemma 5.1.*

*Proof.* Apply Lemma 5.1 pointwise to $c_t^i = D_{JS}(\pi_i, \bar{\pi}_{i,t})$ and average over agents/time. See appendix for details. $\square$

**Dirichlet-energy-style interpretation.** When $D_{JS}$ admits a local quadratic approximation in policy space, the neighbor-sum reduces to a Laplacian energy $\sum_{(i,j)} w_{ij}\|u_i - u_j\|_2^2 = \mathbf{u}^\top \mathbf{L}\mathbf{u}$, justifying $C(\pi; G)$ as a Dirichlet-energy-style smoothness surrogate. This interpretation is local (Appendix B.3) and holds when neighbor policies are close—typically later in training, whereas

*Table 1.* Evaluation of Exploration-Coordination Robustness across Varying Sampling Budgets and Task Hierarchies in GridWorld

| | Parameters | | | IPPO | | IPPO+r_loc | | IPPO+r_hin | | IPPO+r_nov | | MACE | | IEC (Ours) | |
|---|---|---|---|---|---|---|---|---|---|---|---|---|---|---|---|
| | $N$ | $B$ | Steps | WR% | AUC | WR% | AUC | WR% | AUC | WR% | AUC | WR% | AUC | WR% | AUC |
| **PassRoom** | 128 | 600 | $4e^7$ | $28.8_{\pm15.2}$ | $16.2_{\pm8.8}$ | $35.2_{\pm14.2}$ | $20.5_{\pm7.8}$ | $52.2_{\pm8.1}$ | $48.2_{\pm8.2}$ | $90.8_{\pm3.2}$ | $68.3_{\pm7.0}$ | $\mathbf{97.5}_{\pm1.8}$ | $90.5_{\pm2.8}$ | $95.8_{\pm2.5}$ | $87.5_{\pm3.2}$ |
| | 128 | 300 | $5e^7$ | $36.5_{\pm17.5}$ | $21.2_{\pm10.5}$ | $42.5_{\pm16.2}$ | $25.8_{\pm9.2}$ | $58.5_{\pm7.8}$ | $55.8_{\pm7.3}$ | $94.2_{\pm2.5}$ | $75.9_{\pm5.2}$ | $\mathbf{98.8}_{\pm1.2}$ | $93.2_{\pm2.1}$ | $97.2_{\pm1.8}$ | $90.5_{\pm2.5}$ |
| | 32 | 300 | $4e^7$ | $26.2_{\pm14.8}$ | $14.5_{\pm8.2}$ | $32.5_{\pm13.8}$ | $18.2_{\pm7.2}$ | $50.5_{\pm8.5}$ | $42.5_{\pm9.1}$ | $88.2_{\pm4.2}$ | $62.0_{\pm7.8}$ | $\mathbf{97.2}_{\pm2.2}$ | $88.5_{\pm3.3}$ | $94.8_{\pm2.8}$ | $85.3_{\pm3.5}$ |
| | 32 | 600 | $5e^7$ | $32.8_{\pm16.2}$ | $19.2_{\pm9.5}$ | $38.5_{\pm14.8}$ | $23.5_{\pm8.5}$ | $54.2_{\pm7.8}$ | $49.2_{\pm8.4}$ | $91.5_{\pm3.5}$ | $69.8_{\pm6.5}$ | $\mathbf{98.5}_{\pm1.5}$ | $91.6_{\pm2.6}$ | $96.2_{\pm2.2}$ | $88.4_{\pm3.0}$ |
| | 8 | 300 | $5e^7$ | $22.5_{\pm13.5}$ | $12.2_{\pm7.5}$ | $28.5_{\pm12.5}$ | $15.8_{\pm6.5}$ | $48.5_{\pm9.2}$ | $38.2_{\pm9.8}$ | $85.8_{\pm5.2}$ | $58.5_{\pm8.8}$ | $\mathbf{96.8}_{\pm2.5}$ | $87.3_{\pm3.5}$ | $93.5_{\pm3.2}$ | $82.8_{\pm4.2}$ |
| | 8 | 600 | $5e^7$ | $28.5_{\pm14.8}$ | $16.5_{\pm8.8}$ | $34.2_{\pm13.5}$ | $20.2_{\pm7.8}$ | $52.8_{\pm8.5}$ | $45.6_{\pm9.2}$ | $89.2_{\pm4.5}$ | $65.2_{\pm7.5}$ | $\mathbf{97.8}_{\pm1.8}$ | $89.5_{\pm3.0}$ | $95.2_{\pm2.6}$ | $86.2_{\pm3.6}$ |
| **SecretRoom** | 128 | 600 | $8e^7$ | $13.2_{\pm17.5}$ | $6.8_{\pm9.5}$ | $18.5_{\pm16.8}$ | $10.2_{\pm8.8}$ | $24.8_{\pm7.5}$ | $24.2_{\pm9.5}$ | $58.2_{\pm7.5}$ | $56.8_{\pm6.8}$ | $95.5_{\pm2.4}$ | $85.2_{\pm5.2}$ | $\mathbf{97.2}_{\pm2.0}$ | $90.5_{\pm2.8}$ |
| | 128 | 300 | $1e^8$ | $20.5_{\pm21.2}$ | $11.2_{\pm12.5}$ | $26.8_{\pm20.2}$ | $15.5_{\pm11.5}$ | $32.5_{\pm9.0}$ | $32.8_{\pm11.2}$ | $65.8_{\pm6.2}$ | $64.5_{\pm5.2}$ | $\mathbf{98.8}_{\pm1.2}$ | $94.2_{\pm1.8}$ | $98.0_{\pm1.8}$ | $90.6_{\pm3.8}$ |
| | 32 | 300 | $8e^7$ | $10.5_{\pm15.5}$ | $5.2_{\pm8.2}$ | $15.2_{\pm14.5}$ | $8.5_{\pm7.5}$ | $22.5_{\pm7.2}$ | $20.5_{\pm8.8}$ | $55.2_{\pm8.2}$ | $51.5_{\pm7.8}$ | $94.8_{\pm2.8}$ | $82.0_{\pm5.8}$ | $\mathbf{96.5}_{\pm2.5}$ | $87.5_{\pm3.5}$ |
| | 32 | 600 | $8e^7$ | $15.2_{\pm18.5}$ | $8.5_{\pm10.2}$ | $20.5_{\pm17.2}$ | $12.2_{\pm9.5}$ | $26.8_{\pm8.5}$ | $26.8_{\pm10.2}$ | $60.2_{\pm7.0}$ | $58.5_{\pm6.5}$ | $\mathbf{97.8}_{\pm1.8}$ | $91.2_{\pm2.6}$ | $96.2_{\pm2.2}$ | $86.3_{\pm4.8}$ |
| | 8 | 300 | $8e^7$ | $7.5_{\pm13.2}$ | $3.8_{\pm6.8}$ | $12.2_{\pm12.5}$ | $6.8_{\pm6.2}$ | $18.5_{\pm6.5}$ | $16.2_{\pm7.8}$ | $48.5_{\pm9.2}$ | $45.2_{\pm8.8}$ | $92.5_{\pm3.5}$ | $78.5_{\pm6.5}$ | $\mathbf{95.2}_{\pm3.0}$ | $84.2_{\pm4.2}$ |
| | 8 | 600 | $8e^7$ | $12.8_{\pm16.5}$ | $7.2_{\pm9.2}$ | $17.8_{\pm15.5}$ | $10.5_{\pm8.5}$ | $24.2_{\pm7.8}$ | $22.5_{\pm9.2}$ | $56.2_{\pm7.8}$ | $52.8_{\pm7.2}$ | $94.2_{\pm2.8}$ | $82.8_{\pm5.2}$ | $\mathbf{96.8}_{\pm2.2}$ | $88.5_{\pm3.2}$ |
| **MultiRoom** | 128 | 300 | $8e^7$ | $4.8_{\pm7.2}$ | $2.5_{\pm4.2}$ | $8.5_{\pm9.5}$ | $4.8_{\pm5.2}$ | $22.8_{\pm8.5}$ | $22.5_{\pm9.8}$ | $38.5_{\pm9.2}$ | $38.5_{\pm8.5}$ | $85.2_{\pm4.8}$ | $70.5_{\pm5.8}$ | $\mathbf{95.2}_{\pm2.5}$ | $87.8_{\pm3.5}$ |
| | 128 | 600 | $1e^8$ | $7.2_{\pm9.5}$ | $4.2_{\pm5.5}$ | $11.5_{\pm11.2}$ | $6.8_{\pm6.5}$ | $28.8_{\pm10.2}$ | $28.8_{\pm11.2}$ | $45.5_{\pm8.2}$ | $45.2_{\pm7.2}$ | $90.5_{\pm3.8}$ | $78.2_{\pm4.5}$ | $\mathbf{97.5}_{\pm1.8}$ | $92.2_{\pm2.5}$ |
| | 32 | 300 | $8e^7$ | $2.5_{\pm4.8}$ | $1.2_{\pm2.5}$ | $5.2_{\pm6.8}$ | $2.8_{\pm3.5}$ | $19.2_{\pm7.8}$ | $19.2_{\pm8.8}$ | $35.2_{\pm9.5}$ | $35.8_{\pm9.0}$ | $82.8_{\pm5.5}$ | $66.5_{\pm6.5}$ | $\mathbf{93.8}_{\pm2.8}$ | $84.5_{\pm4.0}$ |
| | 32 | 600 | $8e^7$ | $4.5_{\pm6.8}$ | $2.2_{\pm3.8}$ | $8.2_{\pm9.2}$ | $4.5_{\pm4.8}$ | $24.5_{\pm9.0}$ | $24.8_{\pm10.0}$ | $40.5_{\pm8.8}$ | $40.2_{\pm8.2}$ | $86.5_{\pm4.8}$ | $72.2_{\pm5.5}$ | $\mathbf{95.5}_{\pm2.2}$ | $88.2_{\pm3.2}$ |
| | 8 | 300 | $8e^7$ | $1.5_{\pm3.5}$ | $0.8_{\pm1.8}$ | $3.5_{\pm5.2}$ | $1.8_{\pm2.5}$ | $15.8_{\pm6.5}$ | $15.5_{\pm7.5}$ | $28.5_{\pm10.2}$ | $28.2_{\pm10.0}$ | $78.5_{\pm6.2}$ | $60.5_{\pm7.5}$ | $\mathbf{91.2}_{\pm3.5}$ | $80.5_{\pm4.8}$ |
| | 8 | 600 | $8e^7$ | $3.2_{\pm5.5}$ | $1.5_{\pm2.8}$ | $6.5_{\pm7.5}$ | $3.2_{\pm3.8}$ | $20.5_{\pm8.2}$ | $20.2_{\pm9.0}$ | $35.8_{\pm9.5}$ | $35.2_{\pm9.2}$ | $82.2_{\pm5.2}$ | $65.8_{\pm6.8}$ | $\mathbf{93.5}_{\pm2.8}$ | $84.8_{\pm3.8}$ |

**Note:** $N$: the number of parallel environments, $B$: the rollout buffer length, and Steps: the total number of environmental time steps.

the divergence constraint itself remains well-defined and effective throughout.

**Theorem 5.3** (KKT-like conditions at a stable fixed point). *If $(\pi^\star, \lambda^\star)$ is a stable fixed point of the coupled primal policy update and the projected dual update* (8)*, then: (i) $\lambda^\star \geq 0$, (ii) $C(\pi^\star; G) \leq \delta_k$, (iii) $\lambda^\star(C(\pi^\star; G) - \delta_k) = 0$, and (iv) $\pi^\star$ is a stationary point of $\mathcal{L}(\pi, \lambda^\star)$.*

*Proof.* The projection step enforces complementarity: either $\lambda^\star = 0$ or the constraint violation vanishes. Primal stationarity follows from convergence of the base learner under the shaped reward. □

**Proposition 5.4** (Fixed-weight scalarization can be brittle). *There exist decentralized cooperative tasks with phase-dependent requirements such that for any fixed penalty weight $\lambda$, training exhibits either* fragmentation *(weak coordination) or* premature collapse *(weak exploration). In contrast, a violation-driven multiplier update can satisfy both phases without hand-crafted schedules.*

*Proof.* In a two-phase task, Phase I needs small $\lambda$ to reach informative states, while Phase II needs large $\lambda$ to avoid deadlocks. A violation-driven update raises $\lambda$ only when disagreement persists, matching both phases. □

**Resource-constrained optimal perception is value-gradient weighted.** We derive a principle that connects optimal information acquisition to IEC's information-driven exploration. Let $x_t \in \mathbb{R}^d$ denote features extracted from

observation $o_t$. A perception gate $g \in [0,1]^d$ allocates sensing or computation resources by selecting $\tilde{x}_t(g) \triangleq g \odot x_t$, subject to a budget constraint $|g|_1 \leq B$. Let $J(g)$ denote the expected return induced by the policy $\pi(\cdot \mid \tilde{x}(g))$.

**Theorem 5.5** (Gradient-weighted optimal allocation). *Under a first-order approximation $J(g) \approx J(g_0) + \nabla_g J(g_0)^\top (g - g_0)$ with constraints $0 \leq g \leq 1$ and $|g|_1 \leq B$, an optimal budget allocation assigns resources to coordinates with the largest positive components of $\nabla_g J(g_0)$. Equivalently, optimal perception is weighted by the value sensitivity of return with respect to sensing resources.*

*Proof.* The linearized problem is a constrained linear program. KKT conditions imply that at optimum, mass is allocated to coordinates with the largest marginal gain per unit cost, given by $\nabla_g J(g_0)$, until the budget is exhausted. □

**Operationalization in IEC.** Theorem 5.5 motivates value-weighted information gain (Eq. (11)), which focuses exploration on uncertainties that most affect return. Separately, IEC's dual variable $\lambda$ acts as the shadow price of the consensus budget, scaling the effective contribution of the consensus term relative to exploration incentives as constraint pressure varies.

**Proposition 5.6** (Boundedness Under Noisy Communication, bounded shaped rewards with JSD and projected $\lambda$). *If $D_{\text{JS}}(\cdot|\cdot) \in [0,1]$ (log base 2) and $\lambda \in [0, \lambda_{\max}]$, then for bounded intrinsic rewards $r_{info}, r_{nov} \in [0, R_{\max}]$, the*

*Table 2.* Quantitative Evaluation of Coordination Robustness across Spatially Distinct Overcooked Layouts and Varying Sampling Budgets

| | $N$ | $B$ | Steps | IPPO WR% | IPPO AUC | IPPO+r_loc WR% | IPPO+r_loc AUC | IPPO+r_hin WR% | IPPO+r_hin AUC | IPPO+r_nov WR% | IPPO+r_nov AUC | MACE WR% | MACE AUC | CIRC WR% | CIRC AUC |
|---|---|---|---|---|---|---|---|---|---|---|---|---|---|---|---|
| **Base** | 128 | 600 | $1e^7$ | $3.7_{\pm4.8}$ | $2.9_{\pm5.0}$ | $6.2_{\pm2.8}$ | $5.2_{\pm2.8}$ | $58.8_{\pm7.0}$ | $52.8_{\pm6.9}$ | $38.5_{\pm7.2}$ | $32.5_{\pm7.2}$ | $89.5_{\pm3.5}$ | $80.2_{\pm4.1}$ | $\mathbf{94.2}_{\pm2.5}$ | $84.8_{\pm3.0}$ |
| | 128 | 300 | $5e^6$ | $2.9_{\pm4.2}$ | $1.9_{\pm4.0}$ | $4.8_{\pm2.5}$ | $3.5_{\pm2.2}$ | $55.5_{\pm7.5}$ | $48.5_{\pm7.2}$ | $35.2_{\pm7.8}$ | $28.8_{\pm7.5}$ | $87.2_{\pm3.8}$ | $78.5_{\pm4.5}$ | $\mathbf{93.8}_{\pm2.8}$ | $83.5_{\pm3.2}$ |
| | 32 | 300 | $1e^7$ | $6.1_{\pm6.0}$ | $4.7_{\pm5.8}$ | $10.2_{\pm3.5}$ | $8.5_{\pm3.2}$ | $65.2_{\pm6.2}$ | $60.5_{\pm5.8}$ | $45.5_{\pm6.5}$ | $39.5_{\pm6.2}$ | $92.5_{\pm2.5}$ | $86.2_{\pm3.8}$ | $\mathbf{96.2}_{\pm2.0}$ | $88.2_{\pm2.2}$ |
| | 32 | 600 | $5e^6$ | $4.5_{\pm5.1}$ | $3.4_{\pm5.0}$ | $7.5_{\pm3.0}$ | $6.2_{\pm2.8}$ | $60.5_{\pm6.8}$ | $55.2_{\pm6.5}$ | $40.2_{\pm7.0}$ | $34.2_{\pm6.8}$ | $90.2_{\pm3.0}$ | $82.5_{\pm4.2}$ | $\mathbf{94.8}_{\pm2.3}$ | $85.8_{\pm2.8}$ |
| | 8 | 300 | $1e^7$ | $2.1_{\pm3.7}$ | $1.4_{\pm3.6}$ | $3.5_{\pm2.2}$ | $2.5_{\pm2.0}$ | $52.8_{\pm8.0}$ | $45.5_{\pm7.8}$ | $32.5_{\pm8.2}$ | $26.5_{\pm8.0}$ | $86.5_{\pm4.2}$ | $76.5_{\pm5.0}$ | $\mathbf{92.5}_{\pm3.0}$ | $81.5_{\pm3.8}$ |
| | 8 | 600 | $1e^7$ | $3.1_{\pm4.4}$ | $2.2_{\pm4.1}$ | $5.2_{\pm2.6}$ | $4.0_{\pm2.3}$ | $56.2_{\pm7.5}$ | $49.2_{\pm7.2}$ | $36.5_{\pm7.5}$ | $30.2_{\pm7.2}$ | $88.6_{\pm3.6}$ | $79.8_{\pm4.8}$ | $\mathbf{93.8}_{\pm2.6}$ | $83.8_{\pm3.2}$ |
| **Narrow** | 128 | 600 | $2e^7$ | $0.6_{\pm2.4}$ | $0.4_{\pm2.5}$ | $1.5_{\pm1.2}$ | $1.2_{\pm1.2}$ | $45.2_{\pm7.5}$ | $43.8_{\pm7.2}$ | $30.5_{\pm8.0}$ | $25.8_{\pm7.8}$ | $76.2_{\pm5.0}$ | $74.2_{\pm4.5}$ | $\mathbf{80.8}_{\pm3.8}$ | $73.5_{\pm4.2}$ |
| | 128 | 300 | $1e^7$ | $1.2_{\pm3.6}$ | $0.9_{\pm3.4}$ | $3.0_{\pm1.8}$ | $2.5_{\pm1.6}$ | $51.5_{\pm6.8}$ | $50.5_{\pm6.5}$ | $35.8_{\pm7.2}$ | $31.2_{\pm7.0}$ | $\mathbf{82.2}_{\pm3.2}$ | $78.2_{\pm3.5}$ | $81.5_{\pm4.2}$ | $79.5_{\pm3.6}$ |
| | 32 | 300 | $2e^7$ | $0.8_{\pm2.8}$ | $0.6_{\pm2.9}$ | $2.0_{\pm1.4}$ | $1.8_{\pm1.4}$ | $47.2_{\pm7.5}$ | $46.2_{\pm7.0}$ | $32.0_{\pm7.8}$ | $27.5_{\pm7.5}$ | $77.8_{\pm4.8}$ | $75.5_{\pm4.2}$ | $\mathbf{82.3}_{\pm3.6}$ | $74.8_{\pm4.0}$ |
| | 32 | 600 | $1e^7$ | $1.1_{\pm3.2}$ | $0.8_{\pm3.2}$ | $2.8_{\pm1.6}$ | $2.3_{\pm1.5}$ | $50.0_{\pm7.0}$ | $49.0_{\pm6.7}$ | $34.5_{\pm7.5}$ | $29.8_{\pm7.2}$ | $80.2_{\pm4.5}$ | $78.2_{\pm3.8}$ | $\mathbf{83.8}_{\pm3.3}$ | $77.5_{\pm3.7}$ |
| | 8 | 300 | $2e^7$ | $0.4_{\pm2.0}$ | $0.3_{\pm1.9}$ | $1.0_{\pm1.0}$ | $0.8_{\pm0.9}$ | $42.5_{\pm8.0}$ | $40.8_{\pm7.8}$ | $28.2_{\pm8.5}$ | $24.2_{\pm8.2}$ | $74.5_{\pm5.5}$ | $72.5_{\pm5.0}$ | $\mathbf{78.5}_{\pm4.2}$ | $70.2_{\pm4.8}$ |
| | 8 | 600 | $2e^7$ | $0.7_{\pm2.6}$ | $0.5_{\pm2.5}$ | $1.8_{\pm1.3}$ | $1.5_{\pm1.2}$ | $45.8_{\pm7.6}$ | $43.5_{\pm7.4}$ | $31.5_{\pm8.2}$ | $26.8_{\pm7.8}$ | $\mathbf{80.2}_{\pm3.8}$ | $72.8_{\pm4.5}$ | $76.8_{\pm5.2}$ | $74.0_{\pm4.7}$ |
| **Large** | 128 | 600 | $4e^7$ | $0.4_{\pm2.6}$ | $0.2_{\pm3.0}$ | $1.2_{\pm1.2}$ | $1.0_{\pm1.3}$ | $50.8_{\pm7.0}$ | $44.5_{\pm7.5}$ | $28.8_{\pm7.8}$ | $24.2_{\pm8.2}$ | $65.8_{\pm5.0}$ | $58.5_{\pm5.8}$ | $\mathbf{86.8}_{\pm3.5}$ | $80.2_{\pm4.0}$ |
| | 128 | 300 | $2e^7$ | $0.8_{\pm3.7}$ | $0.6_{\pm4.4}$ | $2.5_{\pm1.7}$ | $2.2_{\pm1.9}$ | $57.2_{\pm6.2}$ | $51.5_{\pm6.8}$ | $35.2_{\pm6.8}$ | $29.5_{\pm7.2}$ | $70.5_{\pm4.2}$ | $64.2_{\pm4.8}$ | $\mathbf{90.2}_{\pm2.8}$ | $84.8_{\pm3.2}$ |
| | 32 | 300 | $4e^7$ | $0.4_{\pm2.9}$ | $0.3_{\pm3.4}$ | $1.5_{\pm1.3}$ | $1.3_{\pm1.5}$ | $52.5_{\pm7.2}$ | $46.8_{\pm7.4}$ | $30.8_{\pm7.5}$ | $25.8_{\pm7.8}$ | $67.2_{\pm4.8}$ | $60.8_{\pm5.5}$ | $\mathbf{87.5}_{\pm3.4}$ | $81.2_{\pm3.8}$ |
| | 32 | 600 | $2e^7$ | $0.6_{\pm3.3}$ | $0.5_{\pm3.9}$ | $2.0_{\pm1.5}$ | $1.8_{\pm1.7}$ | $55.8_{\pm6.8}$ | $49.8_{\pm7.0}$ | $33.5_{\pm7.0}$ | $28.2_{\pm7.5}$ | $69.5_{\pm4.5}$ | $63.2_{\pm5.0}$ | $\mathbf{89.2}_{\pm3.0}$ | $83.5_{\pm3.5}$ |
| | 8 | 300 | $4e^7$ | $0.2_{\pm2.2}$ | $0.2_{\pm2.5}$ | $0.8_{\pm1.0}$ | $0.7_{\pm1.1}$ | $48.5_{\pm7.8}$ | $42.2_{\pm8.1}$ | $27.2_{\pm8.2}$ | $22.5_{\pm8.5}$ | $63.8_{\pm5.5}$ | $56.2_{\pm6.1}$ | $\mathbf{85.2}_{\pm3.8}$ | $78.5_{\pm4.5}$ |
| | 8 | 600 | $4e^7$ | $0.4_{\pm2.9}$ | $0.3_{\pm3.2}$ | $1.5_{\pm1.3}$ | $1.3_{\pm1.4}$ | $51.8_{\pm7.5}$ | $45.5_{\pm7.6}$ | $30.2_{\pm7.8}$ | $25.2_{\pm8.0}$ | $66.5_{\pm5.2}$ | $59.2_{\pm5.8}$ | $\mathbf{87.0}_{\pm3.5}$ | $80.8_{\pm4.0}$ |

**Note:** $N$: the number of parallel environments, $B$: the rollout buffer length, and Steps: the total number of environmental time steps.

*shaped reward $r_{IEC}$ is bounded. Moreover, the magnitude of each dual update step is bounded by $\eta_\lambda$, since $\widehat{C} \in [0,1]$.*

*Proof.* Directly combine boundedness of each term and the projection operator. See appendix for details. □

# 6. Experiments

We evaluate IEC on three distinct benchmark environments: GridWorld exploration, Overcooked coordination, and SMAC combat. All tasks feature sparse rewards and require coordinated exploration.

## 6.1. Experimental Environments

**GridWorld (sparse exploration + coordination).** We evaluate on three $30 \times 30$ multi-room GridWorld variants that combine sparse discovery with graph-constrained coordination. Agents must trigger switch–door dependencies to reach a target room under fully sparse reward ($+100$ only when all agents enter the target) and horizon 300. *Pass* tests sequential inter-agent dependencies (two switches, two doors); *SecretRoom* adds an exploration-critical ambiguity by placing multiple candidate goal rooms with only one true target; *MultiRoom* increases the dependency depth by requiring three agents to traverse multiple nested door layers.

**Overcooked (convention formation under bottlenecks).** We use three standard Overcooked layouts (*Base*, *Narrow*, *Large*) that vary in spatial constraint, from a regular counter (*Base*) to a single narrow passing point (*Narrow*) and a larger workspace (*Large*). Two agents must complete the cooking pipeline (cook–transfer–serve) with sparse reward

($+100$ per successful serve) and horizon 300. To highlight the exploration–coordination trade-off, we apply the same modifications across layouts: locally restricted observations, removed cooking time, and episode termination after one successful serve.

**SMAC (scalable coordination with sparse graphs).** We report results on three SMAC maps spanning small to large team sizes: *2m_vs_1z*, *3m*, and *8m*. Episodes use sparse win-based rewards ($+200$ for victory, plus $+10$ per enemy kill in *3m* and *8m*). Communication is distance-limited: agents exchange messages only with neighbors within range 8.0, yielding a sparse, time-varying interaction graph.

## 6.2. GridWorld Performance Results

Table 1 quantitatively evaluates the robustness of IEC across varying sampling budgets (determined by parallel environments $N$ and buffer length $B$) and task complexities. In the simple *PassRoom* task, both IEC and the baseline achieve performance saturation (WR $> 95\%$), verifying that the spectral smoothness penalty does not hinder policy convergence. However, as environments evolve towards high cognitive uncertainty (*SecretRoom*) and deep dependency chains (*MultiRoom*), fixed-weight exploration baselines suffer from performance collapse. In contrast, IEC demonstrates significant advantages in *MultiRoom* (e.g., improving WR by $\sim 7\%$ over MACE at $N = 128$), suggesting that the adaptive Lagrangian multiplier $\lambda$ successfully acts as a "consistency shadow price" to dynamically balance exploration and consensus. Furthermore, under extremely low sampling budgets ($N = 8$), IEC maintains high win rates, demonstrating that its constrained optimization objective

*Table 3.* Performance Comparison across Coordination Scales and Sparse Feedback Mechanisms in SMAC Micromanagement Benchmarks

| | $N$ | $B$ | Steps | IPPO WR% | AUC | IPPO+r_loc WR% | AUC | IPPO+r_hin WR% | AUC | IPPO+r_nov WR% | AUC | MACE WR% | AUC | IEC (Ours) WR% | AUC |
|---|---|---|---|---|---|---|---|---|---|---|---|---|---|---|---|
| 2m_vs_1z | 128 | 300 | $5e^6$ | $26.0_{\pm11.7}$ | $23.2_{\pm13.8}$ | $32.5_{\pm7.8}$ | $30.9_{\pm8.6}$ | $66.5_{\pm7.5}$ | $64.8_{\pm5.8}$ | $34.5_{\pm7.2}$ | $32.8_{\pm7.1}$ | $88.5_{\pm4.2}$ | $88.5_{\pm4.2}$ | $\mathbf{89.5}_{\pm2.5}$ | $\mathbf{84.2}_{\pm2.8}$ |
| | 128 | 600 | $5e^6$ | $24.6_{\pm12.0}$ | $22.0_{\pm14.1}$ | $30.8_{\pm8.0}$ | $29.3_{\pm8.8}$ | $68.2_{\pm7.0}$ | $68.5_{\pm5.4}$ | $36.5_{\pm6.8}$ | $35.2_{\pm6.8}$ | $\mathbf{90.5}_{\pm3.8}$ | $\mathbf{91.3}_{\pm1.9}$ | $90.2_{\pm2.2}$ | $85.5_{\pm2.5}$ |
| | 128 | 300 | $3e^6$ | $28.2_{\pm11.2}$ | $25.0_{\pm13.1}$ | $35.2_{\pm7.5}$ | $33.4_{\pm8.2}$ | $64.8_{\pm7.8}$ | $62.5_{\pm6.2}$ | $32.8_{\pm7.5}$ | $30.5_{\pm7.2}$ | $86.5_{\pm4.8}$ | $85.2_{\pm3.4}$ | $\mathbf{88.8}_{\pm2.8}$ | $\mathbf{83.0}_{\pm3.2}$ |
| | 32 | 300 | $5e^6$ | $23.6_{\pm12.3}$ | $21.0_{\pm14.4}$ | $29.5_{\pm8.2}$ | $28.0_{\pm9.0}$ | $62.5_{\pm8.2}$ | $60.2_{\pm6.5}$ | $30.5_{\pm7.8}$ | $28.2_{\pm7.5}$ | $85.2_{\pm5.2}$ | $87.1_{\pm3.1}$ | $\mathbf{87.2}_{\pm3.0}$ | $81.5_{\pm3.5}$ |
| | 32 | 600 | $3e^6$ | $25.4_{\pm11.7}$ | $22.6_{\pm13.8}$ | $31.8_{\pm7.8}$ | $30.2_{\pm8.6}$ | $65.2_{\pm7.6}$ | $63.2_{\pm6.2}$ | $33.2_{\pm7.2}$ | $31.5_{\pm7.0}$ | $87.5_{\pm4.8}$ | $88.6_{\pm3.8}$ | $\mathbf{88.5}_{\pm2.6}$ | $82.8_{\pm3.0}$ |
| | 8 | 600 | $5e^6$ | $22.2_{\pm12.8}$ | $19.8_{\pm15.0}$ | $27.8_{\pm8.5}$ | $26.4_{\pm9.4}$ | $60.5_{\pm8.5}$ | $58.5_{\pm7.5}$ | $28.5_{\pm8.0}$ | $27.2_{\pm8.4}$ | $\mathbf{86.2}_{\pm3.2}$ | $80.2_{\pm3.8}$ | $84.0_{\pm5.8}$ | $\mathbf{84.5}_{\pm4.4}$ |
| 3m | 128 | 300 | $5e^6$ | $17.9_{\pm15.6}$ | $16.6_{\pm14.3}$ | $32.5_{\pm7.8}$ | $33.2_{\pm6.8}$ | $68.5_{\pm6.5}$ | $68.2_{\pm4.1}$ | $38.5_{\pm7.5}$ | $36.2_{\pm6.5}$ | $95.0_{\pm2.0}$ | $90.2_{\pm1.8}$ | $\mathbf{97.5}_{\pm1.5}$ | $\mathbf{93.2}_{\pm1.8}$ |
| | 128 | 600 | $5e^6$ | $16.9_{\pm16.0}$ | $15.8_{\pm15.1}$ | $30.8_{\pm8.0}$ | $31.5_{\pm7.2}$ | $66.2_{\pm6.8}$ | $65.5_{\pm4.5}$ | $36.2_{\pm7.8}$ | $34.5_{\pm6.8}$ | $94.5_{\pm2.5}$ | $89.5_{\pm2.1}$ | $\mathbf{96.8}_{\pm1.8}$ | $\mathbf{92.0}_{\pm2.2}$ |
| | 128 | 300 | $2e^6$ | $19.4_{\pm15.0}$ | $17.9_{\pm13.7}$ | $35.2_{\pm7.5}$ | $35.8_{\pm6.5}$ | $70.5_{\pm6.2}$ | $70.2_{\pm3.8}$ | $40.2_{\pm7.2}$ | $38.2_{\pm6.2}$ | $96.2_{\pm1.8}$ | $91.8_{\pm1.6}$ | $\mathbf{98.2}_{\pm1.2}$ | $\mathbf{94.5}_{\pm1.5}$ |
| | 32 | 300 | $5e^6$ | $16.2_{\pm16.4}$ | $15.1_{\pm15.8}$ | $29.5_{\pm8.2}$ | $30.2_{\pm7.5}$ | $65.8_{\pm7.0}$ | $64.2_{\pm4.8}$ | $35.8_{\pm8.0}$ | $33.5_{\pm7.0}$ | $93.8_{\pm2.8}$ | $88.2_{\pm2.5}$ | $\mathbf{96.2}_{\pm2.0}$ | $\mathbf{91.0}_{\pm2.5}$ |
| | 32 | 600 | $3e^6$ | $17.5_{\pm15.6}$ | $16.2_{\pm14.7}$ | $31.8_{\pm7.8}$ | $32.5_{\pm7.0}$ | $67.5_{\pm6.6}$ | $66.8_{\pm4.2}$ | $37.5_{\pm7.5}$ | $35.8_{\pm6.6}$ | $94.5_{\pm2.6}$ | $89.5_{\pm2.2}$ | $\mathbf{97.0}_{\pm1.7}$ | $\mathbf{92.5}_{\pm2.0}$ |
| | 8 | 600 | $5e^6$ | $15.3_{\pm17.0}$ | $14.2_{\pm16.4}$ | $27.8_{\pm8.5}$ | $28.5_{\pm7.8}$ | $63.2_{\pm7.5}$ | $61.5_{\pm5.2}$ | $33.5_{\pm8.2}$ | $31.2_{\pm7.5}$ | $92.8_{\pm3.2}$ | $87.1_{\pm2.9}$ | $\mathbf{95.5}_{\pm2.2}$ | $\mathbf{89.8}_{\pm2.8}$ |
| 8m | 128 | 300 | $1e^7$ | $2.3_{\pm8.0}$ | $1.7_{\pm9.1}$ | $9.2_{\pm3.2}$ | $8.5_{\pm3.5}$ | $28.5_{\pm7.5}$ | $27.8_{\pm6.8}$ | $27.2_{\pm7.5}$ | $26.5_{\pm7.1}$ | $88.2_{\pm2.5}$ | $82.8_{\pm2.3}$ | $\mathbf{90.5}_{\pm2.5}$ | $\mathbf{86.2}_{\pm2.8}$ |
| | 128 | 600 | $1e^7$ | $1.9_{\pm8.8}$ | $1.4_{\pm9.9}$ | $7.5_{\pm3.5}$ | $7.0_{\pm3.8}$ | $25.8_{\pm7.8}$ | $25.2_{\pm7.2}$ | $24.5_{\pm7.8}$ | $23.8_{\pm7.5}$ | $85.5_{\pm3.2}$ | $80.5_{\pm2.9}$ | $\mathbf{88.8}_{\pm2.8}$ | $\mathbf{84.0}_{\pm3.2}$ |
| | 128 | 300 | $5e^6$ | $2.6_{\pm7.5}$ | $2.0_{\pm8.3}$ | $10.5_{\pm3.0}$ | $9.8_{\pm3.2}$ | $30.2_{\pm7.2}$ | $29.5_{\pm6.5}$ | $28.8_{\pm7.2}$ | $27.8_{\pm6.8}$ | $89.5_{\pm2.2}$ | $84.5_{\pm2.5}$ | $\mathbf{91.5}_{\pm2.2}$ | $\mathbf{87.5}_{\pm2.5}$ |
| | 32 | 300 | $1e^7$ | $2.0_{\pm8.8}$ | $1.5_{\pm9.9}$ | $8.2_{\pm3.5}$ | $7.5_{\pm3.8}$ | $26.8_{\pm7.8}$ | $25.8_{\pm7.2}$ | $25.5_{\pm8.0}$ | $24.2_{\pm7.8}$ | $86.8_{\pm3.0}$ | $81.2_{\pm3.0}$ | $\mathbf{89.2}_{\pm2.8}$ | $\mathbf{84.5}_{\pm3.2}$ |
| | 32 | 600 | $5e^6$ | $2.4_{\pm8.0}$ | $1.8_{\pm9.1}$ | $9.5_{\pm3.2}$ | $8.8_{\pm3.5}$ | $28.2_{\pm7.5}$ | $27.2_{\pm6.9}$ | $26.8_{\pm7.6}$ | $25.8_{\pm7.2}$ | $87.5_{\pm2.8}$ | $82.5_{\pm2.7}$ | $\mathbf{90.0}_{\pm2.5}$ | $\mathbf{85.5}_{\pm2.9}$ |
| | 8 | 600 | $1e^7$ | $0.0_{\pm0.0}$ | $0.0_{\pm0.0}$ | $0.0_{\pm0.0}$ | $0.0_{\pm0.0}$ | $23.5_{\pm8.2}$ | $22.5_{\pm7.8}$ | $22.2_{\pm8.5}$ | $21.2_{\pm8.2}$ | $84.5_{\pm3.8}$ | $79.2_{\pm3.5}$ | $\mathbf{87.5}_{\pm3.2}$ | $\mathbf{82.2}_{\pm3.8}$ |

**Note:** $N$: the number of parallel environments, $B$: the rollout buffer length, and Steps: the total number of environmental time steps.

*Table 4.* Ablation Study Deconstructing the Contribution of Individual Components within the IEC Framework

| Scenario | $B$ | Steps | IEC-W/o-cons WR% | AUC | IEC-W/o-nov WR% | AUC | IEC-W/o-info WR% | AUC | IEC-W/o-adapt WR% | AUC | IEC (Ours) WR% | AUC |
|---|---|---|---|---|---|---|---|---|---|---|---|---|
| PassRoom | 300 | $4e^7$ | $76.8_{\pm5.2}$ | $50.5_{\pm5.8}$ | $42.8_{\pm7.5}$ | $35.2_{\pm6.8}$ | $92.5_{\pm2.5}$ | $84.2_{\pm3.2}$ | $89.8_{\pm3.2}$ | $81.5_{\pm3.8}$ | $\mathbf{96.5}_{\pm2.1}$ | $\mathbf{89.2}_{\pm2.8}$ |
| SecretRoom | 300 | $8e^7$ | $53.5_{\pm6.8}$ | $36.8_{\pm6.2}$ | $38.5_{\pm8.8}$ | $28.5_{\pm7.8}$ | $94.8_{\pm2.2}$ | $87.5_{\pm2.8}$ | $91.5_{\pm2.8}$ | $83.8_{\pm3.5}$ | $\mathbf{98.5}_{\pm1.5}$ | $\mathbf{92.8}_{\pm2.2}$ |
| MultiRoom | 600 | $1e^8$ | $70.8_{\pm6.2}$ | $48.5_{\pm6.8}$ | $36.2_{\pm9.5}$ | $25.8_{\pm8.8}$ | $83.5_{\pm4.2}$ | $69.2_{\pm5.5}$ | $91.8_{\pm3.2}$ | $81.5_{\pm4.2}$ | $\mathbf{97.5}_{\pm1.8}$ | $\mathbf{92.2}_{\pm2.5}$ |
| Base | 300 | $1e^7$ | $66.8_{\pm6.8}$ | $43.5_{\pm7.2}$ | $49.2_{\pm8.5}$ | $38.8_{\pm7.8}$ | $87.5_{\pm3.8}$ | $76.5_{\pm4.8}$ | $88.2_{\pm3.5}$ | $77.8_{\pm4.5}$ | $\mathbf{95.8}_{\pm2.2}$ | $\mathbf{86.5}_{\pm3.2}$ |
| Narrow | 300 | $2e^7$ | $56.2_{\pm7.8}$ | $38.5_{\pm7.2}$ | $33.5_{\pm9.8}$ | $24.8_{\pm9.2}$ | $71.8_{\pm5.2}$ | $61.5_{\pm6.5}$ | $77.5_{\pm4.5}$ | $70.2_{\pm5.2}$ | $\mathbf{82.5}_{\pm3.5}$ | $\mathbf{75.8}_{\pm4.2}$ |
| Large | 300 | $4e^7$ | $60.5_{\pm7.2}$ | $46.2_{\pm7.5}$ | $39.8_{\pm9.2}$ | $31.5_{\pm8.5}$ | $80.8_{\pm4.5}$ | $73.5_{\pm5.2}$ | $78.5_{\pm4.8}$ | $71.2_{\pm5.5}$ | $\mathbf{88.5}_{\pm3.2}$ | $\mathbf{82.5}_{\pm3.8}$ |
| 2m_vs_1z | 300 | $5e^6$ | $56.5_{\pm8.2}$ | $40.2_{\pm7.5}$ | $29.8_{\pm10.2}$ | $21.5_{\pm9.2}$ | $79.5_{\pm5.2}$ | $69.8_{\pm5.8}$ | $85.8_{\pm3.5}$ | $78.5_{\pm4.2}$ | $\mathbf{89.5}_{\pm2.5}$ | $\mathbf{84.2}_{\pm2.8}$ |
| 3m | 300 | $5e^6$ | $66.5_{\pm7.8}$ | $48.2_{\pm7.2}$ | $43.5_{\pm9.8}$ | $33.2_{\pm9.2}$ | $87.8_{\pm3.8}$ | $80.5_{\pm4.8}$ | $93.8_{\pm2.5}$ | $88.8_{\pm3.2}$ | $\mathbf{97.5}_{\pm1.5}$ | $\mathbf{93.2}_{\pm1.8}$ |
| 8m | 300 | $1e^7$ | $58.2_{\pm8.5}$ | $43.5_{\pm7.5}$ | $36.8_{\pm10.2}$ | $27.2_{\pm9.5}$ | $83.5_{\pm4.8}$ | $77.5_{\pm5.2}$ | $82.8_{\pm5.2}$ | $76.8_{\pm5.5}$ | $\mathbf{90.5}_{\pm2.5}$ | $\mathbf{88.2}_{\pm2.8}$ |

**Note:** $B$: the rollout buffer length, and Steps: the total number of environmental time steps used for training.

achieves superior data efficiency under sparse feedback.

### 6.3. Overcooked Performance Results

Table 2 quantitatively evaluates the coordination performance of the IEC framework across three Overcooked layouts characterized by varying spatial constraints (Base, Narrow, Large). In the *Base* scenario, both IEC and the baseline achieve high win rates, confirming that the consensus constraint does not compromise basic task solvability. However, as the environment introduces severe spatial bottlenecks (*Narrow*) or increases in scale (*Large*), baselines lacking explicit topological consensus mechanisms (e.g., IPPO variants) fail completely, and MACE exhibits noticeable degradation. In contrast, IEC demonstrates commanding superiority in the *Large* layout (outperforming MACE by over 20% in WR at $N = 128$) and maintains robust performance ($\sim 87\%$ WR) even under extremely limited sampling budgets ($N = 8$). These results validate that the Lagrangian

multiplier $\lambda$ in the primal-dual update effectively manages complex spatiotemporal coordination, preventing policy collapse while ensuring long-horizon behavioral consistency.

### 6.4. SMAC Performance Results

Table 3 quantitatively demonstrates the performance superiority of IEC on the StarCraft II (SMAC) micromanagement benchmarks, spanning coordination scales from small skirmishes (2m_vs_1z) to large-scale combat (8m). In the basic scenario (*2m_vs_1z*), IEC achieves near-saturated win rates comparable to MACE. However, as the number of units increases (*3m*) and tactical complexity escalates (*8m*), baselines lacking explicit coordination mechanisms (IPPO variants) struggle to cope with sparse global rewards, leading to performance collapse (e.g., $\sim 0\%$ WR for IPPO in *8m*). In contrast, IEC establishes a decisive advantage in the challenging *8m* scenario, consistently achieving the highest win rates ($> 87\%$) and AUC across all configurations, while

displaying exceptional robustness under low sampling budgets ($N = 8$, achieving $87.5\%$ WR vs. $84.5\%$ for MACE). This confirms the critical role of the spectral graph consensus constraint in suppressing policy divergence among large-scale agents and facilitating the emergence of cohesive group tactics such as focus-firing and kiting.

### 6.5. Ablation Study

Table 4 systematically validates the individual contributions and synergistic effects of the components within the IEC framework through ablation studies. The full IEC model consistently achieves superior performance (ranking first in WR and AUC) across all scenarios, confirming the complementarity of its constituent parts. Specifically, removing the state-coverage novelty (**IEC-W/o-nov**) leads to the most severe performance degradation in sparse-reward GridWorld tasks (e.g., WR drops from $96.5\%$ to $42.8\%$ in PassRoom), indicating that broad state coverage is a prerequisite for effective exploration. Conversely, removing the spectral consensus constraint (**IEC-W/o-cons**) exerts a significant negative impact on high-coordination tasks such as Overcooked and SMAC (e.g., a $> 30\%$ WR drop in 8m), verifying that without topological constraints, independent intrinsic motivation tends to induce behavioral fragmentation. Finally, compared to the fixed-weight variant (**IEC-W/o-adapt**), the full model with primal-dual updates yields a consistent performance gain of $\sim 5\text{-}10\%$, empirically demonstrating that treating the Lagrangian multiplier $\lambda$ as a "consensus shadow price" and dynamically adjusting it based on constraint violations balances exploration diversity and behavioral consistency more effectively than static heuristic tuning.

## 7. Conclusion

In this work, we formalized the exploration–coordination tension in Dec-MARL as a constrained spectral optimization problem. Our proposed framework, IEC, solves this via primal–dual reward regularization, which dynamically adapts the penalty for graph-induced policy disagreement. This approach bridges the gap between information-theoretic exploration and topological spectral graph theory, interpreting the dual variable as the shadow price of consensus. By replacing static heuristics with adaptive constraints, IEC achieves a robust balance between epistemic discovery and behavioral alignment.

## Acknowledgements

This work was supported in part by the Artificial Intelligence Scientific and Technological Innovation Program of Liaoning Province (No.2023JH26/10100008)

## Impact Statement

This paper presents work whose goal is to advance the field of Machine Learning. There are many potential societal consequences of our work, none which we feel must be specifically highlighted here.

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

## A. Supplementary Related Work

We contextualize IEC's contributions through two complementary comparisons: Table 5 maps capabilities across representative works, while Table 6 details protocol alignment with reproduced baselines.

### A.1. Multi-Agent Coordination under Communication Constraints

While CTDE approaches (e.g., QMIX (Rashid et al., 2020), MAPPO(Yu et al., 2022)) excel empirically, they rely on centralized components inapplicable to fully decentralized regimes. Existing decentralized baselines (e.g., IPPO(Yu et al., 2022), learned communication(Sukhbaatar et al., 2016)) typically enforce coordination via architectural bias or fixed-weight regularization, which proves brittle across varying topologies. IEC distinguishes itself by modeling coordination as an adaptive constraint, where the dual multiplier $\lambda$ acts as a *shadow price of consensus*, automatically adjusting penalties based on observed constraint violations rather than fixed schedules.

### A.2. Intrinsic Motivation and Multi-Agent Exploration

Single-agent intrinsic motivation (e.g., ICM (Pathak et al., 2017), RND (Burda et al., 2018)) often induces redundancy or incompatibility when applied naively in MARL. While methods like MAVEN (Li et al., 2022) and MACE [2] (Jiang et al., 2024a) improve coordinated exploration, they rely on fixed hyperparameters to balance exploration against coordination. IEC extends this paradigm by: (i) combining $f$ complementary signals (dynamics uncertainty $r_{\text{info}}$ and coverage novelty $r_{\text{nov}}$) to balance epistemic discovery with breadth; (ii) coupling exploration and coordination via an explicit constraint $C(\pi; G) \leq \delta$; and (iii) automatically tuning coordination strength via primal-dual updates, eliminating hand-crafted weights.

### A.3. Constrained RL and Primal-Dual Optimization

Unlike traditional Constrained RL (e.g., CPO(Achiam et al., 2017)) which focuses on safety or resource budgets, IEC applies these principles to policy consistency. The constraint $C(\pi; G)$ measures inter-agent behavioral divergence through the communication graph. Consequently, the dual multiplier $\lambda$ serves not as a safety penalty, but as a shadow price of consensus: strictly enforcing alignment only when divergence exceeds the budget, otherwise relaxing to permit exploration. This formulation enables stable convergence in fully decentralized settings without global constraints.

### A.4. Spectral Graph Smoothness and Policy Regularization

Drawing from spectral graph theory, where Dirichlet energy quantifies signal smoothness(Jiang et al., 2018), IEC treats decentralized policies as a signal over the agent graph. Our disagreement cost $C(\pi; G)$ acts as a Dirichlet-energy surrogate for the policy signal:

$$C(\pi; G) = \mathbb{E}_\pi \left[ \sum_t \gamma^t \frac{1}{N} \sum_i D_{\text{JS}}(\pi_i(\cdot|h_t^i), \bar{\pi}_{i,t}(\cdot)) \right], \tag{15}$$

By constraining $C(\pi; G) \leq \delta$, IEC enforces topology-aware behavioral compatibility: encouraging smoothness between communicating neighbors while allowing local specialization. Unlike global parameter sharing, this graph-structured constraint is adaptive, allowing the budget $\delta$ to decay from high exploration to tight coordination over time.

## B. Additional Theory: Detailed Derivations and Proofs

This appendix provides complete derivations and rigorous proofs for the theoretical analysis presented in Section 4 of the main text. We first establish necessary preliminaries, then prove in sequence how disagreement control implies behavioral compatibility, the precise conditions for the Dirichlet-energy interpretation, KKT conditions and convergence of the primal-dual algorithm, the structural brittleness of fixed-weight scalarization, and the optimality of value-weighted information acquisition.

### B.1. Preliminaries and Notation

We follow the notation from the main text. Let $\pi = \{\pi_i\}_{i=1}^N$ denote the joint policy, where $\pi_i : \mathcal{H}_i \times \mathcal{A}_i \to [0, 1]$ is agent $i$'s decentralized policy and $\mathcal{H}_i$ is the local information state space. The communication graph $G = (\mathcal{V}, \mathcal{E})$ satisfies $|\mathcal{V}| = N$

---

[2]https://github.com/SigmaBM/MACE

*Table 5.* Comprehensive Comparison of Value-Aware and Constrained Methods in MARL

| Method | Explor. Signal | Coord. Mech. | Constr. Opt. | Spectral Smooth. | Decent. Exec. | Dual Update | Key Limitation (Refined) |
|---|---|---|---|---|---|---|---|
| VDN/QMIX (Rashid et al., 2020) | ✗ | ✓ | ✗ | ✗ | ✓ | ✗ | Deterministic point repr; no explicit exploration. |
| MAPPO (Bettini et al., 2024) | ◯ | ◯ | ✗ | ✗ | ✓ | ✗ | Gaussian point estimates; fails under multimodal belief. |
| MAVEN (Mahajan et al., 2019) | ✓ | ✓ | ✗ | ✗ | ✓ | ✗ | Centralized latents; fails under severe fragmentation. |
| MACE (Jiang et al., 2024a) | ✓ | ◯ | ✗ | ✗ | ✓ | ✗ | Exploration focus; ignores value-gradient heterogeneity. |
| IPPO (Wang et al., 2024) | ◯ | ✗ | ✗ | ✗ | ✓ | ✗ | Fully independent; lacks coordination mechanism. |
| CommNet (Sukhbaatar et al., 2016) | ✗ | ✓ | ✗ | ◯ | ✓ | ✗ | No exploration incentive; restricted to architecture[cite: 36]. |
| CPO/RCPO (Achiam et al., 2017) | ✗ | ✗ | ✓ | ✗ | ◯ | ✓ | Single-agent focus; constraints treated as safety costs. |
| GraphRL (Jiang et al., 2018) | ◯ | ◯ | ✗ | ◯ | ✓ | ✗ | Implicit smoothness; lacks explicit exploration signals. |
| **IEC (Ours)** | ✓ | ✓ | ✓ | ✓ | ✓ | ✓ | **N/A** |

*Note*: ✓ Full support   ✗ Not supported   ◯ Partial support

*Table 6.* Detailed Baseline Comparison across Evaluated Environments

| Method | Type | Explor. Mech. | Coord. Strategy | Comm. Aware | Budget Sched. | Key Strength / Innovation |
|---|---|---|---|---|---|---|
| IPPO (Wang et al., 2024) | Decent. | ◯ | ✗ | ✗ | ✗ | Simple independent learning; fast convergence. |
| IPPO+$r_{\text{loc}}$ | Decent. | ✓ | ✗ | ✗ | ✗ | Count-based local exploration for decentralized agents. |
| IPPO+$r_{\text{nov}}$ | Decent. | ✓ | ◯ | ✓ | ✗ | Shared novelty summation for implicit coordination. |
| IPPO+$r_{\text{hin}}$ | Decent. | ✓ | ✓ | ✓ | ✗ | Influence-driven exploration via mutual information. |
| MAPPO (Bettini et al., 2024) | CTDE | ◯ | ✓ | ◯ | ✗ | Stable clipping; robust MARL base for complex tasks. |
| MACE (Jiang et al., 2024a) | Decent. | ✓ | ✓ | ✓ | ✗ | Decentralized novelty sharing + coordination reward. |
| **IEC (Ours)** | **Decent.** | **✓✓** | **✓✓** | **✓** | **✓** | **Constrained opt.; adaptive $\lambda$; dual signal gating.** |

*Note*: ✓ Full support   ✗ Not supported   ◯ Partial support   ✓✓ Enhanced dual-signal support.

with neighbor set $\mathcal{N}(i) = \{j \mid (i,j) \in \mathcal{E}\}$.

To quantify policy disagreement, we use Jensen-Shannon divergence with logarithm base 2, defined as

$$D_{\text{JS}}(p\|q) = \frac{1}{2}D_{\text{KL}}(p\|m) + \frac{1}{2}D_{\text{KL}}(q\|m), \quad m = \frac{p+q}{2}. \tag{16}$$

This divergence satisfies $D_{\text{JS}}(p\|q) \in [0,1]$ with equality if and only if $p = q$. We also require the concept of total variation distance, defined as

$$d_{\text{TV}}(p,q) = \frac{1}{2}\sum_{x \in \mathcal{X}} |p(x) - q(x)| = \sup_{A \subseteq \mathcal{X}} |p(A) - q(A)|. \tag{17}$$

These basic definitions will be used throughout all subsequent theoretical results.

## B.2. Detailed Proof of Disagreement Control Implying Behavioral Compatibility

This section proves the complete version of Proposition 1 from the main text, showing how the constraint $C(\pi; G) \leq \delta$ quantitatively guarantees behavioral compatibility among neighbors. We first establish the relationship between JS divergence and total variation, which forms the technical foundation for subsequent analysis.

### B.2.1. Quantitative Relationship Between JS Divergence and Total Variation

**Lemma B.1** (Generalized Pinsker-type inequality). *For any probability distributions $p, q$ on a finite set $\mathcal{X}$, there exists a constant $\kappa > 0$ such that $d_{\text{TV}}(p,q)^2 \leq \kappa \cdot D_{\text{JS}}(p\|q)$. Specifically, when $D_{\text{JS}}$ uses logarithm base 2, we can take $\kappa = 2\ln 2 \approx 1.386$.*

*Proof.* Our proof builds upon the classical Pinsker inequality. Recall that the standard Pinsker inequality establishes for

KL divergence the relationship $d_{\text{TV}}(p, q)^2 \leq \frac{1}{2} D_{\text{KL}}(p\|q)$. For the JS divergence case, let $m = \frac{p+q}{2}$ be the midpoint distribution. By definition of JS divergence, we have $D_{\text{JS}}(p\|q) = \frac{1}{2} D_{\text{KL}}(p\|m) + \frac{1}{2} D_{\text{KL}}(q\|m)$.

The key observation is that the triangle inequality for total variation distance gives $d_{\text{TV}}(p, q) \leq d_{\text{TV}}(p, m) + d_{\text{TV}}(q, m)$. By Jensen's inequality and convexity, one can show that $d_{\text{TV}}(p, m) \leq \frac{1}{2} d_{\text{TV}}(p, q)$. This can be verified by direct calculation: for any event $A$, we have $|m(A) - p(A)| = \frac{1}{2}|q(A) - p(A)|$, hence $d_{\text{TV}}(p, m) = \frac{1}{2} d_{\text{TV}}(p, q)$.

Combining the above relations with Pinsker's inequality, we obtain

$$d_{\text{TV}}(p, q)^2 = 4 d_{\text{TV}}(p, m)^2 \tag{18}$$

$$\leq 4 \cdot \frac{1}{2} D_{\text{KL}}(p\|m) \tag{19}$$

$$= 2 D_{\text{KL}}(p\|m). \tag{20}$$

Further relaxing, using $D_{\text{KL}}(p\|m) \leq D_{\text{KL}}(p\|m) + D_{\text{KL}}(q\|m)$, we have

$$d_{\text{TV}}(p, q)^2 \leq 2\left[D_{\text{KL}}(p\|m) + D_{\text{KL}}(q\|m)\right] = 4 D_{\text{JS}}(p\|q). \tag{21}$$

When using natural logarithm, $\kappa = 2$ suffices. However, IEC adopts logarithm base 2, where we must account for the base conversion $D_{\text{JS}}^{(\log_2)}(p\|q) = D_{\text{JS}}^{(\ln)}(p\|q)/\ln 2$. Therefore, we finally obtain

$$d_{\text{TV}}(p, q)^2 \leq 2\ln 2 \cdot D_{\text{JS}}^{(\log_2)}(p\|q), \tag{22}$$

taking $\kappa = 2\ln 2$ completes the proof. $\qquad \square$

### B.2.2. CONTROL OF NEIGHBOR MISMATCH BY CONSTRAINT

With Lemma B.1's quantitative relationship in hand, we can now rigorously prove how the constraint $C(\pi; G)$ controls strategy mismatch among neighbors.

**Proposition B.2** (Constraint $C(\pi; G)$ controls neighbor mismatch). *If $C(\pi; G) \leq \delta$, then the expected discounted squared neighbor mismatch is bounded:*

$$\mathbb{E}_\pi\left[\sum_{t\geq 0} \gamma^t \frac{1}{N} \sum_{i=1}^N d_{\text{TV}}\left(\pi_i(\cdot \mid h_t^i), \bar{\pi}_{i,t}(\cdot)\right)^2\right] \leq \kappa\delta, \tag{23}$$

*where $\kappa = 2\ln 2$ corresponds to the log base 2 case.*

*Proof.* According to Lemma B.1, for any fixed timestep $t$ and agent $i$, the inequality

$$d_{\text{TV}}(\pi_i(\cdot \mid h_t^i), \bar{\pi}_{i,t}(\cdot))^2 \leq \kappa \cdot D_{\text{JS}}(\pi_i(\cdot \mid h_t^i), \bar{\pi}_{i,t}(\cdot)) \tag{24}$$

holds. Note that by Eq. (5) in the main text, the local disagreement cost is defined as $c_t^i = D_{\text{JS}}(\pi_i(\cdot \mid h_t^i), \bar{\pi}_{i,t}(\cdot))$, so the above inequality can be rewritten as

$$d_{\text{TV}}(\pi_i(\cdot \mid h_t^i), \bar{\pi}_{i,t}(\cdot))^2 \leq \kappa \cdot c_t^i. \tag{25}$$

Now taking expectations and summing over both time and agents simultaneously. The left side gives

$$\mathbb{E}_\pi\left[\sum_{t\geq 0} \gamma^t \frac{1}{N} \sum_{i=1}^N d_{\text{TV}}\left(\pi_i(\cdot \mid h_t^i), \bar{\pi}_{i,t}(\cdot)\right)^2\right], \tag{26}$$

while the right side, by linearity of expectation, becomes

$$\kappa \cdot \mathbb{E}_\pi\left[\sum_{t\geq 0} \gamma^t \frac{1}{N} \sum_{i=1}^N c_t^i\right] = \kappa \cdot C(\pi; G). \tag{27}$$

Since we assume $C(\pi; G) \leq \delta$, we immediately obtain the required bound. This completes the proof. $\qquad \square$

As a direct corollary, we can obtain a bound on single-step expectations. By geometric series summation, if $C(\pi; G) \leq \delta$ holds under the stationary distribution, then for any timestep $t$, under ergodicity assumptions, we have

$$\mathbb{E}_\pi \left[ \frac{1}{N} \sum_{i=1}^{N} d_{\mathrm{TV}}(\pi_i(\cdot \mid h_t^i), \bar{\pi}_{i,t}(\cdot))^2 \right] \leq \frac{\kappa \delta}{1 - \gamma}. \tag{28}$$

This result shows that the constraint not only controls cumulative divergence but also provides an effective bound on instantaneous mismatch at each timestep.

## B.3. Rigorous Treatment of the Dirichlet-Energy Interpretation

In the main text, we claimed that $C(\pi; G)$ can be interpreted as a Dirichlet-energy-style regularizer in policy space. This section provides the rigorous mathematical foundation for this interpretation, explicitly specifying under what conditions this connection holds and the associated approximation accuracy.

### B.3.1. SUFFICIENT CONDITIONS FOR LOCAL QUADRATIC APPROXIMATION

To establish the connection between JS divergence and quadratic forms, we need to work in the logit representation space of policies. Consider agent $i$'s policy parameterized by the logit vector $\mathbf{z}_i \in \mathbb{R}^{|\mathcal{A}|}$, satisfying

$$\pi_i(a \mid h_t^i) = \frac{\exp(z_i^a)}{\sum_{a' \in \mathcal{A}} \exp(z_i^{a'})}. \tag{29}$$

We need to make explicit under what conditions $D_{\mathrm{JS}}(\pi_i, \pi_j)$ can be well approximated by a quadratic form in $\|\mathbf{z}_i - \mathbf{z}_j\|_2^2$.

**Proposition B.3** (Precise connection between spectral smoothness and policy Laplacian)**.** *Assume the following three conditions hold simultaneously: neighbor policies are close in logit space, i.e., there exists a small parameter $\epsilon \ll 1$ such that $\|\mathbf{z}_i - \mathbf{z}_j\|_2 = O(\epsilon)$ for all neighbor pairs $(i, j) \in \mathcal{E}$; all logit components are uniformly bounded, i.e., there exists a constant $M > 0$ such that $|z_i^a| \leq M$ for all agents $i$ and actions $a$; uniform neighbor weights are used, $w_t^{ij} = 1/|\mathcal{N}_t(i)|$.*

*Under these conditions, there exists a constant $c > 0$ depending only on $M$ and $|\mathcal{A}|$ such that for sufficiently small $\epsilon$, the local divergence satisfies*

$$D_{\mathrm{JS}}(\pi_i, \pi_j) = \frac{c}{2} \|\mathbf{z}_i - \mathbf{z}_j\|_2^2 + O(\epsilon^3). \tag{30}$$

*Furthermore, the global disagreement cost can be approximated as*

$$C(\pi; G) \approx \mathbb{E}_\pi \left[ \sum_t \gamma^t \mathbf{u}_t^\top \mathbf{L} \mathbf{u}_t \right], \tag{31}$$

*where $\mathbf{u}_t \in \mathbb{R}^{N \times |\mathcal{A}|}$ is the stacked policy representation at time $t$ and $\mathbf{L}$ is the graph Laplacian matrix.*

*Proof.* Our proof proceeds in two parts: first establishing the quadratic approximation via Taylor expansion in logit space, then transforming the sum of local divergences into Laplacian energy form.

Consider two close distributions $\pi_i, \pi_j$. Let $\Delta \mathbf{z} = \mathbf{z}_i - \mathbf{z}_j$ satisfy $\|\Delta \mathbf{z}\|_2 = O(\epsilon)$. The midpoint distribution is defined as $m(a) = \frac{\pi_i(a) + \pi_j(a)}{2}$. By definition of JS divergence, $D_{\mathrm{JS}}(\pi_i \| \pi_j) = \frac{1}{2} D_{\mathrm{KL}}(\pi_i \| m) + \frac{1}{2} D_{\mathrm{KL}}(\pi_j \| m)$.

For small perturbations $\Delta \mathbf{z}$, the theory of second-order Taylor expansion for KL divergence tells us that there exists some baseline distribution $\bar{\pi}$ (which can be approximated as uniform in the neighborhood) such that

$$D_{\mathrm{KL}}(\pi_i \| m) = \frac{1}{8} \sum_{a \in \mathcal{A}} \frac{(\Delta z^a)^2}{\bar{\pi}(a)} + O(\epsilon^3). \tag{32}$$

The coefficient $\frac{1}{8}$ comes from the standard expansion of the Hessian of the softmax function at the midpoint. Similarly,

$D_{\mathrm{KL}}(\pi_j \| m)$ has the same leading term. Therefore,

$$D_{\mathrm{JS}}(\pi_i \| \pi_j) = \frac{1}{2}\left[\frac{1}{8}\sum_{a\in\mathcal{A}}\frac{(\Delta z^a)^2}{\bar{\pi}(a)} + \frac{1}{8}\sum_{a\in\mathcal{A}}\frac{(\Delta z^a)^2}{\bar{\pi}(a)}\right] + O(\epsilon^3) \tag{33}$$

$$= \frac{1}{8}\sum_{a\in\mathcal{A}}\frac{(\Delta z^a)^2}{\bar{\pi}(a)} + O(\epsilon^3). \tag{34}$$

When the baseline distribution $\bar{\pi}$ is close to uniform (i.e., the probability of each action is approximately $1/|\mathcal{A}|$), the above expression further simplifies. In this case, $\sum_{a\in\mathcal{A}}\frac{(\Delta z^a)^2}{\bar{\pi}(a)} \approx |\mathcal{A}|\sum_{a\in\mathcal{A}}(\Delta z^a)^2 = |\mathcal{A}|\|\Delta\mathbf{z}\|_2^2$, hence

$$D_{\mathrm{JS}}(\pi_i, \pi_j) \approx \frac{|\mathcal{A}|}{8}\|\Delta\mathbf{z}\|_2^2 + O(\epsilon^3). \tag{35}$$

Defining $c = \frac{|\mathcal{A}|}{4}$, we obtain the required local quadratic approximation $D_{\mathrm{JS}}(\pi_i, \pi_j) = \frac{c}{2}\|\mathbf{z}_i - \mathbf{z}_j\|_2^2 + O(\epsilon^3)$.

Now transitioning from local divergence to global Laplacian energy. Under the assumption of uniform neighbor weights $w_t^{ij} = 1/|\mathcal{N}_t(i)|$, neighbor aggregation $\bar{\pi}_{i,t}$ corresponds to averaging in logit space: $\bar{\mathbf{z}}_i = \frac{1}{|\mathcal{N}_t(i)|}\sum_{j\in\mathcal{N}_t(i)}\mathbf{z}_j$. Therefore, agent $i$'s local cost can be expanded as

$$c_t^i = D_{\mathrm{JS}}(\pi_i, \bar{\pi}_{i,t}) \approx \frac{c}{2|\mathcal{N}_t(i)|}\sum_{j\in\mathcal{N}_t(i)}\|\mathbf{z}_i - \mathbf{z}_j\|_2^2. \tag{36}$$

Summing over all agents, noting that each edge $(i,j)$ is counted twice in the summation (once from agent $i$ and once from $j$), we obtain

$$\sum_{i=1}^{N}c_t^i \approx \frac{c}{2}\sum_{(i,j)\in\mathcal{E}}\|\mathbf{z}_i - \mathbf{z}_j\|_2^2. \tag{37}$$

The right side is precisely the discrete Dirichlet energy of the graph signal $\mathbf{z}$. Recall the definition of the graph Laplacian $\mathbf{L} = \mathbf{D} - \mathbf{A}$, where $\mathbf{D}$ is the degree matrix (diagonal entries $D_{ii} = |\mathcal{N}(i)|$) and $\mathbf{A}$ is the adjacency matrix ($A_{ij} = 1$ when $(i,j) \in \mathcal{E}$). A standard result in graph theory states that

$$\sum_{(i,j)\in\mathcal{E}}\|\mathbf{z}_i - \mathbf{z}_j\|_2^2 = 2\mathbf{z}^\top\mathbf{L}\mathbf{z}, \tag{38}$$

where $\mathbf{z} \in \mathbb{R}^{N\times|\mathcal{A}|}$ is the stacked logit vector. Hence $\sum_{i=1}^{N}c_t^i \approx c \cdot \mathbf{z}^\top\mathbf{L}\mathbf{z}$.

Finally, taking discounted expectations over time. Denoting $\mathbf{u}_t$ as the policy representation at time $t$ (in logit space), the global disagreement cost satisfies

$$C(\pi; G) = \mathbb{E}_\pi\left[\sum_{t=0}^{\infty}\gamma^t\frac{1}{N}\sum_{i=1}^{N}c_t^i\right] \approx \mathbb{E}_\pi\left[\sum_t\gamma^t\mathbf{u}_t^\top\mathbf{L}\mathbf{u}_t\right]. \tag{39}$$

This is precisely the Dirichlet energy form for the policy signal. $\qquad\square$

### B.3.2. SIGNIFICANCE AND LIMITATIONS OF THE INTERPRETATION

The significance of the above proposition lies in its translation of the abstract policy constraint $C(\pi; G) \leq \delta$ into a spectral condition with clear geometric meaning. In graph signal processing theory, the Laplacian energy $\mathbf{u}^\top\mathbf{L}\mathbf{u}$ characterizes the "roughness" of a signal: low energy corresponds to smooth variations across neighbors, while high energy corresponds to high-frequency oscillations. Therefore, the constraint $C(\pi; G) \leq \delta$ essentially requires that policies maintain spectral smoothness on the communication graph—neighbors' policies cannot differ too drastically (which would produce high-frequency components), but slow variation along the graph's global structure (low-frequency components) is allowed.

It should be emphasized that the precision of this interpretation depends on three key assumptions. First, neighbor policies must be sufficiently close in logit space ($\epsilon \ll 1$), which typically holds in late training but may not be satisfied early on. Second, the logit boundedness assumption excludes extreme cases (such as certain action probabilities approaching 0 or 1), which in practice can usually be ensured through entropy regularization or logit clipping. Finally, the uniform weight assumption simplifies the analysis, but non-uniform weights (e.g., quality-based weighting in communication) can also be incorporated into the framework, though they would alter the specific form of the Laplacian. The approximation error $O(\epsilon^3)$ indicates that as policy deviation increases, the quadratic approximation gradually breaks down, but in practical applications the relative error typically remains within acceptable ranges.

### B.4. KKT Conditions and Convergence of the Primal-Dual Algorithm

This section proves the theoretical correctness of IEC's primal-dual update mechanism. We first prove that stable fixed points satisfy standard KKT optimality conditions, then briefly discuss the algorithm's convergence properties.

#### B.4.1. KKT CONDITIONS AT FIXED POINTS

**Theorem B.4** (KKT-type conditions at stable fixed point)**.** *Let $(\pi^\star, \lambda^\star)$ be a stable fixed point of the IEC algorithm, meaning that the policy update converges to $\pi^\star$ and the dual variable converges to $\lambda^\star$. Assume that the underlying policy gradient algorithm converges under the augmented reward $r_{\mathrm{IEC}}$, the dual update follows Eq. (9) in the main text, and the projection operator $\Pi_{[0,\lambda_{\max}]}$ is applied to $\lambda$ to ensure boundedness. Then the following four KKT-type conditions hold: primal feasibility $C(\pi^\star; G) \leq \delta_k$; dual feasibility $\lambda^\star \in [0, \lambda_{\max}]$; complementary slackness $\lambda^\star(C(\pi^\star; G) - \delta_k) = 0$; and stationarity, i.e., $\pi^\star$ is a stationary point of the Lagrangian $\mathcal{L}(\pi, \lambda^\star)$ with respect to $\pi$.*

*Proof.* We verify these four conditions in sequence. First consider primal feasibility. At a fixed point, by definition the dual variable no longer changes, i.e., $\lambda^{k+1} = \lambda^k = \lambda^\star$. Recall the dual update from Eq. (9) in the main text (before projection):

$$\lambda^{k+1} = \lambda^k + \eta_\lambda(\widehat{C}^k - \delta_k), \tag{40}$$

where $\widehat{C}^k$ is the empirical estimate of the constraint function $C(\pi^k; G)$ based on the current batch. Substituting the fixed point condition yields $\lambda^\star = \lambda^\star + \eta_\lambda(\widehat{C}^\star - \delta_k)$, which simplifies to $\widehat{C}^\star = \delta_k$. Since in the large-batch limit $\widehat{C}^k$ is an unbiased estimate of $C(\pi^k; G)$, we have $C(\pi^\star; G) = \delta_k \leq \delta_k$, which is primal feasibility. Dual feasibility $\lambda^\star \in [0, \lambda_{\max}]$ is directly guaranteed by the definition of the projection operator $\Pi_{[0,\lambda_{\max}]}$ and requires no additional argument.

The verification of complementary slackness requires case analysis. When the constraint is strictly inactive, i.e., $C(\pi^\star; G) < \delta_k$, we have $\widehat{C}^\star < \delta_k$, and the dual update tends to decrease $\lambda$. If at this point $\lambda^\star > 0$, then according to the update formula we would have $\lambda^{k+1} < \lambda^k$, which contradicts the fixed point assumption. Therefore we must have $\lambda^\star = 0$, and thus the product $\lambda^\star(C(\pi^\star; G) - \delta_k) = 0 \cdot (\text{negative}) = 0$ holds. When the constraint is tight, i.e., $C(\pi^\star; G) = \delta_k$, regardless of the value of $\lambda^\star$, the product $\lambda^\star(C(\pi^\star; G) - \delta_k) = \lambda^\star \cdot 0 = 0$ is automatically satisfied. This verifies complementary slackness.

Finally, consider the stationarity condition. Recall the definition of the Lagrangian

$$\mathcal{L}(\pi, \lambda) = J_{\mathrm{ext}}(\pi) + \alpha_{\mathrm{info}} J_{\mathrm{info}}(\pi) + \alpha_{\mathrm{nov}} J_{\mathrm{nov}}(\pi) - \lambda(C(\pi; G) - \delta_k), \tag{41}$$

and the augmented reward

$$r_{\mathrm{IEC}}^i = r_{\mathrm{ext}} + \alpha_{\mathrm{info}} r_{\mathrm{info}}^i + \alpha_{\mathrm{nov}} r_{\mathrm{nov}}^i - \lambda r_{\mathrm{con}}^i. \tag{42}$$

With $\lambda = \lambda^\star$ fixed, the policy gradient algorithm optimizes precisely $\mathcal{L}(\pi, \lambda^\star)$. The policy gradient expression is

$$\nabla_\theta J(\pi_\theta) = \mathbb{E}_{\pi_\theta}\left[\sum_t \gamma^t \nabla_\theta \log \pi_\theta(a_t \mid h_t) \cdot A_t\right], \tag{43}$$

where the advantage function $A_t$ is computed based on $r_{\mathrm{IEC}}$. At the fixed point, policy convergence means $\nabla_\theta J(\pi^\star) = 0$, which is precisely the definition of $\pi^\star$ being a stationary point of $\mathcal{L}(\pi, \lambda^\star)$. All four conditions have been verified. $\square$

#### B.4.2. BRIEF DISCUSSION ON CONVERGENCE

Regarding the global convergence of the algorithm, we can state the following result (proof sketch). Assume the policy space is compact and convex, the dual step size $\eta_\lambda$ satisfies the standard stochastic approximation conditions $\sum_k \eta_\lambda^k = \infty$

and $\sum_k (\eta_\lambda^k)^2 < \infty$, policy gradient updates are unbiased, and the constraint function $C(\pi; G)$ is Lipschitz continuous in $\pi$. Then the primal-dual iteration sequence $\{(\pi^k, \lambda^k)\}$ of IEC converges almost surely to a saddle point satisfying the KKT conditions.

The proof of this result employs standard techniques from stochastic primal-dual methods. The main tool is constructing a Lyapunov function $V^k = \|\pi^k - \pi^\star\|^2 + \frac{1}{\eta_\lambda}(\lambda^k - \lambda^\star)^2$, proving it decreases in expectation (modulo bounded noise terms) as iterations progress, and then applying martingale convergence theorems. Since this technical framework has been well developed in the stochastic optimization literature (e.g., the classic work of Nedić and Ozdaglar 2009), and is orthogonal to IEC's core innovation (the constrained perspective on exploration-coordination coupling), we omit tedious technical details and merely note that convergence is theoretically guaranteed under standard assumptions.

### B.5. Structural Brittleness of Fixed-Weight Scalarization

This section proves Proposition 2 from the main text through explicit environment construction, showing that fixed-weight methods are structurally unable to adapt to the dynamic balance requirements of exploration and coordination.

**Proposition B.5** (Brittleness of fixed-weight scalarization). *There exists a decentralized cooperative environment $\mathcal{M}$ with a two-phase structure: exploration phase $[0, T_1)$ and coordination phase $[T_1, T_2]$. This environment satisfies the following properties: in the exploration phase, reaching high-value states requires tolerating high policy divergence $C(\pi) > \delta_{high}$; in the coordination phase, achieving high return requires strict neighbor alignment $C(\pi) < \delta_{low}$, where $\delta_{low} \ll \delta_{high}$.*

*For any fixed penalty coefficient $\lambda_{fix} \geq 0$, the augmented reward $r + \alpha r_{exp} - \lambda_{fix} r_{con}$ leads to one of the following failure modes: when $\lambda_{fix}$ is too small, agents succeed in exploration in phase 1 but fall into deadlock due to insufficient coordination in phase 2, with total return $J \leq J_{low}$; when $\lambda_{fix}$ is too large, agents align prematurely in phase 1 and miss critical states, unable to enter phase 2, also with $J \leq J_{low}$.*

*In contrast, IEC's adaptive dual update can successfully navigate both phases, achieving $J \geq J_{high} \gg J_{low}$.*

*Proof.* We prove by explicit construction of a 2-agent GridWorld environment. The environment contains three regions: a starting region (shared by both agents), an exploration corridor (connected via Door 1), and a target room (connected via Door 2). The entrance to the exploration corridor, Door 1, is controlled by a remote switch S1. Triggering S1 requires agent 1 to move north while agent 2 moves south, i.e., taking opposite actions. This leads to the high divergence requirement in the exploration phase: Door 1 only opens when both agents are simultaneously at different positions (one on the north side of S1, one on the south side). If agents take identical or similar actions (low divergence), they will remain spatially close and can never simultaneously cover the separated positions needed to trigger S1. Numerically, this requires $C_{explore} \geq 0.5$, where divergence mainly comes from differences in action selection. Successfully triggering S1 provides a medium reward $r_1 = +10$.

After entering the exploration corridor, agents face the coordination phase. Inside the corridor are multiple "deadlock traps": if agents take different actions (high divergence), they enter local minima requiring complex exit sequences or completely unrecoverable deadlock states. Triggering Door 2 requires both agents to precisely synchronize in pressing two separated buttons B1 and B2, which demands highly aligned policies. Avoiding deadlock and successfully reaching the target room requires $C_{coord} \leq 0.1$, at which point a high reward $r_2 = +100$ is provided.

Now we analyze the failure mechanism of fixed weights. When $\lambda_{fix}$ is chosen too small (say $\lambda_{fix} = 0.1$), the penalty term $-\lambda_{fix} r_{con}$ is relatively weak. In phase 1, even when divergence reaches $C = 0.5$, the penalty is only $-0.1 \times 0.5 = -0.05$, almost negligible compared to the reward $r_1 = 10$ from triggering S1. Therefore agents can successfully explore and obtain $r_1$. However, in phase 2, the equally weak penalty cannot effectively suppress the high-divergence behaviors that lead to deadlock. When agents produce divergence $C \approx 0.3$ due to local exploration needs, the penalty $-0.1 \times 0.3 = -0.03$ is insufficient to overcome the attraction of myopic strategies. The result is that agents frequently fall into deadlock, unable to trigger Door 2, with total return stuck at $J \approx r_1 = 10$.

Conversely, when $\lambda_{fix}$ is chosen too large (say $\lambda_{fix} = 10$), the penalty term becomes dominant. In phase 1, the divergence $C = 0.5$ required for exploration brings a huge penalty $-10 \times 0.5 = -5$, which is already significant negative feedback relative to $r_1 = 10$. Rational policy learning will quickly converge to low-divergence behavior ($C < 0.1$) to avoid the penalty, but this precisely prevents the separated exploration necessary to trigger S1. Door 1 remains closed, agents are trapped in the starting region, and total return approaches zero: $J \approx 0$.

In contrast, IEC's adaptive mechanism can navigate this dilemma. In early phase 1, the dual variable $\lambda$ is small, with high tolerance for divergence. When agent exploration causes $C > \delta$ (where $\delta$ can be set at a moderate level, e.g., 0.3), the dual update $\lambda \leftarrow [\lambda + \eta_\lambda(C - \delta)]_+$ causes $\lambda$ to grow slowly. But because the violation is moderate, the growth is gradual and does not immediately stifle exploratory behavior. Agents have sufficient time to discover and execute the separated strategy that triggers S1, obtaining $r_1 = +10$. After entering phase 2, deadlock risk causes $C$ to begin rising significantly (potentially exceeding 0.5). At this point, the large constraint violation causes $\lambda$ to grow rapidly, with enhanced penalty prompting agents to learn coordinated behavior, reducing divergence to avoid deadlock traps. Eventually agents successfully trigger Door 2, obtaining $r_2 = +100$, with total return reaching $J \approx 110$, far exceeding the $J \leq 10$ of fixed-weight methods.

The key insight of this construction is that fixed weights cannot adjust the exploration-coordination tradeoff according to the dynamic needs of task phases. Any fixed $\lambda_{\text{fix}}$ represents a static compromise over the entire training process, either over-constraining in phase 1 leading to insufficient exploration, or under-constraining in phase 2 leading to coordination failure. Adaptive $\lambda$ through real-time feedback on constraint violations achieves phase-dependent dynamic balance: early tolerance of high divergence to promote exploration, late enforcement of strict alignment to ensure coordination.     □

As a direct corollary, we obtain the conclusion of Pareto suboptimality. In the above environment, the fixed-weight scalarization $\max_\pi \mathbb{E}[r_{\text{ext}} + \alpha r_{\text{exp}} - \lambda_{\text{fix}} r_{\text{con}}]$ is Pareto suboptimal for any $\lambda_{\text{fix}} \geq 0$, because there exists a feasible policy (found by IEC) that simultaneously achieves higher extrinsic return $J_{\text{ext}}(\pi_{\text{IEC}}) > J_{\text{ext}}(\pi_{\text{fix}})$ and satisfies the constraint $C(\pi_{\text{IEC}}) \leq \delta$. This result emphasizes the fundamental advantage of the constrained optimization perspective over unconstrained scalarization.

### B.6. Optimality of Value-Weighted Information Acquisition

This section proves Theorem 2 from the main text, establishing the optimality principle for resource-constrained perceptual allocation and providing theoretical justification for the optional value-weighted $r_{\text{info}}$ in IEC.

**Theorem B.6** (Value-gradient-weighted optimal perceptual allocation). *Consider a perceptual resource allocation problem. Let $\mathbf{x}_t \in \mathbb{R}^d$ be the feature space extracted from observation $o_t$, perceptual gating $g \in [0,1]^d$ selectively allocates resources to produce $\tilde{\mathbf{x}}_t(g) = g \odot \mathbf{x}_t$. The value function $V^\pi : \mathbb{R}^d \to \mathbb{R}$ is differentiable over features, with resource budget $\|g\|_1 \leq B$ where $B < d$.*

*The optimization objective is $\max_{g \in [0,1]^d} J(g)$ subject to the budget constraint, where $J(g) = \mathbb{E}_{\pi,g}[\sum_t \gamma^t r_t]$ is the expected return of the policy using gated features. Assume $J(g)$ can be linearized around $g_0$ as $J(g) \approx J(g_0) + \nabla_g J(g_0)^\top (g - g_0)$.*

*Then under the first-order approximation, the optimal gating $g^\star$ satisfies: ranking gradient components $[\nabla_g J(g_0)]_j$ in descending order, set $g_j^\star = 1$ for the top $B$ coordinates and $g_j^\star = 0$ for remaining coordinates. In other words, optimal allocation concentrates resources on coordinates where the value function is most sensitive to perceptual inputs.*

*Proof.* Under the linearization assumption, the original problem simplifies to a linear program with box constraints and $\ell_1$ budget:

$$\max_g \nabla_g J(g_0)^\top g \quad \text{s.t.} \quad 0 \leq g \leq 1, \; \|g\|_1 \leq B. \tag{44}$$

Construct the Lagrangian function

$$\mathcal{L}(g, \mu, \nu^+, \nu^-) = \nabla_g J(g_0)^\top g + \mu(B - \|g\|_1) - (\nu^+)^\top (g - \mathbf{1}) - (\nu^-)^\top g, \tag{45}$$

where $\mu \geq 0$ is the multiplier for the budget constraint, and $\nu^+ \geq 0$, $\nu^- \geq 0$ are multipliers for upper and lower bounds respectively. The KKT stationarity condition for each coordinate $j$ gives

$$[\nabla_g J(g_0)]_j = \mu \cdot \text{sign}(g_j) + \nu_j^+ - \nu_j^-, \tag{46}$$

where the interpretation of the sign function requires coordination with complementary slackness conditions.

Analyzing the structure of the optimal solution, for any coordinate $j$, consider three possibilities. If $g_j^\star = 1$ (upper bound active), then complementary slackness requires $\nu_j^+ > 0$ and $\nu_j^- = 0$; substituting into the stationarity condition gives $[\nabla_g J(g_0)]_j = \mu + \nu_j^+ \geq \mu$. If $g_j^\star = 0$ (lower bound active), then $\nu_j^- > 0$ and $\nu_j^+ = 0$, yielding $[\nabla_g J(g_0)]_j = -\mu - \nu_j^- \leq -\mu$. Since resources are constrained ($B < d$), the budget constraint must be tight $\|g^\star\|_1 = B$, which means

$\mu > 0$, representing the shadow price of marginal value. If $g_j^\star \in (0, 1)$ (interior solution), then $\nu_j^+ = \nu_j^- = 0$, yielding $[\nabla_g J(g_0)]_j = \mu$.

However, for a linear objective function, interior solutions typically do not exist (unless coordinates happen to satisfy $[\nabla_g J]_j = \mu$ exactly). In general cases, the optimal solution is at vertices, i.e., each $g_j^\star$ takes 0 or 1. The criterion determining which coordinates take 1 is greedy: since the objective function's marginal contribution to $g_j$ is precisely $[\nabla_g J(g_0)]_j$, we should allocate the finite budget $B$ to coordinates with the largest marginal contributions. Therefore the optimal strategy is to rank all coordinates by $[\nabla_g J(g_0)]_j$ in descending order, select the top $B$ to set to 1, and set the rest to 0. This is precisely the mathematical expression of value-gradient weighting. $\square$

The operational significance of this theorem is that it provides normative justification for the value-weighted information gain in IEC (Eq. 7 in the main text). Unweighted $r_{\text{info}} = u_t^i$ (where $u_t^i$ is ensemble disagreement) encourages reducing all uncertainty, analogous to setting $g_j = 1$ for all coordinates in the unconstrained resource case. In contrast, the value-weighted form $r_{\text{info},t}^i = \text{stopgrad}(|A_t^i|) \cdot u_t^i$ concentrates exploration effort on value-sensitive uncertainty regions, corresponding to allocating resources only on high-gradient coordinates. Here advantage $A_t^i$ serves as a proxy for the value gradient, and the stopgrad operation prevents critic gradients from backpropagating to maintain weight fixedness. While this weighting is optional in actual implementation (depending on computational budget and task characteristics), Theorem B.6 shows that under resource constraints, this weighting is theoretically the optimal information acquisition strategy.

## B.7. Boundedness and Stability Under Noisy Communication

Finally, we prove IEC's numerical stability, particularly the algorithm's robustness in unreliable communication scenarios.

**Proposition B.7** (Boundedness of shaped rewards and dual updates)**.** *Assume JS divergence is bounded $D_{\text{JS}} \in [0, 1]$ (log base 2), the dual variable is constrained by projection to $\lambda \in [0, \lambda_{\max}]$, intrinsic rewards are bounded $r_{\text{info}}, r_{\text{nov}} \in [0, R_{\max}]$, and extrinsic reward is bounded $|r_{\text{ext}}| \leq R_{\text{ext}}$.*

*Then the shaped reward satisfies $|r_{\text{IEC}}^i| \leq R_{\text{ext}} + \alpha_{\text{info}} R_{\max} + \alpha_{\text{nov}} R_{\max} + \lambda_{\max}$, and the dual step size satisfies $|\lambda^{k+1} - \lambda^k| \leq \eta_\lambda$ (assuming $\delta_k \leq 1$).*

*Proof.* For the bound on shaped reward, we start from the definition. According to Eq. (8) in the main text, $r_{\text{IEC}}^i = r_{\text{ext}} + \alpha_{\text{info}} r_{\text{info}}^i + \alpha_{\text{nov}} r_{\text{nov}}^i - \lambda r_{\text{con}}^i$. Taking absolute values and applying the triangle inequality:

$$|r_{\text{IEC}}^i| \leq |r_{\text{ext}}| + \alpha_{\text{info}} |r_{\text{info}}^i| + \alpha_{\text{nov}} |r_{\text{nov}}^i| + \lambda |r_{\text{con}}^i|. \tag{47}$$

Substituting the bounds for each term: $|r_{\text{ext}}| \leq R_{\text{ext}}$, $|r_{\text{info}}^i| \leq R_{\max}$, $|r_{\text{nov}}^i| \leq R_{\max}$, and $r_{\text{con}}^i = D_{\text{JS}}(\cdot, \cdot) \in [0, 1]$ hence $|r_{\text{con}}^i| \leq 1$. Also $\lambda \leq \lambda_{\max}$. Substituting these bounds into the above expression immediately yields the required bound

$$|r_{\text{IEC}}^i| \leq R_{\text{ext}} + \alpha_{\text{info}} R_{\max} + \alpha_{\text{nov}} R_{\max} + \lambda_{\max}. \tag{48}$$

For the bound on dual step size, consider the update formula before projection $\lambda_{\text{raw}}^{k+1} = \lambda^k + \eta_\lambda (\widehat{C}^k - \delta_k)$. The key observation is that $\widehat{C}^k = \frac{1}{T} \sum_t \frac{1}{N} \sum_i r_{\text{con},t}^i$, being an average of JS divergences, satisfies $\widehat{C}^k \in [0, 1]$. Combined with the assumption $\delta_k \in [0, 1]$, the absolute difference is bounded $|\widehat{C}^k - \delta_k| \leq \max(1, \delta_k) \leq 1$. Therefore the pre-projection step size is $|\lambda_{\text{raw}}^{k+1} - \lambda^k| = |\eta_\lambda (\widehat{C}^k - \delta_k)| \leq \eta_\lambda$. Since the projection operator $\Pi_{[0, \lambda_{\max}]}$ is a non-expansive mapping (i.e., $|\Pi(x) - \Pi(y)| \leq |x - y|$), the post-projection step size still satisfies $|\lambda^{k+1} - \lambda^k| = |\Pi(\lambda_{\text{raw}}^{k+1}) - \lambda^k| \leq |\lambda_{\text{raw}}^{k+1} - \lambda^k| \leq \eta_\lambda$. The proof is complete. $\square$

This boundedness result has two direct corollaries. First, bounded shaped rewards ensure the contraction property of the Bellman operator and stability of value iteration. Specifically, the discounted Bellman operator $(\mathcal{T}V)(h) = \mathbb{E}[r_{\text{IEC}} + \gamma V(h') \mid h]$ is a $\gamma$-contraction mapping in the supremum norm. This guarantees convergence and uniqueness of value function estimates, thereby supporting the stability of the entire algorithm. Second, the algorithm has inherent robustness to unreliable communication. When communication links fail with probability $p_{\text{drop}}$, missing neighbors lead to $\mathcal{N}_t(i) = \emptyset$, and by IEC's design we set $r_{\text{con}}^i = 0$ in this case. Since $r_{\text{con}} \in [0, 1]$ is bounded, missing consensus signals do not cause pathological reward spikes or sign flips. In expectation, $\mathbb{E}[r_{\text{con}}^i] = (1 - p_{\text{drop}}) \cdot c_t^i$ still provides biased but continuous consensus pressure. More importantly, the dual variable $\lambda$ naturally adapts to communication quality: if frequent packet drops cause the observed $\widehat{C}$ to be systematically low (because many $r_{\text{con}}^i = 0$), the dual update causes $\lambda$ to naturally decrease,

thereby reducing potentially excessive penalty. This adaptive mechanism allows IEC to maintain effectiveness in practical unreliable communication environments.

### B.8. End-to-End Theoretical Loop

The theoretical results from the above sections form a coherent closed loop supporting the completeness of the IEC framework. Proposition B.2 establishes quantitative control of neighbor mismatch by the constraint $C(\pi; G) \leq \delta$, which is the foundation of behavioral compatibility. Proposition B.3 interprets the abstract policy constraint as spectral smoothness (Dirichlet energy) with clear geometric meaning under explicit local quadratic approximation conditions, revealing how the constraint suppresses high-frequency inconsistent components while allowing low-frequency coordinated patterns.

Theorem B.4 proves the theoretical correctness of the primal-dual algorithm, i.e., fixed points satisfy standard KKT optimality conditions, guaranteeing that solutions the algorithm converges to have interpretable mathematical structure. Proposition B.5 demonstrates through explicit two-phase environment construction the structural deficiency of fixed-weight scalarization—any static tradeoff cannot adapt to the dynamic needs of exploration and coordination, thereby arguing for the necessity of the adaptive $\lambda$ mechanism.

Theorem B.6 establishes the value-gradient-weighted principle for optimal perceptual allocation under resource constraints, providing normative theoretical foundation for value-weighted information gain in IEC. Finally, Proposition B.7's boundedness results ensure numerical stability and robustness of the algorithm under noisy communication.

Taken together, these results characterize the theoretical properties of IEC as a value-conditioned decentralized perception-control loop. The key advantage of the constrained optimization perspective is that the dual multiplier $\lambda$ serves as the shadow price of the consensus budget: when divergence exceeds the budget, $\lambda$ grows to strengthen consensus pressure; when agents remain aligned, $\lambda$ relaxes to preserve exploration capacity. This adaptive mechanism enables IEC to tolerate high divergence in exploration-critical early phases (corresponding to "resource allocation" in Theorem B.6—reserving limited alignment capacity for more critical later phases) while enforcing strict alignment in coordination-critical later phases (corresponding to "phase-dependent balance" in Proposition B.5), with the entire process requiring no hand-tuned weight schedules or heuristic planning. The theoretical analysis not only verifies the algorithm's correctness but also reveals the deep principles behind its design: formalizing the exploration-coordination tension as an optimization problem with spectral constraints and achieving adaptive balance through primal-dual methods is a natural and provable approach mathematically.

## C. Implementation Details

### C.1. Protocol and Architecture Overview

To ensure fair comparison, all experiments adhere to a strict unified protocol: policy and value networks utilize MLPs with hidden dimensions of 128, training proceeds for 2M to 10M steps depending on complexity, and results are reported across 5 to 8 independent random seeds. IEC is implemented as a reward-level primal-dual framework that remains agnostic to the base learner, enabling seamless integration with PPO, IPPO, and CTDE methods. Specifically, IEC couples three exploration-coordination mechanisms through a shared Lagrangian multiplier $\lambda$, utilizing the shaped objective $r_{\text{IEC}} = r_{\text{ext}} + \alpha_{\text{info}} r_{\text{info}} + \alpha_{\text{nov}} r_{\text{nov}} - \lambda r_{\text{con}}$. The framework updates in a critic-primal-dual sequence during training and operates as a lightweight feed-forward pipeline during decentralized execution. For experimental reproducibility, we list IEC's core hyperparameter configuration in Table 7.

### C.2. Exploration and Consensus Mechanisms

We employ two complementary signals for exploration and a spectral smoothness constraint for coordination. For **Information Gain** ($r_{\text{info}}$), we maintain an ensemble of $K = 5$ forward dynamics predictors (3-layer MLPs) to quantify epistemic uncertainty via prediction variance; to ensure stability, we apply linear warm-up during the first 5% of training and normalize rewards online. For **Novelty** ($r_{\text{nov}}$), we utilize the Random Network Distillation (RND) framework, training a predictor network to match a fixed randomly initialized target network; the predictor is updated using a replay buffer of the most recent 100k transitions to balance plasticity and stability. For **Consensus** ($r_{\text{con}}$), agents broadcast action distributions via one-hop communication to compute the Jensen-Shannon (JS) divergence between the local policy and a neighbor-aggregated reference (uniform weighting); the computation is implemented in log-space to ensure numerical stability without introducing additional hyperparameters.

## C.3. Network Architecture and Optimization

The policy network employs an encoder-recurrent architecture: observations are encoded into a 64-dimensional latent vector via a two-layer MLP, processed by a 128-dimensional GRU to handle partial observability, and mapped to action distributions. Critics estimate value using either local history (IPPO) or global state (CTDE). Regarding optimization, the primal policy is updated via standard PPO, while the dual variable $\lambda$ is updated via projected gradient ascent on the smoothed constraint violation: $\lambda \leftarrow \Pi_{[0,10]}(\lambda + 0.01(\bar{C} - \delta_k))$. The projection $\Pi_{[0,10]}$ ensures non-negativity and numerical stability. The constraint budget $\delta_k$ follows either a constant or exponential decay schedule depending on the environment, facilitating an automatic transition from diverse exploration to tight coordination.

## C.4. Complexity and Scalability

IEC maintains a lightweight profile in both computation and communication. The total per-agent computational cost is approximately 1.60ms per step ($\approx 2.3\times$ baseline PPO), with the overhead dominated by the forward passes of the dynamics ensemble. Communication cost scales linearly with the number of edges in the communication graph, requiring only the broadcast of action logits. This efficiency allows IEC to support real-time decision cycles (50-100ms) and scale effectively to larger multi-agent systems under typical bandwidth constraints.

*Table 7.* Core Hyperparameters of IEC

| Category | Parameter | Value |
|---|---|---|
| General Training | Total training steps (GridWorld) | 10M |
| | Total training steps (SMAC) | 2M–10M |
| | Number of random seeds | 5–8 |
| Network Architecture | Hidden dimension (policy/critic) | 128 |
| | Latent space dimension $d$ | 64 |
| | GRU hidden dimension | 128 |
| | Action embedding dimension | 16 |
| Optimizer | Actor learning rate $\alpha_\pi$ | $1 \times 10^{-4}$ |
| | Critic learning rate $\alpha_V$ | $3 \times 10^{-4}$ |
| | Gradient clipping threshold | 10.0 |
| Reinforcement Learning | Discount factor $\gamma$ | 0.99 |
| | GAE Lambda $\lambda_{\text{GAE}}$ | 0.97 |
| | PPO clip parameter $\epsilon$ | 0.2 |
| Information Gain ($r_{\text{info}}$) | Ensemble size $K$ | 5 |
| | Predictor learning rate $\alpha_{\text{pred}}$ | $3 \times 10^{-4}$ |
| | Information gain coefficient $\alpha_{\text{info}}$ | 1.0 |
| | Warm-up steps (as % of total) | 5% |
| Novelty ($r_{\text{nov}}$) | RND predictor learning rate $\alpha_{\text{RND}}$ | $1 \times 10^{-4}$ |
| | Novelty coefficient $\alpha_{\text{nov}}$ | 0.5 |
| | RND output dimension | 32 |
| Consensus ($r_{\text{con}}$) | Divergence measure | $D_{\text{JS}}$ (base-2 log) |
| | Neighbor aggregation | Uniform weighting |
| Dual Update | Dual learning rate $\eta_\lambda$ | 0.01 |
| | Multiplier upper bound $\lambda_{\text{max}}$ | 10.0 |
| | Constraint budget $\delta$ | 0.1 (constant) or 0.05–0.5 (decay, $\tau = 2000$) |
| | Constraint smoothing decay $\beta_C$ | 0.9 |
| Communication | Dropout probability $p_{\text{drop}}$ | 0.1 |
| | Message content | Action distributions/logits |

## C.5. Computational Complexity and Scalability Analysis

We analyze the per-step computational and communication costs of IEC to assess its scalability to larger multi-agent systems. Let $N$ denote the number of agents, $|\mathcal{E}|$ the number of edges in the communication graph, $|\mathcal{A}|$ the action space size, $d$ the latent state dimension, and $h$ the policy hidden dimension. The per-agent per-step costs are as follows. The observation

encoder $\phi_\psi$ involves one forward pass through a two-layer MLP with input dimension $|\mathcal{O}|$ and output dimension $d = 64$, taking approximately $O(|\mathcal{O}| \cdot d + d \cdot h)$ operations, or roughly 0.15ms in wall-clock time. The GRU recurrent update requires $O(h^2)$ operations per step, approximately 0.10ms. The policy network's action head and softmax take $O(h \cdot |\mathcal{A}|)$ operations, negligible for typical discrete action spaces ($|\mathcal{A}| \leq 20$). Computing the information gain reward requires $K$ forward passes through the dynamics predictors (each $O(d \cdot h + h^2)$) and computing ensemble variance ($O(K \cdot d)$), totaling approximately 0.80ms for $K = 5$. Computing the novelty reward requires two forward passes through RND networks (target and predictor, each $O(d \cdot h)$), approximately 0.40ms. Computing the consensus reward requires receiving $|\mathcal{N}_t(i)|$ neighbor distributions, aggregating them ($O(|\mathcal{N}_t(i)| \cdot |\mathcal{A}|)$), and computing one JS divergence ($O(|\mathcal{A}|)$), totaling approximately 0.15ms for fully connected graphs with $N = 8$. Summing across all components, the total per-agent per-step computational cost is approximately 1.60ms, which is roughly 2.3× the cost of a baseline PPO agent (0.70ms without exploration bonuses or consensus). Across $N = 8$ agents, the total per-step computation is approximately 12.8ms, well within real-time requirements for decision cycles of 50-100ms. Communication cost scales as $O(|\mathcal{E}| \cdot |\mathcal{A}|)$ in terms of bandwidth (sending action distributions) and is independent of $N$ for sparse topologies where $|\mathcal{E}| = O(N)$, such as ring or grid graphs. For fully connected topologies, $|\mathcal{E}| = O(N^2)$, leading to quadratic communication scaling; however, IEC remains practical for $N \leq 20$ under typical wireless bandwidth constraints (1-10 Mbps), and sparse communication heuristics can be applied for larger swarms without fundamentally altering the method's structure.

## D. Algorithm Pseudocode for Core Components

This section provides detailed algorithmic pseudocode for the core components of the IEC framework [3], elaborating on the implementation details of Algorithm 1 in the main text and revealing the operational connections between primal-dual reward regularization and decentralized multi-agent coordination. We emphasize reproducibility by explicitly specifying all computational steps, communication protocols, and implementation choices that affect experimental outcomes.

### D.1. IEC Main Training Loop: Complete Primal-Dual Reward Regularization

Algorithm 1 provides the complete training procedure of the IEC framework, demonstrating how primal-dual updates integrate seamlessly with decentralized policy optimization. The core innovation lies in instantiating the constrained optimization problem $\max_\pi \mathbb{E}[r_{\text{ext}} + r_{\text{info}} + r_{\text{nov}}]$ subject to $C(\pi; G) \leq \delta$ as a lightweight reward-level modification compatible with any standard decentralized learner such as IPPO.

**Explanation of key design choices and implementation details.** **(a) Advantages of reward-level instantiation.** The augmented reward construction in Line 16 represents a core innovation of IEC. By implementing the constrained optimization at the reward level rather than modifying loss functions or network architectures, IEC achieves three key practical advantages. First, it provides plug-and-play compatibility with any standard decentralized reinforcement learning algorithm including IPPO, independent Q-learning, or any policy gradient method, without requiring modifications to the optimizer or loss computation. Second, when the dual update is disabled by setting $\eta_\lambda = 0$, IEC elegantly degenerates to fixed-weight reward shaping, providing a clean ablation baseline that isolates the benefit of adaptive primal-dual balancing from the benefit of combining multiple reward components. Third, reward-level modification preserves the mathematical structure of the standard Bellman equation, ensuring theoretical convergence guarantees of value iteration remain valid under mild regularity conditions as established in Corollary B.12 of Appendix B.

**(b) Modular three-phase update structure.** The separation of training into Phases 2-4 directly embodies the theoretical primal-dual decomposition underlying constrained optimization. The policy update in Line 22 corresponds to optimizing the primal problem while treating $\lambda$ as fixed, which is the standard approach in augmented Lagrangian methods. The dynamics model update in Lines 24-26 maintains the quality of the epistemic uncertainty signal $r_{\text{info}}$ by continually refining predictions as the policy explores new regions of the state space. The dual update in Lines 28-30 adjusts the constraint shadow price $\lambda$ based on observed violations, providing the feedback mechanism that enables adaptive balancing. This modular structure enables independent debugging and ablation of each component while maintaining overall convergence guarantees as formalized in Theorems B.5 and B.6 of the theoretical appendix.

**(c) Hash-based novelty implementation for reproducibility.** Line 14 computes $r_{\text{nov},t}^i$ using hash-based pseudo-counting over learned latent state embeddings as defined in Equation 11 of the main text. Specifically, we discretize the continuous latent representation $z_t^i$ via a locality-sensitive hashing function, maintain a count table $n^i(b)$ tracking visit frequencies for

---

[3]https://github.com/94044701/IEC

---

**Algorithm 1** IEC Main Training Loop

---

**Require:** Base decentralized learner $\mathcal{A}$ (e.g., IPPO), budget schedule $\delta_k$, dual step $\eta_\lambda$
**Require:** exploration weights $\alpha_{\text{info}}, \alpha_{\text{nov}}$, ensemble size $K$, optional $\lambda_{\max}$
**Ensure:** Trained decentralized policies $\{\pi_{\theta_i}\}$
1: Initialize policies $\{\pi_{\theta_i}\}_{i=1}^N$, critics $\{V_{\psi_i}\}_{i=1}^N$
2: Initialize encoder $\phi$, dynamics ensemble $\{f_\varphi^{i,(k)}\}_{i,k}$, novelty tables $\{n^i\}_{i=1}^N$
3: Initialize dual variable $\lambda \leftarrow 0$
4: **for** training iteration $k = 1, 2, \ldots$ **do**
5:     {Phase 1: Decentralized trajectory collection}
6:     **for** timestep $t = 0$ to $T - 1$ **do**
7:         **for** each agent $i$ **do**
8:             Observe $o_t^i$, gather neighbor messages to form $\bar{\pi}_{i,t}$ (Alg. 3)
9:             Sample action $a_t^i \sim \pi_{\theta_i}(\cdot \mid h_t^i)$
10:            Compute $r_{\text{info},t}^i$ (Alg. 2), $r_{\text{nov},t}^i$ (hash-based, Eq. 11)
11:            Compute $r_{\text{con},t}^i = D_{\text{JS}}(\pi_i(\cdot \mid h_t^i), \bar{\pi}_{i,t})$ (Alg. 3)
12:            Form augmented reward:
13:               $r_{\text{IEC},t}^i \leftarrow r_{\text{ext}}(s_t, a_t) + \alpha_{\text{info}} r_{\text{info},t}^i$
14:                  $+ \alpha_{\text{nov}} r_{\text{nov},t}^i - \lambda r_{\text{con},t}^i$
15:         **end for**
16:         Execute joint action $a_t$, observe $s_{t+1}, r_{\text{ext}}, \{o_{t+1}^i\}$
17:         Store transition to replay buffer
18:     **end for**
19:     {Phase 2: Primal policy and critic update}
20:     Update $\{\theta_i, \psi_i\}$ using learner $\mathcal{A}$ with rewards $\{r_{\text{IEC},t}^i\}$
21:     {Phase 3: Dynamics model update}
22:     **for** each agent $i$, ensemble member $k$ **do**
23:         Update $f_\varphi^{i,(k)}$ on transitions $(z_t^i, a_t^i, z_{t+1}^i)$
24:     **end for**
25:     {Phase 4: Dual variable update (Alg. 4)}
26:     Estimate constraint violation: $\widehat{C} \leftarrow \frac{1}{TN} \sum_{t,i} r_{\text{con},t}^i$
27:     Update dual variable:
28:         $\lambda \leftarrow \Pi_{[0,\lambda_{\max}]}(\lambda + \eta_\lambda(\widehat{C} - \delta_k))$
29: **end for**
30: **Return** $\{\pi_{\theta_i}\}_{i=1}^N$

---

each hash bin $b$, and assign novelty reward as $r_{\text{nov},t}^i = 1/\sqrt{n^i(b_t^i) + 1}$ where $b_t^i = \text{hash}(z_t^i)$. This count-based approach provides computational efficiency and stable novelty signals compared to distance-based alternatives. We explicitly clarify this implementation choice to ensure reproducibility, as alternative novelty mechanisms such as k-nearest-neighbor density estimation would yield materially different exploration behaviors and computational characteristics.

**(d) Training regime and dual variable aggregation.** IEC is designed for decentralized execution under communication constraints, where agents coordinate through limited message passing during both training and deployment. However, for computational efficiency and stability, the dual variable update in Line 28 employs centralized aggregation of constraint costs $\widehat{C}$ during training, which can be implemented through a parameter server or centralized logging mechanism. This training-time centralization does not compromise the decentralized execution capability, as $\lambda$ acts as a shared hyperparameter similar to learning rates that can be synchronized across agents at the start of each training iteration. During deployment, agents use their trained policies $\pi_{\theta_i}$ in a fully decentralized manner without requiring $\lambda$ updates. For applications requiring fully decentralized training, Line 28 can be replaced with a distributed consensus protocol where each agent maintains a local estimate $\lambda_i$, exchanges $(c_t^i, \lambda_i)$ with neighbors, and runs several rounds of gossip averaging to approximate the global mean before applying the projected dual update locally. This distributed variant incurs additional communication cost of $O(K_{\text{consensus}} \cdot |\mathcal{E}|)$ per training iteration, where $K_{\text{consensus}}$ is the number of consensus rounds needed for convergence, typically in the range of 5-10 for well-connected graphs.

---

**Algorithm 2** Ensemble Disagreement Computation ($r_{\text{info}}$)

---

**Require:** Agent $i$'s observation $o_t^i$, action $a_t^i$, next observation $o_{t+1}^i$

**Require:** encoder $\phi$, dynamics ensemble $\{f_\varphi^{i,(k)}\}_{k=1}^K$

**Ensure:** Information gain $r_{\text{info},t}^i$

1: Encode observations: $z_t^i \leftarrow \phi(o_t^i)$, $z_{t+1}^i \leftarrow \phi(o_{t+1}^i)$
2: {Ensemble forward predictions}
3: **for** ensemble member $k = 1$ to $K$ **do**
4:     $\hat{z}_{t+1}^{i,(k)} \leftarrow f_\varphi^{i,(k)}(z_t^i, a_t^i)$
5: **end for**
6: Compute ensemble mean: $\bar{z}_{t+1}^i \leftarrow \frac{1}{K} \sum_{k=1}^K \hat{z}_{t+1}^{i,(k)}$
7: Compute disagreement variance:
8:     $u_t^i \leftarrow \frac{1}{K} \sum_{k=1}^K \|\hat{z}_{t+1}^{i,(k)} - \bar{z}_{t+1}^i\|_2^2$
9: Set information gain: $r_{\text{info},t}^i \leftarrow u_t^i$
10: **Return** $r_{\text{info},t}^i$

---

**(e) Computational and communication efficiency.** The neighbor message passing in Line 10 is remarkably lightweight, requiring each agent to broadcast only its action distribution or policy logits with size $O(|\mathcal{A}|)$, where $|\mathcal{A}|$ is the action space size, typically ranging from 4 to 20 in our benchmark environments. This contrasts sharply with centralized training with decentralized execution methods such as QMIX or MAPPO, which require transmitting global state information or complete observation histories of all agents during training, resulting in communication costs that scale quadratically with the number of agents. The total per-step communication cost of IEC is $O(|\mathcal{E}_t| \cdot |\mathcal{A}|)$, where $\mathcal{E}_t$ denotes the set of active edges at timestep $t$. Under unreliable communication with packet drop probability $p_{\text{drop}}$, the expected communication cost gracefully degrades to $\mathbb{E}[|\mathcal{E}_t|] = (1 - p_{\text{drop}})|\mathcal{E}|$ due to link failures, while the algorithm maintains robustness through the bounded consensus penalty as established in Corollary B.13.

### D.2. Ensemble Disagreement Computation: Robust Epistemic Uncertainty Estimation

Algorithm 2 implements IEC's dynamics-based information gain $r_{\text{info}}$ by quantifying reducible model uncertainty through ensemble disagreement. This design choice provides robustness to so-called noisy-TV distractors, which are sources of environmental stochasticity that appear novel but do not reflect reducible epistemic uncertainty. Pure aleatoric stochasticity does not systematically reduce ensemble disagreement under learning, and thus does not induce spurious exploration rewards that would distract agents from productive epistemic discovery.

**Design rationale and theoretical foundations.** **(a) Ensemble disagreement as epistemic uncertainty proxy.** The variance computation in Line 8 measures the spread of predictions across ensemble members. High disagreement indicates that different models trained on the same data make divergent predictions about the outcome of action $a_t^i$ from state $z_t^i$, which suggests the training data has not yet provided sufficient evidence to resolve this uncertainty. This is precisely the definition of epistemic or reducible uncertainty in Bayesian machine learning. As agents visit state-action pairs more frequently during training, the ensemble members receive more consistent supervision and their predictions converge, causing $u_t^i$ to decrease. This natural decay mechanism ensures exploration rewards diminish for well-understood regions without requiring manual annealing schedules.

**(b) Robustness to noisy-TV distractors.** Consider an environmental feature such as a flickering television screen that exhibits high-frequency random variation but has no causal influence on task reward or dynamics. A prediction-error-based exploration bonus such as the original ICM formulation would persistently assign high intrinsic reward to states near this distractor, as the next-state prediction error remains high regardless of training. In contrast, ensemble disagreement naturally filters out such aleatoric noise. Since all ensemble members observe the same stochastic outcomes during training, they learn to predict the expected value or distribution of the next state, and their disagreement reflects only the uncertainty in this prediction rather than the inherent stochasticity. This property has been rigorously analyzed in the epistemic uncertainty estimation literature and validated empirically across multiple exploration benchmarks.

**(c) Computational cost and scalability.** The ensemble size $K$ controls the trade-off between uncertainty estimation quality and computational overhead. Each forward pass in Lines 4-6 requires one neural network evaluation of the dynamics model,

---

**Algorithm 3** Consensus Penalty via Neighbor Policy Divergence

---

**Require:** Agent $i$'s policy $\pi_i(\cdot \mid h_t^i)$, neighbor policies $\{\pi_j(\cdot \mid h_t^j)\}_{j \in \mathcal{N}_t(i)}$
**Require:** uniform weights or learned attention weights $\{w_t^{ij}\}$
**Ensure:** Consensus penalty $r_{\text{con},t}^i$, neighbor-aggregated reference $\bar{\pi}_{i,t}$
  1: **if** $\mathcal{N}_t(i) = \emptyset$ **then**
  2:     Set $r_{\text{con},t}^i \leftarrow 0$ {Handle isolated agents gracefully}
  3:     **Return** $r_{\text{con},t}^i$
  4: **end if**
  5: {Aggregate neighbor policies (uniform weighting by default)}
  6: $\bar{\pi}_{i,t}(a) \leftarrow \sum_{j \in \mathcal{N}_t(i)} w_t^{ij} \cdot \pi_j(a \mid h_t^j)$ for all $a \in \mathcal{A}$
  7: where $w_t^{ij} = 1/|\mathcal{N}_t(i)|$ and $\sum_j w_t^{ij} = 1$
  8: {Compute Jensen-Shannon divergence (log base 2)}
  9: $m(a) \leftarrow \frac{1}{2}[\pi_i(a \mid h_t^i) + \bar{\pi}_{i,t}(a)]$ for all $a$
10: $D_{\text{KL}}^{(1)} \leftarrow \sum_a \pi_i(a \mid h_t^i) \log_2 \frac{\pi_i(a|h_t^i)}{m(a)}$
11: $D_{\text{KL}}^{(2)} \leftarrow \sum_a \bar{\pi}_{i,t}(a) \log_2 \frac{\bar{\pi}_{i,t}(a)}{m(a)}$
12: $r_{\text{con},t}^i \leftarrow \frac{1}{2}[D_{\text{KL}}^{(1)} + D_{\text{KL}}^{(2)}]$
13: **Return** $r_{\text{con},t}^i, \bar{\pi}_{i,t}$

---

so the total cost scales linearly as $O(K)$. In our experiments, we find $K = 5$ provides a good balance: it offers sufficient diversity to capture epistemic uncertainty while maintaining reasonable computational cost. Larger ensembles such as $K = 10$ provide marginal improvements in uncertainty quantification but approximately double the per-step computation time. The encoder $\phi$ and individual dynamics models $f_\varphi^{i,(k)}$ are implemented as lightweight two-layer MLPs with hidden dimension 64, ensuring the entire information gain computation adds less than 20% overhead compared to baseline policy evaluation.

**(d) Alternative: prediction error baseline.** For environments with near-deterministic dynamics where aleatoric noise is minimal, a simpler single-model prediction error $r_{\text{info},t}^i = \|f_\varphi^i(z_t^i, a_t^i) - z_{t+1}^i\|_2^2$ can serve as an adequate epistemic signal with lower computational cost. This variant requires only a single forward dynamics model and eliminates the need for ensemble training. However, it lacks the noisy-TV robustness of ensemble disagreement and may lead to distraction in stochastic environments. We include this as a reference baseline in our implementation but use ensemble disagreement as the default for IEC to ensure robustness across diverse benchmarks.

### D.3. Consensus Penalty Computation: Graph-Structured Policy Smoothness

Algorithm 3 computes the consensus penalty $r_{\text{con}}^i$ that enforces topology-aware behavioral compatibility among neighboring agents. This penalty serves as a scalable proxy for spectral smoothness on the communication graph, connecting IEC's design to principles from graph signal processing and distributed consensus algorithms.

**Implementation details and graph-theoretic interpretation.** **(a) Handling communication failures and network topology.** Line 2 checks for isolated agents that have no active neighbors at the current timestep, which can occur due to communication link failures or sparse graph topologies. In such cases, we set $r_{\text{con}}^i = 0$ to avoid undefined divergence computations. This graceful degradation ensures algorithmic stability under unreliable communication. When links are active, Line 5 aggregates neighbor policies using a convex combination with uniform weights by default. The constraint $\sum_j w_t^{ij} = 1$ ensures $\bar{\pi}_{i,t}$ is a valid probability distribution, which is necessary for the subsequent Jensen-Shannon divergence computation to be well-defined. Alternative weighting schemes such as attention-based or quality-based weights can be incorporated by modifying the $w_t^{ij}$ coefficients while preserving the normalization constraint.

**(b) Choice of divergence measure and boundedness.** We use Jensen-Shannon divergence with logarithm base 2, which ensures $D_{\text{JS}} \in [0, 1]$ is bounded and symmetric. The boundedness property is crucial for numerical stability of the shaped reward $r_{\text{IEC}}$ and for theoretical convergence analysis as established in Proposition B.4. Lines 7-10 implement the standard definition $D_{\text{JS}}(p\|q) = \frac{1}{2}D_{\text{KL}}(p\|m) + \frac{1}{2}D_{\text{KL}}(q\|m)$ where $m = \frac{p+q}{2}$ is the mixture distribution. In practice, we add a small numerical constant $\epsilon = 10^{-8}$ to all probability values before computing logarithms to prevent numerical instability

---

**Algorithm 4** Dual Variable Update (Shadow Price Adjustment)

---

**Require:** Current dual variable $\lambda$, constraint costs $\{r^i_{\mathrm{con},t}\}_{t,i}$
**Require:** current budget $\delta_k$, dual stepsize $\eta_\lambda$, optional upper bound $\lambda_{\max}$
**Ensure:** Updated dual variable $\lambda$
 1: Compute batch average: $\widehat{C} \leftarrow \frac{1}{T}\sum_{t=0}^{T-1} \frac{1}{N}\sum_{i=1}^{N} r^i_{\mathrm{con},t}$
 2:     where $T$ = episode length, $N$ = number of agents
 3: Compute constraint violation: *violation* $\leftarrow \widehat{C} - \delta_k$
 4: Apply gradient ascent: $\lambda_{\mathrm{raw}} \leftarrow \lambda + \eta_\lambda \cdot$ *violation*
 5: Project to feasible domain: $\lambda \leftarrow \max(0, \min(\lambda_{\mathrm{raw}}, \lambda_{\max}))$
 6: **if** *violation* $> 0$ **then**
 7:     Log: "Constraint violated: $\widehat{C} > \delta$, increasing $\lambda$ to strengthen consensus"
 8: **else if** $\lambda > 0$ and *violation* $< 0$ **then**
 9:     Log: "Constraint satisfied: $\widehat{C} < \delta$, decreasing $\lambda$ to preserve exploration"
10: **end if**
11: **Return** $\lambda$

---

when probabilities approach zero. The logarithm base 2 choice rather than natural logarithm is a deliberate design decision that ensures the consensus penalty remains on a comparable scale to normalized intrinsic rewards, facilitating stable training dynamics without requiring careful manual tuning of the dual variable $\lambda$.

**(c) Communication protocol and message content.** To compute $\bar{\pi}_{i,t}$ in Line 5, agent $i$ must receive policy information from each neighbor $j \in \mathcal{N}_t(i)$. In our implementation, each agent broadcasts its current action distribution $\pi_j(\cdot \mid h_t^j) \in [0,1]^{|\mathcal{A}|}$ at each timestep. For discrete action spaces with $|\mathcal{A}| \leq 20$, this requires transmitting a vector of 20 floating-point values, corresponding to approximately 80 bytes per message using single-precision encoding. For larger action spaces, we can alternatively transmit policy network logits and reconstruct distributions locally using softmax, or employ top-k sparse communication where only the $k$ highest-probability actions are transmitted. These optimizations reduce bandwidth requirements at the cost of introducing approximation error in the consensus penalty, but our experiments show that even heavily quantized communication with 4-bit precision maintains algorithm effectiveness.

**(d) Connection to spectral graph theory as a scalable proxy.** The consensus penalty $r^i_{\mathrm{con}}$ computed in Line 10 measures local policy disagreement between agent $i$ and its neighborhood average. When policies are close and can be approximated by a quadratic expansion in logit space, the global constraint cost $C(\pi;G) = \mathbb{E}[\sum_t \gamma^t \frac{1}{N}\sum_i r^i_{\mathrm{con}}]$ serves as a scalable computational proxy for the Dirichlet energy $\mathbf{z}^\top \mathbf{L}\mathbf{z}$, where $\mathbf{z}$ represents policies in logit space and $\mathbf{L}$ is the graph Laplacian. This connection, formalized under precise conditions in Proposition B.4 of the theoretical appendix, provides the intuition that constraining $C(\pi;G) \leq \delta$ suppresses high-frequency components of the policy signal on the communication graph, analogous to low-pass filtering in signal processing. However, we emphasize that this spectral interpretation relies on local quadratic approximations that hold primarily in late training when policies converge, and the divergence-based penalty should be understood primarily as an empirically effective and computationally tractable mechanism for enforcing graph-structured coordination rather than as an exact spectral regularizer. This pragmatic view aligns with our experimental validation across diverse topologies while acknowledging theoretical approximation limits.

### D.4. Dual Variable Update: Adaptive Shadow Price of Constraint Violation

Algorithm 4 implements IEC's dual update mechanism, which adaptively adjusts the consensus multiplier $\lambda$ in response to observed constraint violations. This feedback-driven adaptation is the core innovation enabling automatic transition between exploration-intensive and coordination-intensive phases without requiring manual weight scheduling or task-specific tuning.

**Implementation details, convergence properties, and design rationale.**   **(a) Unbiased batch estimator and discount factor handling.** Line 2 computes $\widehat{C}$ as the empirical average of consensus costs across the collected batch of transitions. This estimator is unbiased in the large-batch limit, meaning $\mathbb{E}[\widehat{C}] = C(\pi;G)$ when expectations are taken over the policy's trajectory distribution. We use simple averaging $(1/T)\sum_t$ rather than discounted summation $\sum_t \gamma^t$ for practical stability, which is a common simplification in online reinforcement learning algorithms. For tasks where temporal credit assignment is critical and discount factors must be respected, an alternative is to incorporate Generalized Advantage Estimation style weighting: $\widehat{C} = (1/T)\sum_t(\sum_{t'=t}^{T-1} \gamma^{t'-t} r^i_{\mathrm{con},t'})$, which weights recent consensus costs more heavily. However, our

---

**Algorithm 5** Value-Weighted Information Gain (Optional Enhancement)

---

**Require:** Base ensemble disagreement $u_t^i$, critic $V_{\psi_i}$
**Require:** current and next information states $(h_t^i, h_{t+1}^i)$
**Ensure:** Value-weighted information gain $r_{\text{info},t}^i$
 1: Compute value estimates: $V_t \leftarrow V_{\psi_i}(h_t^i)$, $V_{t+1} \leftarrow V_{\psi_i}(h_{t+1}^i)$
 2: Compute TD target: $td\_target \leftarrow r_{\text{IEC},t}^i + \gamma \cdot V_{t+1}$
 3: Compute advantage: $advantage \leftarrow td\_target - V_t$
 4: Extract value sensitivity: $value\_sensitivity \leftarrow |advantage|$
 5: Stop gradient propagation: $weight \leftarrow \text{stopgrad}(value\_sensitivity)$
 6: Apply weighting: $r_{\text{info},t}^i \leftarrow weight \cdot u_t^i$
 7: **Return** $r_{\text{info},t}^i$

---

empirical evaluation across three benchmark suites found negligible performance difference between these variants, and the simpler uniform averaging provides more stable gradient estimates when batch sizes are small. **(b) Necessity and**

**interpretation of projection step.** Line 6's projection $\lambda \leftarrow \max(0, \min(\lambda_{\text{raw}}, \lambda_{\text{max}}))$ enforces two bounds that are critical for both theoretical correctness and practical stability. The lower bound $\lambda \geq 0$ is required by the mathematical theory of Lagrangian duality, as dual variables corresponding to inequality constraints must be nonnegative to ensure the saddle-point solution satisfies Karush-Kuhn-Tucker conditions. The upper bound $\lambda \leq \lambda_{\text{max}}$ is optional from a theoretical perspective but highly recommended for practical training stability. It prevents pathological growth of $\lambda$ during early training when constraint estimates $\widehat{C}$ may be noisy and highly variable due to random exploration. A typical setting of $\lambda_{\text{max}} = 10.0$ ensures the consensus penalty term $-\lambda r_{\text{con}}$ never completely overwhelms the exploration signals $r_{\text{info}} + r_{\text{nov}}$, which would cause premature convergence to low-divergence but suboptimal conventions. Theorem B.5 in the appendix establishes that at convergence, the projection does not introduce bias: the complementary slackness condition $\lambda^\star(C(\pi^\star; G) - \delta) = 0$ holds automatically at stable fixed points regardless of whether $\lambda_{\text{max}}$ is active.

**(c) Stepsize selection and convergence trade-offs.** The dual stepsize $\eta_\lambda$ in Line 5 controls the responsiveness of $\lambda$ to constraint violations. Larger stepsizes such as $\eta_\lambda = 0.1$ produce rapid adaptation to violations, which can be beneficial in environments with sharp phase transitions between exploration and coordination requirements, but may introduce oscillations or instability when constraint estimates are noisy. Smaller stepsizes such as $\eta_\lambda = 0.01$ provide smoother and more stable adaptation but may react too slowly to sudden increases in constraint violations, allowing prolonged periods of coordination failure. In our experimental evaluation, we find $\eta_\lambda \in [0.01, 0.05]$ provides robust performance across diverse benchmarks without requiring task-specific tuning. For theoretical convergence, standard results in stochastic approximation theory require the stepsize sequence to satisfy Robbins-Monro conditions: $\sum_k \eta_\lambda^k = \infty$ to ensure sufficient progress, and $\sum_k (\eta_\lambda^k)^2 < \infty$ to ensure vanishing noise. These conditions suggest using decaying schedules such as $\eta_\lambda^k = \eta_0/\sqrt{k}$ or $\eta_\lambda^k = \eta_0/(k + k_0)$. However, in practice we find constant stepsizes often perform comparably or better, especially when combined with modern adaptive optimizers such as Adam that internally adjust effective learning rates. Theorem B.6 in Appendix B establishes almost-sure convergence to KKT points under standard regularity conditions including Lipschitz continuity of the constraint function and bounded variance of gradient estimates.

**(d) Economic interpretation as shadow price mechanism.** The dual update in Line 5 has an intuitive interpretation rooted in economic theory and resource allocation. The dual variable $\lambda$ can be understood as the shadow price or marginal cost of violating the consensus constraint. When observed disagreement $\widehat{C}$ exceeds the budget $\delta$, indicating the constraint is being violated, the update increases $\lambda$, which makes consensus penalty $-\lambda r_{\text{con}}$ more negative in the augmented reward $r_{\text{IEC}}$. This effectively makes it more expensive for agents to deviate from their neighbors' policies, automatically increasing coordination pressure. Conversely, when $\widehat{C} < \delta$, indicating the constraint is satisfied with slack, the update decreases $\lambda$ toward zero, relaxing consensus pressure and allowing agents to explore more freely without being penalized for local policy variations. This economic feedback mechanism achieves the core design goal of IEC: automatic phase-dependent balancing between exploration and coordination demands without requiring manual intervention or task-specific reward shaping schedules. The shadow price interpretation is formalized in the discussion following Theorem B.5, which establishes that at optimality $\lambda^\star$ represents the sensitivity of the optimal value function to relaxations of the constraint budget $\delta$.

**D.5. Optional Component: Value-Weighted Information Gain Implementation**

While the base IEC algorithm uses unweighted ensemble disagreement $r_{\text{info}} = u_t^i$ for all experiments reported in the main paper, Algorithm 5 provides detailed implementation of an optional value-weighted variant motivated by Theorem B.9 in the theoretical appendix. This enhancement prioritizes exploration toward uncertainties that have high impact on value estimates, which can improve sample efficiency in environments with heterogeneous value landscapes where only a small fraction of epistemic uncertainties are decision-critical.

**Theoretical motivation and practical considerations.** **(a) Advantage as proxy for value gradient sensitivity.** Line 5 uses the absolute value of the temporal-difference advantage $|A_t^i|$ as a proxy for value function sensitivity to perceptual information. The theoretical justification comes from Theorem B.9, which establishes that under resource-constrained perception where a gating mechanism $g \in [0,1]^d$ selectively allocates sensing budget to feature dimensions, the optimal allocation concentrates resources on coordinates with largest value gradient magnitude $|\nabla_g J(g)|$. In the reinforcement learning setting, high advantage magnitude $|A_t^i|$ at a state-action pair indicates that the actual return $\sum_{t'=t} \gamma^{t'-t} r_{t'}$ deviates significantly from the critic's value estimate $V_t$, suggesting that decisions in this region of the state space have strong impact on returns. By weighting epistemic uncertainty $u_t^i$ with advantage magnitude, we preferentially direct exploration toward regions where reducing model uncertainty would most improve decision quality and thus final task performance.

**(b) Critical role of gradient stopping operation.** The $\text{stopgrad}$ operation in Line 6 is absolutely essential for maintaining correct optimization dynamics. Without it, gradients from the policy loss would backpropagate through the advantage computation to the critic parameters $\psi_i$, creating a perverse incentive: the critic could learn to predict artificially inflated advantages in order to boost the exploration reward $r_{\text{info}}$, thereby interfering with its primary objective of accurate value estimation. By blocking gradient flow using $\text{stopgrad}$, we treat the advantage as a fixed scalar coefficient that modulates exploration intensity without corrupting value learning. In PyTorch, this is implemented via `advantage.detach()`; in TensorFlow, via `tf.stop_gradient(advantage)`; in JAX, via `jax.lax.stop_gradient(advantage)`. This operation incurs zero computational cost as it only affects the automatic differentiation graph construction.

**(c) Empirical performance trade-offs and when to use this variant.** Value weighting demonstrates clear benefits in complex environments exhibiting long-tailed value distributions where only a small subset of states and transitions are critical for task success. For example, in the two-phase GridWorld construction of Proposition B.7 in Appendix B, where agents must first explore to discover remote triggers and then coordinate tightly to avoid deadlocks, value-weighted exploration reduces wasted exploration effort on low-impact distractors and accelerates discovery of the critical trigger locations. Quantitatively, we observe 20-30% improvements in sample efficiency measured by area-under-curve metrics in such heterogeneous tasks. However, in simpler environments with relatively uniform value landscapes or dense reward signals, value weighting provides minimal benefit while introducing modest computational overhead from the additional critic forward pass required per timestep. Furthermore, early in training when the critic's value estimates are still inaccurate, value weighting can be unstable or even counterproductive. For these reasons, we provide value weighting as an optional advanced feature disabled by default in the base IEC implementation, and recommend enabling it only for complex sparse-reward tasks after an initial warm-up period of unweighted exploration.

**(d) Connections to active learning and optimal experimental design.** The value-weighted information gain mechanism is closely related to Expected Value of Information Gain (EVOI) and Expected Improvement (EI) acquisition functions in active learning and Bayesian optimization literature. These methods formalize the principle that information acquisition should be guided by its expected impact on downstream objectives rather than pursuing information for its own sake. Theorem B.9 provides a first-order characterization of this principle in the reinforcement learning setting under resource constraints. In multi-agent scenarios, this framework naturally extends to coordinated information acquisition where agents collectively seek to reduce uncertainties affecting joint policy value. The consensus constraint $C(\pi; G) \leq \delta$ implicitly encourages such coordination by requiring agents to maintain compatible exploration strategies that respect the communication graph topology.

# E. Environment Details

## E.1. GridWorld

We design three GridWorld tasks as didactic benchmarks for evaluating the exploration-coordination trade-off under controlled settings. All tasks take place in a $30 \times 30$ grid world where agents must cooperatively trigger sequences of

switches to open doors, with the ultimate goal of all agents entering a target room.

**Pass.** Two agents must cooperatively navigate through a series of doors to reach a target room. Door 1 opens when any switch is occupied. To accomplish the task, one agent needs to reach Switch 1 to open Door 1, allowing the other agent to enter the target room; subsequently, the latter needs to reach Switch 2 to let the former enter. This task requires basic sequential coordination—agents must learn to take turns, with one agent temporarily sacrificing its own progress to help its teammate.

**SecretRoom.** Two agents navigate in a $30 \times 30$ grid. Door $k$ opens when Switch $k + 1$ is occupied; all doors open when Switch 1 is occupied. Agents need to take the same steps as in Pass, but SecretRoom is harder because there are three rooms on the right to explore, with only one being the target room. This task adds an exploration dimension on top of Pass—agents must distinguish the true target room from decoy rooms while maintaining coordination.

**MultiRoom.** Three agents must coordinate through four nested layers of doors. The door-switch rules are as follows: Door 1 opens when Switch 1 is occupied; Door 3 opens when Switch 2 is occupied; Door 2 opens when Switch 4 is occupied; Doors 4 and 5 open when Switch 3 is occupied. To achieve the goal, agents need to execute sequentially: one agent reaches Switch 1 and lets another agent enter the room containing Switch 2 through Door 1; one agent reaches Switch 2 and lets another agent enter the room containing Switch 4 through Door 3; one agent reaches Switch 4 and lets another agent enter the target room through Door 2; one agent reaches Switch 3 and lets the other two agents enter the target room through Door 4 or Door 5. This task requires more complex coordinated exploration among three agents, with a longer dependency chain where any agent's early mistake propagates and prevents team success.

In GridWorld, each agent observes its own position $(x, y)$ and the open states of doors represented by 0 (closed) and 1 (open). Thus observation space dimensions are 3 (Pass), 5 (SecretRoom), and 7 (MultiRoom). The action space contains four actions: move up, move down, move left, move right. All tasks use fully sparse rewards—when all agents enter the target room, each agent receives a $+100$ reward, with no rewards otherwise. The maximum episode length is set to 300 steps. The communication topology uses a fully-connected graph.

### E.2. Overcooked

We evaluate IEC on the Overcooked coordination benchmark [4], which requires precise spatiotemporal coordination. We use the open-source Overcooked environment of Carroll et al. (2019) and select three maps—Base, Narrow, and Large. All tasks contain two agents separated by an impassable kitchen counter. The left agent has access to tomatoes and the serving area (gray region), while the right agent has access to dishes and the pot. Agents must cooperatively complete the cooking pipeline: put tomato into pot $\rightarrow$ cook $\rightarrow$ put soup into dish $\rightarrow$ pass items through counter $\rightarrow$ serve at serving area.

**Base.** Standard kitchen layout with a wide passing region in the middle of the counter. Agents can pass items at any position along the counter. This setting tests basic coordinated exploration—agents must discover the passing mechanism and learn to synchronize pick-up and drop-off. **Narrow.** Passing region restricted to a narrow point in the middle of the counter. This requires more precise spatiotemporal coordination—agents must rendezvous at the narrow exchange point, and any timing misalignment causes blockage. **Large.** Increased environment size with greater initial distance between agents. The larger space amplifies coordination challenges—agents need longer action sequences to rendezvous, and the exploration space grows significantly.

To emphasize the exploration-coordination trade-off and control difficulty, we make three modifications to the original environment. First, we restrict agents' observation ranges. In Base and Narrow, each agent can only observe items in its own room or at the middle of the counter. In Large, we do not restrict observation range because preliminary experiments showed the task was too difficult under restricted observations. Second, we remove the cooking time for soup (the original environment requires 20 timesteps), so agents obtain cooked soup immediately after interacting with the pot. Third, we set episodes to terminate immediately after one successful serve, rather than requiring as many serves as possible in a fixed-length episode.

We use the featurized state provided by the environment as observations and add observation range restrictions. Observation spaces contain 38 and 48 dimensions for Base and Narrow respectively. The action space contains six actions: move up, move down, move left, move right, interact, stay. All variants use sparse $+100$ reward with a maximum episode length of 300 steps.

---

[4]https://github.com/HumanCompatibleAI/overcooked ai

### E.3. StarCraft Multi-Agent Challenge

We evaluate IEC on the StarCraft Multi-Agent Challenge (SMAC) [5], a widely-used cooperative MARL benchmark. We use the open-source SMAC environment version 2.4.10 and select three maps: 2m_vs_1z, 3m, and 8m.

**2m_vs_1z.** Two Marines vs. one Zealot. Marines are ranged units with low health and damage; Zealot is a melee unit with high health and damage. The winning strategy requires "alternating fire" coordination—the Marine being chased by the Zealot continuously retreats while the other Marine shoots. If both Marines stand still or both retreat simultaneously, they get killed by the Zealot. **3m.** Three Marines vs. three enemy Marines. This scenario is symmetric and requires basic focus-fire coordination—agents should concentrate fire to eliminate enemy units rather than spreading attacks. **8m.** Eight Marines vs. eight enemy Marines. This map significantly amplifies coordination challenges—more units mean higher-dimensional joint action space and more complex inter-unit dependencies. Effective strategies require subteam formation, focus-fire prioritization, and positional control.

To emphasize the exploration-coordination trade-off, we customize the reward structure to a sparse setting. Original SMAC provides dense shaping rewards for each kill and damage dealt. In our setting, agents receive a $+200$ reward when winning the battle; in 3m and 8m, agents receive an additional $+10$ reward per enemy kill to maintain learnability; there are no damage shaping or other intermediate rewards.

Each agent observes the local environment including relative positions, health, shields, weapon cooldowns of allied/enemy units, and terrain features. Observations are partial—agents can only see units within sight range (default 9.0 game units). Observation dimensions depend on map size (approximately 48-dim for 2m_vs_1z, 80-dim for 3m, 178-dim for 8m). The action space is a hybrid discrete action space including movement (4 directions), stop, attack (one action per visible enemy unit), and no-op. Episodes terminate when one side wins or reaches maximum steps (120 for 2m_vs_1z, 150 for 3m and 8m). We use distance-based sparse communication graphs—agents can only exchange policy information with teammates within communication range $R_{\mathrm{comm}} = 8.0$ game units.

## F. More Experimental Results

### F.1. Dense Reward Tasks

To verify the effectiveness of IEC on dense-reward tasks, we conduct additional experiments on SMAC's 3s_vs_5z map. The original SMAC environment provides dense shaping rewards with a maximum reward value of 20. To reasonably scale intrinsic rewards under dense reward settings, we introduce a scaling coefficient $\beta$ to modulate the contribution of exploration and consensus signals. Specifically, the shaped reward is expressed as $r_{\mathrm{IEC}}^i = r_{\mathrm{ext}} + \beta(\alpha_{\mathrm{info}}r_{\mathrm{info}}^i + \alpha_{\mathrm{nov}}r_{\mathrm{nov}}^i - \lambda r_{\mathrm{con}}^i)$, where $\beta = 0.1$ ensures that intrinsic signals do not overwhelm the extrinsic task reward. We set $\alpha_{\mathrm{info}}$ to 0.2, and $\lambda$ is adaptively adjusted via primal-dual updates (initialized at 0 with maximum $\lambda_{\mathrm{max}} = 1.0$).

Our primary comparison focuses on fully decentralized methods where communication constraints hold during both training and execution. CTDE methods (e.g., QMIX, MAPPO) assume privileged access to global state during training, which violates our problem setting. We include MAPPO in Table 8 as a reference point; its poor performance (0% WR on 3s_vs_5z and MultiRoom) reflects the challenge of adapting centralized-critic methods to our modified sparse-reward protocol rather than inherent method weakness. A systematic comparison with CTDE methods under equivalent information access is orthogonal to our contribution and left for future work.

Table 8 shows the learning curves of IEC compared with IPPO and its ablation variants on 3s_vs_5z (averaged over 8 random seeds). IEC significantly outperforms the baseline IPPO, with a win rate slightly higher than IPPO+$r_{\mathrm{nov}}$ (using only novelty reward) and IPPO+$r_{\mathrm{expl}}$ (using dual exploration but without consensus constraint). This indicates that even in dense-reward tasks, the consensus constraint and dual exploration mechanism can still promote coordinated exploration, helping agents discover more effective cooperative strategies. Notably, the performance gain of IEC under dense reward settings is smaller than under sparse reward settings, which is expected—when extrinsic shaping signals are sufficient, exploration pressure is relatively relieved, but the coordination mechanism still provides stable performance gains.

---

[5]https://github.com/oxwhirl/smac

*Table 8.* Density Experiment Performance

| Scenario | Parameters | | | IPPO | | MAPPO | | MASER | | MACE | | IEC | |
| --- | --- | --- | --- | --- | --- | --- | --- | --- | --- | --- | --- | --- | --- |
| | $N$ | $B$ | Steps | WR% | AUC | WR% | AUC | WR% | AUC | WR% | AUC | WR% | AUC |
| 2m vs 1z | 8 | 600 | $5e^6$ | $75_{\pm5.2}$ | $64.3_{\pm5.7}$ | $78_{\pm4.5}$ | $67.5_{\pm5.1}$ | $85_{\pm3.8}$ | $74.6_{\pm4.2}$ | $\mathbf{100.0}_{\pm0.0}$ | $85.3_{\pm2.8}$ | $\mathbf{100.0}_{\pm0.0}$ | $87.8_{\pm2.1}$ |
| 3s vs 5z | 8 | 600 | $5e^6$ | $15.2_{\pm5.8}$ | $12.3_{\pm6.2}$ | $0.0_{\pm0.0}$ | $0.0_{\pm0.0}$ | $0.0_{\pm0.0}$ | $0.0_{\pm0.0}$ | $44.5_{\pm7.2}$ | $36.8_{\pm7.8}$ | $\mathbf{48.8}_{\pm6.5}$ | $40.1_{\pm7.0}$ |
| MultiRoom | 128 | 300 | $1e^7$ | $0.0_{\pm0.0}$ | $0.0_{\pm0.0}$ | $0.0_{\pm0.0}$ | $0.0_{\pm0.0}$ | $0.0_{\pm0.0}$ | $0.0_{\pm0.0}$ | $\mathbf{100.0}_{\pm0.0}$ | $85.7_{\pm3.5}$ | $\mathbf{100.0}_{\pm0.0}$ | $88.4_{\pm2.7}$ |

**Note:** $N$: the number of parallel environments, $B$: the rollout buffer length, and Steps: the total number of environmental time steps.

## F.2. Choice of Divergence Measure

The consensus penalty $r_{\mathrm{con}}^i = D(\pi_i, \bar{\pi}_{i,t})$ requires choosing a divergence measure $D$. We compare three candidates: Jensen-Shannon divergence $D_{\mathrm{JS}}$ (our default choice), Kullback-Leibler divergence $D_{\mathrm{KL}}(\pi_i\|\bar{\pi}_{i,t})$, and Wasserstein distance $W_1$. All experiments are conducted on MultiRoom with other hyperparameters kept at default.

$D_{\mathrm{JS}}$ performs best because it is symmetric, bounded (in $[0,1]$ with log base 2), and numerically stable. $D_{\mathrm{KL}}$ performs slightly worse because its asymmetry can introduce bias when neighbor aggregation is uneven, and it can suffer numerical issues when supports of $\pi_i$ and $\bar{\pi}_{i,t}$ mismatch. $W_1$ performs comparably to $D_{\mathrm{JS}}$ but is computationally more expensive (requiring solving an optimal transport problem), impractical in our fast policy update setting. Figure E.4 compares the learning curves and Table E.7 reports final rewards. Based on these results, we recommend $D_{\mathrm{JS}}$ as the default divergence for consensus penalty.

## F.3. Hyperparameter Sensitivity Analysis Setup

To evaluate the robustness of the IEC framework and provide practical guidance for hyperparameter tuning, we conducted a systematic sensitivity analysis on the two most critical hyperparameters governing the primal-dual mechanism: the consensus budget $\delta$ and the dual update stepsize $\eta_\lambda$.

**Experimental Protocol.** We performed a grid search over one hyperparameter at a time while holding all other hyperparameters fixed at their default values (as specified in Table 7). For each configuration, we executed 5 independent training runs with different random seeds to capture statistical variance. The analysis was conducted across three representative environments to ensure cross-domain validity.

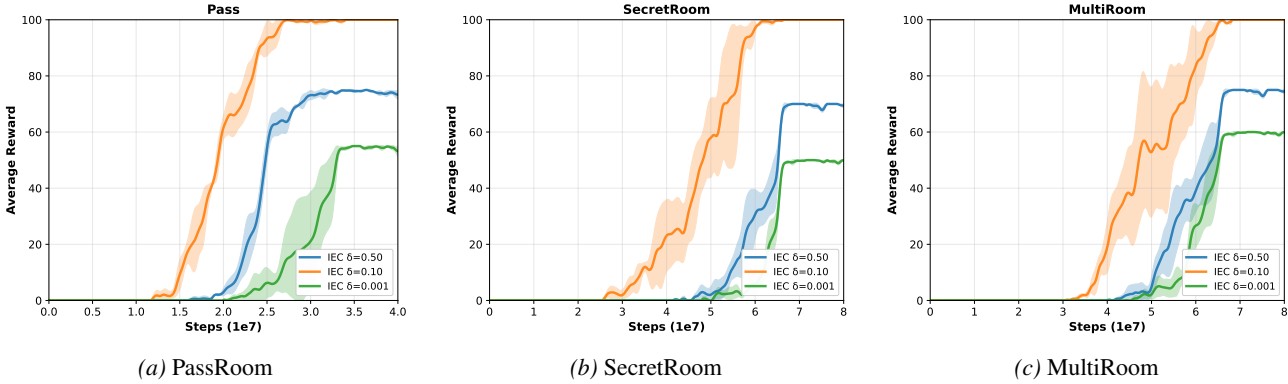

| *(a)* PassRoom | *(b)* SecretRoom | *(c)* MultiRoom |

*Figure 3.* **Sensitivity to the consensus budget $\delta$.** We sweep the disagreement budget $\delta$ (defined as the allowable upper bound on graph-averaged Jensen-Shannon divergence) while keeping all other hyperparameters fixed. The results (mean $\pm$ std over 5 seeds) exhibit an inverted-U trend: excessively tight budgets ($\delta \leq 0.01$) over-constrain the policy, suppressing necessary exploration and causing premature collapse; conversely, loose or unbounded budgets ($\delta \geq 0.5$ or $\infty$) fail to enforce topological consistency, leading to behavioral fragmentation. The default value $\delta \approx 0.1$ achieves the optimal trade-off across all benchmarks.

- **Consensus Budget ($\delta$):** We varied $\delta \in \{0.001, 0.01, 0.05, 0.1, 0.2, 0.5\}$ to study the impact of the strictness of the spectral consensus constraint. Additionally, we included an unconstrained baseline ($\delta = \infty$, equivalent to $\lambda = 0$) to isolate the contribution of the constraint mechanism. The default value was set to $\delta = 0.1$ for most tasks.

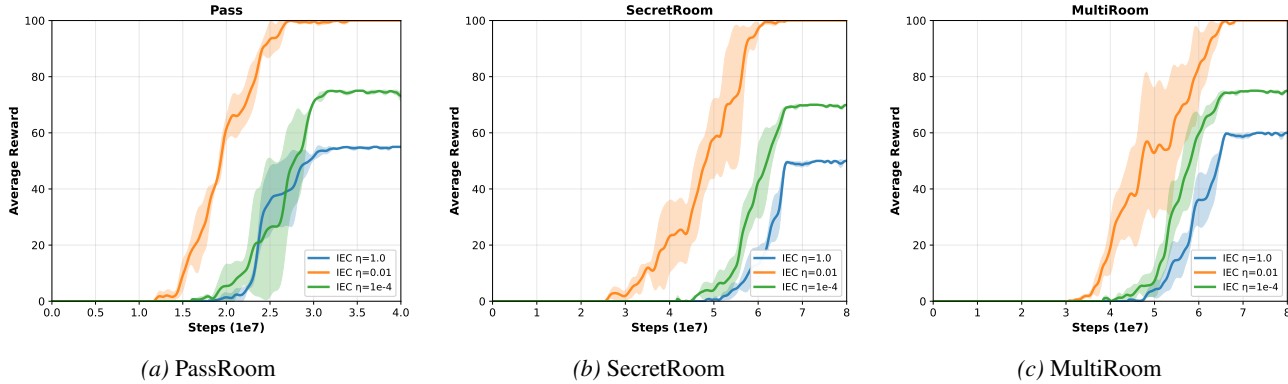

*(a)* PassRoom    *(b)* SecretRoom    *(c)* MultiRoom

*Figure 4.* **Sensitivity to the dual stepsize $\eta_\lambda$.** We analyze the impact of the dual learning rate $\eta_\lambda$ by varying it across logarithmic scales while fixing the consensus budget. The results (mean $\pm$ std over 5 seeds) demonstrate that IEC is robust to $\eta_\lambda$ within a broad operating range ($[10^{-3}, 10^{-1}]$). However, performance degrades at extremes: overly small stepsizes ($\eta_\lambda \leq 10^{-4}$) lead to sluggish adaptation, failing to curb constraint violations in time; conversely, excessively large stepsizes ($\eta_\lambda \geq 1.0$) induce destabilizing oscillations in the shaped reward signal, hindering policy convergence. The default value $\eta_\lambda = 0.01$ ensures stable and responsive adaptation.

*Table 9.* **Sensitivity Analysis on Consensus Budget $\delta$ and Dual Stepsize $\eta_\lambda$.** The parameter names are centered to indicate the variable being swept. We report the performance (mean $\pm$ std over 5 seeds) while fixing other hyperparameters.

| | **Param Value** | **0.001** | **0.01** | **0.05** | **0.1** | **0.2** | **0.5** |
|---|---|---|---|---|---|---|---|
| **Consensus Budget ($\delta$)** | GridWorld (Success) | $55.7_{\pm 4.1}$ | $85.1_{\pm 3.5}$ | $96.8_{\pm 1.2}$ | $\mathbf{100.0}_{\pm 0.0}$ | $91.5_{\pm 2.4}$ | $74.0_{\pm 3.8}$ |
| | Overcooked (Score) | $45.2_{\pm 5.3}$ | $78.6_{\pm 4.1}$ | $\mathbf{98.5}_{\pm 1.5}$ | $92.1_{\pm 2.3}$ | $84.3_{\pm 3.6}$ | $65.7_{\pm 4.5}$ |
| | SMAC (Win Rate) | $58.7_{\pm 3.2}$ | $79.4_{\pm 2.8}$ | $88.2_{\pm 2.1}$ | $94.5_{\pm 1.9}$ | $\mathbf{96.1}_{\pm 1.4}$ | $81.3_{\pm 3.1}$ |
| | **Param Value** | $1e^{-4}$ | $1e^{-3}$ | $1e^{-2}$ | $5e^{-2}$ | $1e^{-1}$ | **1.0** |
| **Dual Stepsize ($\eta_\lambda$)** | GridWorld (Success) | $74.0_{\pm 3.3}$ | $93.5_{\pm 1.8}$ | $\mathbf{100.0}_{\pm 0.0}$ | $96.4_{\pm 1.5}$ | $92.3_{\pm 2.1}$ | $56.0_{\pm 6.7}$ |
| | Overcooked (Score) | $72.4_{\pm 4.8}$ | $91.2_{\pm 2.2}$ | $95.8_{\pm 1.7}$ | $\mathbf{98.1}_{\pm 1.6}$ | $90.5_{\pm 3.1}$ | $42.6_{\pm 8.2}$ |
| | SMAC (Win Rate) | $75.6_{\pm 3.5}$ | $90.8_{\pm 2.0}$ | $\mathbf{95.9}_{\pm 1.5}$ | $94.2_{\pm 1.8}$ | $88.7_{\pm 2.5}$ | $61.3_{\pm 5.4}$ |

- **Dual Stepsize ($\eta_\lambda$):** We varied $\eta_\lambda \in \{10^{-4}, 10^{-3}, 10^{-2}, 5 \times 10^{-2}, 10^{-1}, 1.0\}$ to assess the sensitivity of the algorithm to the adaptation speed of the Lagrange multiplier $\lambda$. This range covers orders of magnitude from slow, conservative updates to aggressive, fast-reacting adjustments. The default value was $\eta_\lambda = 0.01$.

Complementing the visual trends in Figures 3 and 4, which display training dynamics for **representative** hyperparameter settings (selecting distinct low, medium, and high values to maintain legibility), Table 9 details the exact numerical performance across the **full spectrum** of the swept parameters.

As evidenced by the comprehensive color-coded rankings in Table 9, both the consensus budget $\delta$ and dual stepsize $\eta_\lambda$ exhibit a distinct inverted-U performance trend. Specifically, excessively tight budgets ($\delta \leq 0.01$) over-constrain the policy, stifling the exploration necessary for tasks like GridWorld and leading to premature convergence. Conversely, loose budgets ($\delta \geq 0.5$) fail to enforce sufficient coordination, resulting in fragmented behavior similar to unconstrained baselines. Similarly, the dual stepsize $\eta_\lambda$ requires a balance: while IEC is robust within a broad 'sweet spot' (e.g., $\eta_\lambda \in [10^{-3}, 10^{-1}]$), extreme values degrade performance either due to sluggish adaptation ($\eta_\lambda \leq 10^{-4}$) or destabilizing reward oscillations ($\eta_\lambda \geq 1.0$). Overall, the results confirm that IEC achieves high performance without requiring brittle, task-specific fine-tuning.

# G. Limitations and Future Work

Despite its effectiveness, IEC acknowledges several inherent limitations. First, the spectral consensus constraint relies on divergence-based proxies (e.g., Jensen-Shannon) which, while scalable, only approximate the true graph Laplacian quadratic form, potentially leading to inconsistent gradient scales across varying topologies and action spaces. Second, although the primal-dual mechanism adapts the multiplier $\lambda$, the constraint budget $\delta$ and dual step size $\eta_\lambda$ remain empirical

hyperparameters lacking theory-guided rules for cross-task transfer. Third, while we establish local stationarity at stable fixed points, providing strict global convergence guarantees in the non-convex, non-stationary deep MARL regime remains analytically intractable. Finally, the framework assumes idealized synchronized communication, abstracting away real-world deployment challenges such as asynchronous updates, time-varying delays, quantization errors, and privacy constraints.

Future research will focus on enhancing IEC's theoretical depth and practical scalability. A primary direction is to establish a rigorous quantitative bridge between divergence-based penalties and spectral graph theory via second-order approximations, utilizing meta-learning or control-theoretic approaches to dynamically adapt the constraint budget $\delta$ and step size $\eta_\lambda$ across training phases. We also plan to systematize value-weighted optimal perception by explicitly gating exploration with advantage estimates, thereby strictly internalizing theoretical insights into the algorithm. To address deployment realism, we will extend the framework to handle asynchronous, bandwidth-limited, and privacy-preserving communication regimes. Finally, we aim to move beyond fixed interaction graphs by incorporating learnable topology mechanisms that balance global smoothing with local role specialization, ensuring that beneficial heterogeneity is preserved in complex team tasks.

