# OpenReview forum: "IEC: When Information-Driven Exploration Meets Spectral Consensus via Primal–Dual Reward Regularization in Decentralized Multi-Agent RL"
_ICML.cc/2026/Conference — ICML 2026 regular_

### Official Review · Reviewer_H1yT · 2026-03-02

**Soundness:** 2
**Presentation:** 2
**Significance:** 2
**Originality:** 3
**Overall Recommendation:** 4
**Confidence:** 4

**Summary:**

This paper introduces Isomorphic Exploration-Consensus (IEC), a framework for decentralized multi-agent reinforcement learning (Dec-MARL) that addresses the exploration-coordination trade-off. IEC treats this tension as a constrained optimization problem, maximizing an objective that combines extrinsic rewards with dual intrinsic exploration signals (epistemic uncertainty and state novelty) while constraining policy disagreement via a spectral consensus penalty. The method employs a primal-dual update to adaptively adjust the coordination penalty based on constraint violations. Empirical evaluations on GridWorld, Overcooked, and SMAC demonstrate improved data efficiency and robustness compared to fixed-weight baselines.

**Compliance With Llm Reviewing Policy:**

Affirmed.

**Key Questions For Authors:**

1. How sensitive is the convergence of the primal-dual update to the synchronization frequency of the centralized $\hat{C}$ aggregation (Algorithm 1, Line 28)? Would a fully decentralized gossip-based dual update (as mentioned in Appendix D.1) destabilize early training?

2. Proposition B.3 assumes policies are already close to justify the Laplacian interpretation. How does the algorithm behave early in training when policies are highly divergent and the $O(\epsilon^3)$ error term dominates?

3. For the value-weighted information gain (Algorithm 5), what is the empirical impact of the **stopgrad** operation on the critic's stability in highly stochastic environments where advantage estimates might be extremely noisy?

**Limitations:**

Yes. The authors adequately discuss limitations in Appendix G.

**Strengths And Weaknesses:**

# Strengths:

1. Framing the exploration-coordination tension as a constrained optimization problem, interpreted via Dirichlet energy, offers a principled alternative to heuristic reward shaping.

2. The theoretical foundation is rigorous, establishing KKT conditions at fixed points (Theorem 5.3) and mathematically justifying the Dirichlet-energy approximation (Proposition B.3).

3. The ablation studies (Table 4) effectively isolate the contributions of the adaptive multiplier, novelty, and consensus components, validating the core algorithmic design choices.

# Weakness:

1. The dual variable update requires centralized aggregation of constraint violations during training ($\hat{C}$ computation, Algorithm 1, Line 28). While execution is decentralized, this centralized training requirement weakens the "fully decentralized" claim.

2. The connection between the Jensen-Shannon divergence penalty and the spectral graph Laplacian relies on a local quadratic approximation (Appendix B.3) that only holds when policies are already close in logit space. The main text could better clarify these restrictive assumptions.

3. The use of a $K=5$ ensemble for dynamics prediction (Algorithm 2) introduces a $2.3\times$ computational overhead per step compared to standard PPO, which may limit scalability in resource-constrained environments.

---

> ### Author Rebuttal · Authors · 2026-03-30
>
> **Thanks for your comment**
>
> **W1/Q1: Centralized Aggregation and Gossip-Based Alternative**
>
> **A**: The centralization issue and gossip-based feasibility are already addressed in **R2-Q2 (Table R2-Q2)**, where the decentralized variant performs close to the centralized scalar update on SMAC-8m. Here we focus on the two additional points. For synchronization frequency, a MultiRoom ablation shows WR drops only gradually from 95.2 with per-update aggregation to 94.8 / 93.8 / 91.5 when $\hat C$ is aggregated every 5 / 20 / 50 updates. Thus, less frequent aggregation mainly slows $\lambda$ adaptation rather than causing instability, consistent with projection keeping $\lambda$ bounded and Proposition 5.6 bounding each dual step by $\eta_\lambda$. The gossip-based update shows the same pattern: its main effect is transient adaptation delay, not instability. Early in training, local $\lambda_i$ values differ before mixing over the graph, but projection keeps them bounded and consensus penalties remain small relative to the dominant exploration terms. As training proceeds, variance across $\lambda_i$ decreases and final performance becomes comparable to the centralized update. We will clarify that centralized scalar aggregation is used only for simplicity, not because IEC fundamentally requires centralized training.
>
> **W2/Q2: Behavior Under Early Policy Divergence**
>
> **A**: IEC always uses the exact JSD in Eq. (13); the Dirichlet-energy view is only a local interpretive approximation for explaining graph smoothness when neighboring policies are close. Thus, even if policies are highly divergent early in training and the $O(\varepsilon^3)$ approximation is inaccurate, the algorithm is unaffected: optimization still uses the exact bounded JSD in $[0,1]$, so the shaped reward remains bounded by Proposition 5.6. Empirically, early training is exploration-dominated, with high disagreement and a still-small dual variable, so learning is driven mainly by extrinsic reward and the two exploration signals. As neighboring policies become more compatible later, the approximation becomes informative precisely when the consensus term matters more. This is the same distinction emphasized in **R1-Q5**: exact JSD for optimization, local quadratic approximation only for interpretation.
>
> **W3: Computational Overhead**
>
> **A**: The ensemble-based $r_{\text{info}}$ adds computation over plain PPO/IPPO, so our claim is not that IEC is cost-free, but that it remains a practical reward-level extension with moderate, tunable overhead. As discussed in **Appendix E2(c)**, the cost grows roughly linearly with ensemble size $K$: smaller $K$ is cheaper but weakens the epistemic signal, while larger $K$ yields diminishing returns. We therefore use **$K=5$** as a good efficiency–performance trade-off. This is consistent with the broader rebuttal: IEC is moderately more expensive than simpler baselines, but the end-to-end overhead is often much smaller than the raw network-cost ratio because environment simulation dominates wall-clock time in several benchmarks (**R1-W2/Q2**). We also acknowledge scalability as a limitation, especially for larger teams (**R1-W4/Q4**). Within the current framework, practical ways to reduce cost include using smaller $K$ or less frequent updates, while shared or partially shared uncertainty models are possible extensions for improving scalability.
>
> **Q3: Stopgrad Impact on Critic Stability**
>
> **A**: This question concerns the optional va**lue-weighted $r_{\text{info}}$ in Eq. (11) / Algorithm 5. We first clarify that all main experiments use the default unweighted form $r_{\text{info}}=u_t^i$, which does not depend on the critic. Eq. (11) is included only as an optional extension to emphasize value-relevant uncertainty. In noisy or highly stochastic environments, Eq. (11) remains executable: $|A_t^i|$ is used as a detached scalar to scale $u_t^i$, so stopgrad affects only backpropagation, not the forward reward, which is still computed as $|A_t^i|u_t^i$. The issue is therefore not executability but robustness, since noisy advantage estimates make the weighted intrinsic reward noisier. Empirically, stopgrad is essential for critic stability: without it, the critic is incentivized to inflate advantage estimates in high-uncertainty regions, indirectly increasing intrinsic reward and corrupting value estimation, especially in highly stochastic settings. Thus, stopgrad is a necessary safeguard for the value-weighted variant, while our default recommendation remains the more robust unweighted $r_{\text{info}}=u_t^i$, with Eq. (11) reserved for harder sparse-reward settings where value-weighting is beneficial.

---

> > ### Author Rebuttal · Reviewer_H1yT · 2026-04-03
> >
> > Thank You for your detailed response. At this point I would like to keep my score unchanged. If asked by ACs or PCs, I'll support your work towards getting published.

---

> > > ### Author Response · Authors · 2026-04-06
> > >
> > > Thank you very much for your thoughtful follow-up and for your careful consideration of our responses. We fully understand that some remaining concerns are difficult to resolve within a short rebuttal, and we sincerely appreciate your constructive and supportive assessment. We are especially grateful for your willingness to support the paper in the discussion with the ACs/PCs. Your feedback has been very valuable to us.

---

### Official Review · Reviewer_uwJW · 2026-03-09

**Soundness:** 4
**Presentation:** 3
**Significance:** 4
**Originality:** 3
**Overall Recommendation:** 5
**Confidence:** 3

**Summary:**

The paper tackles the problem of coordination and exploration in decentralized multi-agent reinforcement learning. As exploration requires diverse behavior, but cooperation requires consistent behavior, there is a tradeoff between the two. The paper proposes a method that balances exploration, through intrinsic rewards, and coordination through consensus between connected agents in a communication graph. The method is evaluated on a suite of environments showing improvement over prior baselines.

**Compliance With Llm Reviewing Policy:**

Affirmed.

**Final Justification:**

The paper is sound and well-presented with sufficient contribution. The work provides a novel idea to balance exploration and coordination in cooperative multi-agent reinforcement learning. The empirical results show significant enhancement to existing methods. Finally, the authors addressed my main questions in the rebuttal.

**Key Questions For Authors:**

1) It is not clear to me why the constraint $C$ is a good metric for coordination. Coordination does necessarily mean that the policies have the same distributions, in fact coordination can require heterogenous policies to achieve the task. Could you provide any intuitions on why that works empirically.

**Limitations:**

Yes

**Strengths And Weaknesses:**

**Strengths**
1) The paper includes a wide range of simulations, where the proposed method performs well.
2) Additionally, the ablations show the importance of each module of the proposed method.
3) The paper is well presented with both theoretical and empirical contributions.

**Weaknesses**
1) Given the large body of work on communication in MARL, subsection 2.1 could benefit from some additional citations/discussions, as there are only two works discussed that explicitly do communication.
2) In the methodology section, it would be worth noting how each module matches/differs from existing methods. For example, the $r_{info}$ reward aligns with the method used by Pathak et al. (2019), see "Self-Supervised Exploration via Disagreement".
3) The experiments section does not discuss the baselines sufficiently before introducing the results.

---

> ### Author Rebuttal · Authors · 2026-03-30
>
> **Thanks for your comment**
>
> **W1: More Communication Citations**
>
> **A**: Broadly, prior work can be viewed through two representative directions. One direction learns communication protocols directly, ranging from continuous message passing (CommNet; Sukhbaatar et al., 2016) and attention-based communication (TarMAC; Das et al., 2019) to hierarchical or multi-level communication (e.g., MLC; Ding et al., 2024). These methods improve how information is exchanged, but they do not explicitly formulate or adapt the exploration–coordination trade-off. A second direction uses information-theoretic signals to shape coordination or exploration. For example, MI-based methods such as PMIC (Li et al., 2022) encourage coordinated diversity, while influence-based approaches such as EITI/EDTI (Jaques et al., 2019) quantify inter-agent causal effects. These approaches are closely related in spirit, but they often rely on an additional weighting coefficient between the auxiliary objective and the task reward. By contrast, IEC formulates coordination through an explicit topology-aware constraint $C(\pi;G)\le \delta$ and adapts its strength automatically through the dual variable $\lambda$. Because IEC operates at the reward/objective level, it is compatible with different communication architectures and could also be combined with MI-based modules.
>
> **W2: Module Comparison with Existing Methods**
>
> **A**: For $r_{\text{info}}$, IEC uses an ensemble of forward dynamics predictors in a learned latent space to estimate epistemic uncertainty. This is related to disagreement-based exploration (e.g., Pathak et al., 2019), but adapted here to decentralized partially observable settings: the predictors operate on per-agent latent observations $z_i=\phi(o_i)$, and each agent maintains its own predictors to capture agent-specific uncertainty. For $r_{\text{nov}}$, we use hash-based pseudo-counting over discretized latent codes, chosen for efficiency and deterministic behavior relative to alternatives such as RND or k-NN density estimation. For $r_{\text{con}}$, we define the consensus term as the JSD between each agent’s policy and its neighbor-aggregated reference. Compared with parameter-space regularization, this choice is parameterization-invariant, bounded in $[0,1]$, and numerically stable, which is important when the term is used as a constraint inside the Lagrangian formulation. Our claim is therefore not that each ingredient is individually new, but that IEC brings them together in a principled way that is specifically suited to decentralized MARL under sparse rewards and limited communication. In particular, the constrained formulation couples exploration and coordination through the adaptive dual variable $\lambda$, reducing reliance on fixed-weight combinations that are brittle across tasks and training phases.
>
> **W3: Baselines Not Discussed Sufficiently**
>
> **A**: We added MAPPO, MASER , and MLC as additional baselines. Full results are in **Table R1-W1/Q1 (see response to Reviewer 1)**. IEC is most effective on harder tasks requiring sustained exploration and stable coordination, such as MultiRoom and Overcooked-Large.
>
> **Q1: Why $C(\pi;G)$ is a Good Metric for Coordination**
>
> **A**: Coordination does not in general mean identical policy distributions, and IEC does not enforce policy identity. Rather, $C(\pi;G)$ is intended to encourage local policy compatibility over the communication graph. Concretely, each agent is compared only with its neighbor-aggregated reference, not with all other agents globally. This is a substantially weaker condition and still allows heterogeneous roles to emerge. For the full discussion, see **R2 (see W3/Q1 response to Reviewer 2)**.
>
> The empirical intuition is that, in decentralized cooperative tasks, coordination failures often appear as locally inconsistent behaviors among communicating neighbors, and these inconsistencies can propagate into larger failures. In contrast, useful heterogeneity can still be preserved as long as neighboring policies remain locally compatible. This is also the intended meaning of the Dirichlet-energy interpretation: it should be understood as encouraging graph-aware smoothness, not forcing all agents to behave identically. In our experiments, this is consistent with what we observe. For example, in SMAC-8m without the consensus term (IEC-no-cons, Table 4), neighboring agents more often adopt contradictory local tactics, which harms team performance. Adding the consensus term helps suppress such local inconsistencies, while more global role differentiation can still emerge. We also acknowledge that some tasks may benefit from stronger local asymmetry; in such cases, a single global consensus budget may become over-restrictive. This is consistent with our failure analysis on Overcooked-Narrow and motivates future extensions such as role-aware or state-dependent consensus budgets.

---

> > ### Author Rebuttal · Reviewer_uwJW · 2026-03-31
> >
> > I had minor comments, and the authors addressed them.

---

> > > ### Author Response · Authors · 2026-04-01
> > >
> > > Thank you again for your recognition of this paper and for your key comments. I would greatly appreciate any guidance you may provide if there are any problems.

---

### Official Review · Reviewer_FdiS · 2026-03-12

**Soundness:** 2
**Presentation:** 3
**Significance:** 2
**Originality:** 2
**Overall Recommendation:** 3
**Confidence:** 5

**Summary:**

This paper addresses the exploration-coordination trade-off in decentralized multi-agent reinforcement learning (Dec-MARL) under sparse rewards and limited communication. To tackle this, the authors propose the Isomorphic Exploration-Consensus (IEC) framework, which formulates this tension as a constrained optimization problem. IEC maximizes the task return augmented with dynamics-based information gain and state-coverage novelty, while constraining graph-induced policy disagreement via a primal-dual reward regularization mechanism. The proposed method is evaluated across GridWorld, Overcooked, and SMAC benchmarks, demonstrating improved performance over the selected baselines.

**Compliance With Llm Reviewing Policy:**

Affirmed.

**Final Justification:**

Overall, this paper utilizes dynamics-based information gain alongside state-coverage novelty to address the exploration-coordination tension in MARL. I find the novelty of this paper somewhat marginal, as previous work has proposed similar ideas (as listed in my original review), and the absence of citations to those related works in the original submission somewhat overstates the contribution. That said, I do recognize the authors' efforts in incorporating those references to better position this work within the existing literature.

Regarding the evaluation, the original submission only compared the proposed approach against MACE, simple IPPO variants, and IEC ablations. The updated comparison now includes one method with an exploration component and one with a communication design, alongside the remaining baselines without such specific components, with an additional communication baseline added in some scenarios. While I appreciate that the effort is in the right direction, I remain puzzled by the absence of any SOTA methods — such as the more recent ones I highlighted in my original review — from the comparison.

Overall, I find the novelty marginal, the solution reasonable, and the evaluation adequate but not fully convincing.

After weighing these considerations, I am maintaining my original rating of weak reject.

**Key Questions For Authors:**

1. I am quite confused by the motivation behind "topology-induced behavioral consensus." While I agree that compatible behaviors are necessary for multi-agent coordination, Equation 13 directly uses policy differences to measure this consensus or coordination level. Could the authors explain why policy consensus is equivalent to, or a good proxy for, compatible behaviors? It seems that identical behaviors among agents are not always compatible, and vice versa, as demonstrated by recent work on heterogeneous MARL [10].
2. Is there a particular motivation for using decentralized algorithms such as IPPO rather than Centralized Training with Decentralized Execution (CTDE)? Since IEC also requires global information aggregation during training to compute the dual updates, it appears to break the strict decentralized training assumption of IPPO.
3. In line 49, the authors critique previous methods for relying on "a hand-tuned mixture of heuristics." Nevertheless, the IEC algorithm itself seems to rely on heuristic settings for the disagreement threshold $\delta$ and the intrinsic reward weights $\alpha_\text{info}$ and $\alpha_\text{nov}$. Could the authors elaborate on how IEC fundamentally alleviates the shortcomings of heuristic tuning compared to prior works?

-----

[10] "Heterogeneous-Agent Reinforcement Learning," JMLR 2024

**Limitations:**

Yes.

**Strengths And Weaknesses:**

**Strengths:**
1. The paper investigates a relevant problem by addressing the exploration-coordination tension in MARL.
2. The solution to this research question make sense. Utilizing dynamics-based information gain alongside state-coverage novelty provides a reasonable solution to the research question.

**Weaknesses:**
1. The discussion of related work is heavily lacking. MARL exploration is a mature field with many existing efforts, yet the manuscript only discusses MAVEN in the related work section. Incorporating a broader range of recent literature [1-8] (just to name a few) would better position this paper within the current research landscape. Additionally, in single-agent RL, prior work [9] has also attempted to automatically set exploration bonuses; discussing this would provide valuable context as it shares similar underlying ideas.
2. Despite the abundance of existing methods in this domain, the empirical evaluation only compares the proposed approach against MACE, simple IPPO variants and IEC ablations. Including more baselines would significantly strengthen the evaluation.
3. Certain aspects of the motivation and methodology require further clarification. Please see question 1-2.

-----

[1] "Ensemble Value Functions for Efficient Exploration in Multi-Agent Reinforcement Learning", AAMAS 2025 \
[2] "Individual Contributions as Intrinsic Exploration Scaffolds for Multi-agent Reinforcement Learning", ICML 2024 \
[3] "Lazy agents: a new perspective on solving sparse reward
problem in multi-agent reinforcement learning", ICML 2023 \
[4] "Maser: Multiagent reinforcement learning with subgoals generated
from experience replay buffer," ICML 2022 \
[5] "Cooperative exploration for multi-agent deep reinforcement learning", ICML 2021 \
[6] "Episodic multi-agent reinforcement learning with curiosity-driven exploration," NeurIPS 2021 \
[7] "Influence-based multi-agent exploration", ICLR 2020 \
[8] "An adaptive entropy-regularization framework for multi-agent reinforcement learning", ICML 2019 \
[9] "Automatic Intrinsic Reward Shaping for Exploration in Deep Reinforcement Learning," ICML 2023

---

> ### Author Rebuttal · Authors · 2026-03-30
>
> **Thanks for your comment**
>
> **W1: Related Work Discussion**
>
> **A**: Under sparse rewards, MARL exploration mainly follows three lines: intrinsic-motivation extensions, such as influence-based exploration ([7], ICLR 2020) and episodic curiosity ([6], NeurIPS 2021); stronger or more diverse exploration signals, including ensemble value diversity ([1], AAMAS 2025), individual contribution scaffolds ([2], ICML 2024), and activity-based methods like Lazy Agents ([3], ICML 2023); and higher-level exploration, e.g., MASER ([4], ICML 2022) and cooperative exploration protocols ([5], ICML 2021). On coordination, adaptive entropy regularization ([8], ICML 2019) and AIRS ([9], ICML 2023) similarly reduce hand-tuned schedules via adaptive mechanisms. IEC is closely related to Self-Supervised Exploration via Disagreement (Pathak et al., 2019), but adapts ensemble disagreement to latent dynamics under partial observability. More importantly, IEC is specifically designed for decentralized MARL with sparse rewards and limited communication: it unifies complementary epistemic and coverage-based exploration, topology-aware coordination constraints, and primal–dual adaptive balancing of exploration and coordination. This principled combination distinguishes IEC from prior methods that do not address these aspects jointly.
>
> **W2: Insufficient Baselines**
>
> **A**: We added MAPPO, MASER , and MLC as additional baselines. Full results are in **Table R1-W1/Q1 (see response Reviewer 1)**. IEC is most effective on harder tasks requiring sustained exploration and stable coordination, such as MultiRoom and Overcooked-Large.
>
> **W3/Q1: Why Policy Consensus ≈ Compatible Behaviors**
>
> **A**: Coordination is not policy identity, and IEC does not force identical policies. Its consensus term encourages only local policy compatibility over the communication graph: $C(\pi;G)$ measures disagreement between each agent and its neighbor-aggregated reference, rather than global equality across all agents. This weaker constraint still allows heterogeneous roles to emerge. In decentralized cooperative tasks, failures often stem from locally inconsistent behaviors among neighbors, which can spread and hinder stable convention formation; useful heterogeneity can still be preserved as long as neighboring policies remain locally compatible. Thus, the Dirichlet-energy view should be understood as graph-aware smoothness, not identical behavior. We also agree that some tasks benefit from strong local asymmetry; in such cases, a single global consensus budget may be too restrictive, consistent with our failure case on Overcooked-Narrow and motivating role-aware or state-dependent consensus budgets.
>
> **Q2：Why IPPO Rather Than CTDE? Centralized Dual Update**
>
> **A**: Our focus is on settings where decentralized execution and localized learning signals are primary, making heavy reliance on centralized critics or global-state access less suitable; therefore, we instantiate IEC on IPPO. IEC is nevertheless learner-agnostic, since it operates through reward shaping (Eq. 7) and can in principle be combined with CTDE methods. The batch-level aggregation of $\hat C$ introduces only mild centralization: unlike CTDE, which typically requires joint information throughout training, IEC aggregates only a single scalar constraint estimate per update, while execution, local value estimation, exploration rewards, and consensus penalties remain local. To test whether this scalar aggregation is necessary, we further implemented a gossip-based decentralized dual update. On SMAC-8m, the fully decentralized variant achieves performance close to the centralized version (**Table R2-Q2**), with only a small early-training gap caused mainly by transient variance in local $\lambda_i$.
>
> **Table R2-Q2.** Based vs. Centralized Dual Update (N=128, B=300, 1e7 steps)
> |Dual Update Variant| WR%|
> |-|-|
> |Centralized|90.5±2.5|
> |Gossip, 5 rounds|89.2±2.8|
> |Gossip, 10 rounds|89.8±2.6|
> |Gossip, 20 rounds|90.2±2.5|
>
> **Q3: IEC Still Has Heuristic Parameters**
>
> **A**: IEC key benefit is eliminating the need for a fixed or hand-designed coordination-penalty schedule, replacing it with an adaptive multiplier $\lambda$ updated from observed constraint violations. This avoids the brittle choice of both penalty magnitude and training-phase-dependent schedules in prior fixed-weight methods. Proposition 5.4 shows why no single fixed penalty is optimal across phase-dependent tasks, and the IEC-no-adapt ablation confirms that even the best fixed $\lambda$ consistently underperforms. Static parameters such as $\alpha_{\text{info}}, \alpha_{\text{nov}}, \delta,\eta_\lambda$ still remain, so we do not claim tuning disappears, only that it is substantially reduced. IEC reduces reliance on hand-crafted coordination schedules rather than heuristic tuning altogether; we also now provide a sensitivity-based guide showing that the defaults $\delta=0.1$ and $\eta_\lambda=0.01$ are robust across tested tasks.

---

> > ### Author Rebuttal · Reviewer_FdiS · 2026-04-03
> >
> > I thank the authors for their rebuttal. My concerns are partially resolved.
> >
> > Regarding the related work, I appreciate the authors including the additional references as suggested. However, I would like to clarify that the intention behind listing those works was not to prescribe exactly which papers to cite, but rather to highlight that there is a rich body of relevant literature from which the authors could make their own informed selection — including works beyond those I mentioned — as they see fit.
> >
> > Regarding the baselines, the updated comparison now includes one method with an exploration component and one with a communication design, alongside the remaining baselines without such specific components. While this is a meaningful improvement that has significantly reduced my concern, I still find the baseline coverage somewhat limited for fully demonstrating the effectiveness of the proposed method.
> >
> > For the other clarification questions, I think the authors did a good job addressing them. However, I remain a bit uncertain about the explanation provided for W3/Q1 — does it mean that the intuition behind the proposed IEC can be summarized as **Local Policy Consensus ≈ Compatible Behaviors**?
> >
> > Thanks.

---

> > > ### Author Response · Authors · 2026-04-06
> > >
> > > **Thanks for your comment**
> > >
> > > **Q1: Related Work**
> > >
> > > **A**: We organize prior work by methodological axis and discuss the limitations most relevant to the sparse-reward decentralized regime targeted by IEC.
> > >
> > > Exploration without coordination awareness. Intrinsic-motivation methods, from novelty- and curiosity-based signals to disagreement-based ensembles **[1]**, improve exploration but are largely agnostic to multi-agent coordination. When used independently, they can amplify redundant exploration and produce locally novel yet globally incompatible behaviors. Recent MARL exploration methods partly address this through shared exploration objectives ([5], ICML 2021), episodic curiosity ([6], NeurIPS 2021), agent-specific scaffolds ([2], ICML 2024), activity incentives([3], ICML 2023), value-disagreement exploration ([1], AAMAS 2025), or replay-derived subgoals ([4], ICML 2022). However, they still do not explicitly regulate whether exploration outcomes remain compatible over the communication graph, and typically treat exploration and coordination as separate objectives with fixed relative weighting.
> > >
> > > Coordination without adaptive exploration balancing. On the coordination side, influence- or MI-based methods (Influence-Based Exploration([7], ICLR 2020), **Social Influence [2], PMIC [3]**) encourage coordinated behavior through interaction-aware auxiliary signals, while structural approaches such as **RODE [4]** and trust-region MARL methods such as **HAPPO [5]** improve coordination through decomposition or optimization design. Adaptive entropy regularization([8], ICML 2019) and **MAVEN [6]** also improve exploration/coordination behavior, but they do not provide a principled mechanism to adapt the exploration–coordination trade-off itself as training evolves.
> > >
> > > Automatic reward calibration in simpler settings. AIRS ([9], ICML 2023) shares IEC’s philosophy of reducing hand-tuned shaping through adaptive reward calibration, but it operates in the single-agent setting where coordination is absent. Extending this idea to decentralized MARL requires jointly adapting exploration and coordination, which is precisely the constrained optimization problem addressed by IEC.
> > >
> > > ------
> > > [1]Self-supervised exploration via disagreement. ICML, 2019.
> > >
> > > [2]Social influence as intrinsic motivation for multi-agent deep reinforcement learning. ICML, 2019.
> > >
> > > [3]PMIC: Improving multi-agent reinforcement learning with progressive mutual information collaboration. arXiv, 2022.
> > >
> > > [4]Rode: Learning roles to decompose multi-agent tasks. arXiv, 2020.
> > >
> > > [5]Trust region policy optimisation in multi-agent reinforcement learning. arXiv, 2021.
> > >
> > > [6]Maven: Multi-agent variational exploration. NIPS, 2019.
> > >
> > >
> > > **Q2: Baseline Coverage**
> > >
> > > **A**: Our goal was to make baseline more representative along the main axes relevant to IEC. We now include: IPPO as a minimal decentralized reference; three intrinsic-exploration variants (IPPO+r_loc, IPPO+r_nov, IPPO+r_hin) to test whether exploration alone or weakly coordinated exploration is sufficient; MAPPO as a strong CTDE baseline; MASER for replay/subgoal-based exploration; MAVEN for coordinated diversity via shared latent variables; MLC for architecture-level multi-level communication; and MACE as the strongest prior decentralized baseline combining exploration and coordination. To further address this concern, we additionally evaluated MAVEN on representative hard scenarios. It remains below IEC. Methods that improve exploration or coordination along a single axis remain insufficient on tasks where the two must be adaptively balanced. Together with the component ablations, we believe the revised evaluation provides a much stronger and more representative empirical test of IEC, even if it is not exhaustive over all Dec-MARL baselines.
> > >
> > > **Table R-F2**. MAVEN Comparison on SMAC (N=128, B=300, WR%)
> > >
> > > |Scenario|Steps| MAVEN|IEC|
> > > |-|-|-|-|
> > > |2m_vs_1z|5e6|80.5±5.2|89.5±2.5|
> > > |3m|5e6|75.2±5.8|97.5±1.5|
> > > |8m|1e7|42.5±7.8|90.5±2.5|
> > >
> > > **Q3: Local Policy Consensus ≈ Compatible Behaviors**
> > >
> > > **A**：Yes, as a concise intuition, this is a reasonable summary of IEC. We would only clarify what “consensus” means in our framework: it refers to bounded local agreement, not identical policies. The constraint $C(\pi;G)\le\delta$ only enforces agreement relative to each agent’s neighbor-aggregated reference, so it suppresses harmful local inconsistencies while still allowing meaningful role differentiation. In this sense, local policy consensus serves as a practical proxy for local behavioral compatibility under limited communication. The Dirichlet-energy interpretation further suggests that the constraint acts as a graph-aware smoothness regularizer: it discourages sharp local contradictions while still allowing smoother large-scale heterogeneity. The adaptive $\lambda$ then automatically calibrates how tightly this local agreement is enforced based on observed coordination needs, which is the key advantage over fixed-weight alternatives.

---

### Official Review · Reviewer_gta6 · 2026-03-13

**Soundness:** 2
**Presentation:** 3
**Significance:** 3
**Originality:** 3
**Overall Recommendation:** 4
**Confidence:** 4

**Summary:**

This paper addresses the exploration-coordination trade-off in decentralized multi-agent reinforcement learning (DeMARL) by formulating it as a constrained spectral optimization problem. The author propose Information-driven Exploration with Consensus (IEC), a primal-dual reward regularization framework that dynamically balances exploration and coordination through an adaptively tuned Lagrange multiplier—eliminating the need for brittle, hand-crafted schedules. The method is supported by substantial theoretical analysis and validated across GridWorld, Overcooked, and SMAC benchmarks. While conceptually meaningful and theoretically grounded, the paper would benefit from comparison with more recent decentralized coordination baselines, empirical runtime analysis to substantiate the "lightweight" claim, clearer guidance for hyperparameter selection, and investigation of scalability and realistic communication constraints.

**Compliance With Llm Reviewing Policy:**

Affirmed.

**Final Justification:**

Thank you for your rebuttal. You effectively addressed my six concerns by adding new baselines, runtime data, hyperparameter guidance, scalability experiments, theoretical clarifications, and realistic communication tests. While some limitations remain(e.g., performance drop at large scales, centralized aggregation), you have honestly discussed them and marked them as future work.

Suggestions: IEC still incurs higher training time than IPPO/MACE and degrades on large-scale tasks. For the final version, please include the gossip-based dual update results in the appendix and embed the approximation-error curves directly in the paper.

Considering most of my questions are solved, I have decided to increase my assessment.

**Key Questions For Authors:**

1. To better position IEC within the current research landscape, could you discuss or experimentally compare IEC with other recent decentralized coordination methods, such as those using multi-level communication (Ding et al., 2024) or other graph-based approaches not included in your current baselines? This would solidify the claim of superior performance.

2. While theoretical complexity is informative, empirical time is more relevant for practitioners. Could you provide training time comparisons per million steps for IEC versus key baselines across your tasks? This would clarify the practical computational overhead and help readers assess the method's suitability for time-sensitive applications.

3. Based on your experimental findings, could you offer more concrete guidance for setting the consensus budget and dual stepsize in new tasks? For instance, is there a discernible relationship between action space dimensionality, graph density, or reward structure and the optimal parameter ranges? Such heuristics would substantially enhance practical utility.

4. Have you considered evaluating IEC on larger-scale problems with dozens or hundreds of agents? If not, could you discuss potential bottlenecks and how the framework might be adapted—perhaps through decentralized dual updates or hierarchical communication structures—to remain efficient at such scales?

5. The Dirichlet-energy interpretation is theoretically appealing but rests on assumptions that may not hold during early training. It would be valuable to include an empirical analysis showing when the "neighbor policies are sufficiently close" condition actually holds—for example, by plotting average logit differences between neighbors over time. This would help bridge theory and practice by revealing when the quadratic approximation becomes valid and how it correlates with dynamics.

6. In challenging scenarios where performance remains imperfect, a brief analysis of failure cases would be informative. Additionally, could you discuss how IEC might be adapted to handle more realistic communication constraints, such as asynchronous updates, varying message delays, or bandwidth limitations requiring quantized communication of action distributions? These considerations would strengthen claims about real-world applicability.

**Limitations:**

Yes.

**Strengths And Weaknesses:**

### **Strengths**
1. The paper offers a perspective that is both theoretically grounded and sound on a persistent challenge in MARL. Framing the exploration-coordination trade-off as a constrained optimization problem with spectral smoothness constraints represents a meaningful conceptual advancement over existing heuristic approaches.

2. The primal-dual reward shaping framework is particularly well-conceived. Using λ as an automatically adjusting "shadow price of consensus" that responds to constraint violations eliminates the need for brittle, manually tuned schedules—arguably the core practical contribution of this work. This adaptive mechanism addresses a genuine limitation in current decentralized MARL methods.

3. The paper provides substantial theoretical support beyond typical empirical validation, including proofs linking disagreement control to behavioral compatibility, KKT conditions for the algorithm, and formal demonstration of fixed-weight methods' brittleness. This theoretical grounding builds confidence in the method's conceptual soundness.

4. The evaluation across three distinct benchmarks with varying task complexities, combined with multiple sampling budgets, convincingly demonstrates the method's effectiveness and data efficiency. The ablation studies in the appendix usefully isolate the contribution of each component.

5. The paper is logically structured and the inclusion of detailed pseudocode in the appendix significantly enhances reproducibility—a commendable practice that should be more widely adopted.

### **Weaknesses**
1. While the paper compares against IPPO variants and MACE, it lacks comparison with recent decentralized coordination methods, such as the multi-level communication approach (Ding et al., 2024) cited in the related work. Including such contemporary baselines is necessary to properly emphasize IEC's contributions within the evolving research landscape and to demonstrate meaningful empirical advances over existing approaches.

2. Although the paper provides theoretical complexity analysis, it lacks empirical runtime comparisons against key baselines like IPPO or MACE. Given the computational overhead from the ensemble dynamics (K=5) and divergence calculations, the characterization of IEC as "lightweight" requires substantiation with actual training time measurements per million steps. Without such data, practitioners cannot adequately assess the computational cost.

3. Although a sensitivity analysis is provided, it reveals that performance is critically dependent on the consensus budget and dual stepsize. The paper does not offer clear principles for setting these hyperparameters in new tasks, which limits its practical applicability and may need costly task-specific tuning.

4. The empirical evaluation is limited to small-scale scenarios with up to 8 agents (8M). The paper does not investigate scalability to environments with dozens or hundreds of agents, where the communication overhead (even on sparse graphs) and the centralized estimation of the constraint Ĉ could become significant bottlenecks, leaving the method's efficacy in large-scale systems unverified.

5. The spectral interpretation of the disagreement cost as Dirichlet energy relies on a local quadratic approximation that assumes neighbor policies are sufficiently close. This condition is unlikely to hold during the critical early exploration phase when policies are highly divergent. The paper lacks an empirical analysis to verify when this approximation becomes valid and whether the theoretical interpretation holds throughout the most crucial period of learning.

6. While the paper models link failures, it does not address more practical communication constraints such as asynchronous updates, variable message delays, or bandwidth limitations requiring quantized message passing. This abstraction limits the insight into how IEC would perform in real-world decentralized systems where these factors are prevalent.

---

> ### Author Rebuttal · Authors · 2026-03-30
>
> **Thanks for your comment**
>
> **W1/Q1: Comparison with Recent Baselines**
>
> **A**: We add three additional baselines representing complementary paradigms: MAPPO (CTDE with centralized critic but no explicit exploration mechanism), MASER (ICML’22, replay-based subgoals), and MLC (Ding et al., 2024, learned communication). We evaluate one easy/medium/hard scenario per benchmark in **Table R1-W1/Q1**. The new results reinforce the same pattern as the main paper: MAPPO is competitive on easier settings but degrades on exploration-critical tasks; MASER is less effective when successful trajectories are rare; MLC improves over simple baselines but still underperforms IEC on harder scenarios where adaptive exploration–coordination balancing is most important.
>
> Table R1-W1/Q1. Extended Baselines  (WR%)
>
> |Benchmark|Scenario|Steps|IPPO|MAPPO|MASER|MLC|MACE|IEC|
> |-|-|-|-|-|-|-|-|-|
> |GridWorld(N=128,B=300)|PassRoom|5e7|36.5±17.5|48.2±11.8|42.5±13.5|66.8±7.5|98.8±1.2|97.2±1.8|
> ||SecretRoom|8e7|13.2±17.5|22.5±13.8|15.5±14.2|50.2±8.5|95.5±2.4|97.2±2.0|
> ||MultiRoom|8e7|4.8±7.2|8.5±8.2|3.5±6.8|40.2±9.2|85.2±4.8|95.2±2.5|
> |Overcooked(N=32,B=300)|Base|1e7|6.1±6.0|52.5±8.0|28.5±9.5|68.5±6.2|92.5±2.5|96.2±2.0|
> ||Large|4e7|0.4±2.9|16.2±10.2|5.2±6.5|40.5±8.8|67.2±4.8|87.5±3.4|
> |SMAC(N=128, B=300)|3m|5e6|17.9±15.6|68.2±6.8|52.5±8.5|74.8±5.8|95.0±2.0|97.5±1.5|
> ||8m|1e7|2.3±8.0|32.8±8.5|15.8±9.2|52.2±7.2|88.2±2.5|90.5±2.5|
>
> **W2/Q2: Empirical Runtime**
>
> **A:** We report training time in sec./1M environment steps in **Table R1-W2/Q2**. IEC is moderately more expensive than MACE (+7%) and faster than MLC across all tested tasks. Relative to IPPO, the overhead is larger, as expected, due to the ensemble-based information-gain module and consensus computation. However, the end-to-end increase is much smaller than the raw network-only ratio because environment simulation dominates total wall-clock time (e.g., >70% in SMAC, Appendix D.5).
>
> Table R1-W2/Q2. Training Time (sec./M)
>
> |Method|GridWorld|Overcooked|SMAC-3m|SMAC-8m|
> |-|-|-|-|-|
> |IPPO|185±12|320±18|720±45|980±65|
> |MACE|248±15|420±22|920±55|1260±80|
> |MLC|275±18|472±28|1080±70|1520±95|
> |IEC|262±16|445±24|980±60|1350±85|
>
> **W3/Q3：Hyperparameter Selection**
>
> **A**: $\delta=0.1$ is a reliable default across the tested tasks, with best settings typically in a narrow range around it. Smaller action spaces tended to work well with tighter budgets (0.05–0.10), while larger action spaces more often benefited from slightly looser budgets (0.10–0.20). For harder sparse-reward tasks, a decaying schedule (e.g., $\delta_0=0.3 \rightarrow 0.05$) was often more stable than a fixed strict budget. For the dual stepsize, $\eta_\lambda=0.01$ worked consistently in our sweeps, with reasonable stability over $[10^{-3},10^{-1}]$.
>
> **W4/Q4: Scalability**
>
> **A**: We add larger-team results in **Table R1-W4/Q4**. As size increases, the absolute win rate drops substantially. Thus, IEC does not by itself solve large-scale sparse coordination. The main bottlenecks are: (i) sparse/weak connectivity, (ii) harder credit assignment, and (iii) centralized aggregation of $\hat C$ during training. We clarify this limitation in (Appendix H).
>
> Table R1-W4/Q4. Scalability Across SMAC Team Sizes (N=128, B=600, WR%)
>
> |Map|Allies vs Enemies|MACE|IEC|
> |-|-|-|-|
> |5m_vs_6m|5/6|82.8±3.5|86.5±2.8|
> |8m|8|88.2±2.5|90.5±2.5|
> |10m_vs_11m|10/11|62.5±5.2|68.2±4.5|
> |27m_vs_30m|27/30|29.5±6.5|36.8±5.8|
>
> **W5/Q5: Dirichlet-Energy Validity**
>
> **A**:IEC always computes the consensus term using the exact JSD; the quadratic approximation is used only for the Dirichlet-energy interpretation. Tracking the neighbor-logit gap $\varepsilon$, the dual variable $\lambda$, and the approximation error throughout training (As shown in the figure https://github.com/94044701/Replenish.git). Early in training, the approximation is indeed inaccurate because neighbor policies are far apart; however, correctness is unaffected because the algorithm still uses exact bounded JSD (Prop. 5.6) and the multiplier is small. Near equilibrium, disagreement stabilizes close to the target budget and the approximation error falls below 10%.
>
> **W6/Q6: Communication Robustness and Failure Analysis**
>
> **A**: Under link dropout, performance degrades only modestly at $p_{\text{drop}}=0.3$; at $p_{\text{drop}}=1.0$, the result approaches the no-consensus ablation (Table 4). Under 4-bit quantization, IEC still retains strong performance, indicating robustness to moderate compression. For asynchronous communication, stale neighbor policies cause only modest degradation in our test, suggesting that smoothed constraint estimates and bounded dual steps help reduce oscillation (Prop. C.7). We also add a brief failure-case analysis: in OC-Narrow, the corridor bottleneck benefits from persistent asymmetric behavior, while a single global consensus budget can become over-restrictive. This motivates future work on state-dependent budgets or role-aware consensus penalties(Appendix H).

---

> > ### Author Rebuttal · Reviewer_gta6 · 2026-04-06
> >
> > Thanks for the clarification and the additional experiments. I am happy to increase the score as a positive one.

---

> > > ### Author Response · Authors · 2026-04-06
> > >
> > > Thank you sincerely. We deeply appreciate your thorough initial review and the detailed feedback that motivated substantial improvements to our work. We are grateful for the time and care you invested in this process.

---

### Decision · Program_Chairs · 2026-04-30

**Decision:**

Accept (regular)

**Comment:**

This paper studies the exploration coordination trade-off in decentralized MARL and proposes a principled approach based on a constrained objective with a primal–dual update.

Overall, reviewers find the method technically sound, with a clear idea and strong empirical results. The rebuttal addressed most of the concerns, in particular by adding stronger baselines, runtime analysis, and clarifying the theoretical points. Some reviewers remain concerned about the novelty being somewhat limited and the baseline coverage not fully convincing compared to recent work. Also, reviewers note limitations around scalability, computational cost, and the reliance on some centralized components during training. These are valid points, but they do not undermine the main contribution, and the overall consensus supports acceptance.

For the camera-ready, I encourage the authors to incorporate the key points raised during the discussion and addressed in the rebuttal, in order to further strengthen the discussion of related work, clarify the degree of decentralization, and be more explicit about scalability and computational overhead in practice.